# Factored Classifier-Free Guidance

**Tian Xia**[1]  **Fabio De Sousa Ribeiro**[1]  **Rajat R Rasal**[1]  **Avinash Kori**[1]  **Raghav Mehta**[1]  **Ben Glocker**[1]

## Abstract

Counterfactual generation aims to simulate realistic hypothetical outcomes under causal interventions. Diffusion models have emerged as a powerful tool for this task, combining DDIM inversion with conditional generation and classifier-free guidance (CFG). In this work, we identify a key limitation of CFG for counterfactual generation: it prescribes a global guidance scale for all attributes, leading to significant spurious changes in inferred counterfactuals. To mitigate this, we propose *Factored Classifier-Free Guidance* (FCFG), a flexible and model-agnostic guidance technique that enables attribute-wise control following a causal graph. FCFG complements recent advances in classifier-free guidance and can be seamlessly extended to advanced guidance schemes such as CFG++ and APG. Our experiments demonstrate that FCFG significantly improves the axiomatic soundness of inferred counterfactuals across both natural and medical image datasets, mitigating spurious amplification effects, and enhancing counterfactual reversibility.

## 1. Introduction

Counterfactual generation is considered to be fundamental to causal reasoning (Peters et al., 2017; Bareinboim et al., 2022), allowing us to explore hypothetical scenarios such as: *'How would this patient's disease have progressed if they had received treatment A instead of treatment B?'*. Answering such causal questions is important across various domains, such as healthcare (Castro et al., 2020), fairness (Kusner et al., 2017) and scientific discovery (Narayanaswamy et al., 2020). There has been a growing interest in generating counterfactual images using deep generative models, aiming to simulate how visual data would change under hypothetical interventions. Recent works build Structural Causal Models (SCMs) (Pearl,

2009) using deep generative model components such as normalizing flows (Rezende & Mohamed, 2015), Variational Autoencoders (VAEs) (Kingma & Welling, 2013; Child, 2020) and diffusion models (Sohl-Dickstein et al., 2015; Ho et al., 2020), enabling principled counterfactual inferences via *abduction-action-prediction* (Pawlowski et al., 2020; Sanchez & Tsaftaris, 2022; Ribeiro et al., 2023; Wu et al., 2025; Rasal et al., 2025; Ribeiro et al., 2025a).

Diffusion models have emerged as the state-of-the-art approach for image synthesis, achieving unprecedented fidelity and perceptual quality (Dhariwal & Nichol, 2021; Podell et al., 2023). Many previous works have explored diffusion models for counterfactual generation (Sanchez et al., 2022b;a; Pérez-García et al., 2024; Kumar et al., 2025; Rasal et al., 2025; Ribeiro et al., 2025b), leveraging Denoising Diffusion Implicit Models (DDIM) (Song et al., 2020) to deterministically encode images into a latent space, followed by conditional generation with modified attributes. Conditioning is typically enforced through guidance mechanisms, either with external classifier scores (Dhariwal & Nichol, 2021) or through Classifier-Free Guidance (CFG) (Ho & Salimans, 2022), which combines conditional and unconditional score estimates using a scalar guidance weight $\omega$. Combining DDIM inversion and guided conditional decoding has also become the dominant paradigm in diffusion-based image editing (Couairon et al., 2022; Wallace et al., 2023; Hertz et al., 2022; Epstein et al., 2023).

In counterfactual generation, CFG is often used to improve *intervention effectiveness*, by making the generated sample more strongly reflect the intended attribute change. While recent works have proposed refinements to classifier-free guidance to improve sampling fidelity (Chung et al., 2025; Sadat et al., 2025), both standard CFG and these extensions rely on a single global guidance signal applied to the entire conditioning embedding. In this work, we identify that in diffusion-based counterfactual generation, this uniform strengthening can inadvertently amplify attributes that should remain stable under a given causal intervention. We refer to this failure mode as *attribute amplification*, following the terminology introduced by Xia et al. (2024).

While Xia et al. (2024) observed attribute amplification in previous counterfactual models due to predictor-based fine-tuning (Ribeiro et al., 2023), we identify a distinct CFG-induced failure mode: even when a causal graph or se-

[1]Department of Computing, Imperial College London, UK. Correspondence to: Tian Xia <t.xia@imperial.ac.uk>.

*Proceedings of the 43rd International Conference on Machine Learning*, Seoul, South Korea. PMLR 306, 2026. Copyright 2026 by the author(s).

mantic grouping is known, a single global guidance weight can amplify intervention-relevant and intervention-invariant attributes together. This behavior is induced by the guidance mechanism itself, rather than merely by dataset artefacts or causal-graph misspecification. It can violate the intended causal graph by modifying attributes outside the causal pathway and can also cause the generation trajectory to drift from the original data manifold, degrading identity preservation (Mokady et al., 2023). Thus, addressing CFG-induced attribute amplification is critical for the reliable use of guided diffusion models in counterfactual inference.

To address the spurious effects of CFG under causal interventions, we propose *Factored Classifier-Free Guidance* (FCFG), a general inference-time guidance technique that significantly mitigates attribute amplification without requiring any changes to the underlying diffusion model. Unlike standard CFG, FCFG assigns separate weights to attribute groups, allowing for fine-grained, interpretable control over the generative process. In contrast to compositional diffusion approaches that rely on spatial masking (Shen et al., 2024) or multiple conditional models (Liu et al., 2022), FCFG operates at the semantic attribute level using a single diffusion model. The FCFG framework is general and supports arbitrary partitions of semantic attributes under mild independence assumptions. For counterfactual generation, we instantiate FCFG by grouping attributes according to their causal roles (e.g., *intervened* vs. *invariant*) and applying distinct guidance to each group. This assumes access to structured conditioning variables and a pre-specified causal or semantic grouping; FCFG does not address the upstream problem of causal discovery. Crucially, by factoring guidance and focusing it on the intended intervention, FCFG reduces the risk of the generation trajectory drifting away from the original data manifold (Yang et al., 2023; Mokady et al., 2023; Tang et al., 2024). Additionally, FCFG modifies only how the conditioning signal is factored and weighted, enabling its integration with standard CFG as well as recent guidance refinements such as CFG++ (Chung et al., 2025) and APG (Sadat et al., 2025). The contributions of this work are the following:

1. We identify and analyze the problem of *attribute amplification* in CFG, where a single global guidance scale indiscriminately strengthens all conditioning attributes, causing spurious changes in attributes that should remain invariant under a given intervention;

2. We propose *Factored Classifier-Free Guidance* (FCFG), a simple, flexible, and model-agnostic extension of CFG that assigns separate guidance weights to attribute groups and supports arbitrary groupings at inference time under mild independence assumptions;

3. Through extensive experiments on challenging real-world datasets, including medical imaging, we show

that FCFG reduces spurious effects, improves intervention effectiveness, and enhances counterfactual reversibility, and is compatible with recent guidance refinements such as CFG++ and APG.

## 2. Background

**Structural Causal Models.** SCMs (Pearl, 2009) consist of a triplet $\langle U, A, F \rangle$, where $U = \{u_i\}_{i=1}^K$ denotes the set of exogenous variables, $A = \{a_i\}_{i=1}^K$ the set of endogenous variables, and $F = \{f_i\}_{i=1}^K$ a collection of structural assignments such that each variable $a_k$ is determined by a function $f_k$ of its parents $\mathbf{pa}_k \subseteq A \backslash a_k$ and its corresponding noise $u_k$, such that $a_k := f_k(\mathbf{pa}_k, u_k)$. SCMs enable causal reasoning and interventions via the *do*-operator, e.g., setting a variable $a_k$ to a fixed value $c$ through $do(a_k := c)$. In this work, we focus on generating image counterfactuals and implement the underlying image synthesis mechanism using diffusion models. We assume access to structured semantic parent variables and a pre-specified causal graph or semantic grouping over them; learning this graph is an orthogonal causal discovery problem and is outside the scope of this work.

**Counterfactual inference.** A counterfactual represents a *'what-if'* scenario given observed events. We denote an image by $\mathbf{x}$, which is generated via a structural assignment $\mathbf{x} := f(\mathbf{u}, \mathbf{pa})$, given its causal parents $\mathbf{pa}$ and exogenous noise variable $\mathbf{u}$. Counterfactual inference (Pearl, 2009) proceeds in three steps: (i) *Abduction:* infer or approximate the latent noise $\mathbf{u}$ from the observed data and its parents, denoted $\mathbf{u} \approx f^{-1}(\mathbf{x}, \mathbf{pa})$; (ii) *Action:* apply an intervention to alter selected parent variables, yielding the counterfactual parents $\widetilde{\mathbf{pa}}$; (iii) *Prediction:* propagate the effect of the intervention through the model to compute a counterfactual as follows: $\tilde{\mathbf{x}} \approx f(f^{-1}(\mathbf{x}, \mathbf{pa}), \widetilde{\mathbf{pa}})$. Recent advancements have sought to implement these steps using deep generative model components, such as normalizing flows (Pawlowski et al., 2020), VAEs (Ribeiro et al., 2023; Monteiro et al., 2023), and diffusion models (Komanduri et al., 2024; Rasal et al., 2025; Pan & Bareinboim, 2025). The general idea is to model each structural assignment $f_\theta$ and its inverse $f_\phi^{-1}$ using deep generative models with trainable parameters $\{\theta, \phi\}$. For invertible models, $\theta$ and $\phi$ coincide.

### 2.1. Diffusion Models for Counterfactual Inference

Diffusion models (DMs) (Sohl-Dickstein et al., 2015; Ho et al., 2020) are latent variable models designed to generate data by gradually removing Gaussian noise from $\mathbf{x}_T \sim \mathcal{N}(\mathbf{0}, \mathbf{I})$ over $T$ steps. Given a clean data sample $\mathbf{x}_0 \sim p_{\text{data}}$, the forward noising process is defined as follows:

$$\mathbf{x}_t = \sqrt{\alpha_t}\, \mathbf{x}_0 + \sqrt{1 - \alpha_t}\, \boldsymbol{\epsilon}, \qquad \boldsymbol{\epsilon} \sim \mathcal{N}(\mathbf{0}, \mathbf{I}), \quad (1)$$

where $\{\alpha_t\}_{t=0}^{T}$ is a chosen noise schedule with $\alpha_t \in (0, 1]$, $\alpha_0 = 1$ and $\alpha_T \approx 0$. To learn the reverse process, a parameterized network $\boldsymbol{\epsilon}_\theta(\mathbf{x}_t, t, \mathbf{c})$ is trained to predict the added noise from noisy inputs. We adopt the conditional diffusion model formulation, where $\mathbf{c}$ denotes an embedding of semantic parent attributes $\mathbf{pa}$ used as conditioning. In our implementation, $\mathbf{c}$ is formed from attribute-specific MLP embedders that are trained jointly with the denoising network end-to-end, rather than as independently pretrained feature extractors. The training objective minimizes the noise prediction loss:

$$\min_{\theta} \mathbb{E}_{\mathbf{x}_0 \sim p_{\text{data}}, \boldsymbol{\epsilon} \sim \mathcal{N}(\mathbf{0}, \mathbf{I}), t \sim p_t} \left[ \| \boldsymbol{\epsilon} - \boldsymbol{\epsilon}_\theta(\mathbf{x}_t, t, \mathbf{c}) \|_2^2 \right]. \quad (2)$$

At inference time, data samples are generated by progressively denoising $\mathbf{x}_T$ from time $T$ to time 0. Following Song et al. (2020), the denoising step from $\mathbf{x}_t$ to $\mathbf{x}_{t-1}$ is given by:

$$\mathbf{x}_{t-1} = \sqrt{\alpha_{t-1}} \left( \frac{\mathbf{x}_t - \sqrt{1 - \alpha_t}\, \boldsymbol{\epsilon}_\theta(\mathbf{x}_t, t, \mathbf{c})}{\sqrt{\alpha_t}} \right)$$
$$+ \sqrt{1 - \alpha_{t-1} - \sigma_t^2}\, \boldsymbol{\epsilon}_\theta(\mathbf{x}_t, t, \mathbf{c}) + \sigma_t\, \boldsymbol{\epsilon}_t, \quad (3)$$

where $\boldsymbol{\epsilon}_t \sim \mathcal{N}(\mathbf{0}, \mathbf{I})$. Setting $\sigma_t = 0$ yields a deterministic sampling process known as DDIM (Song et al., 2020), which defines an invertible trajectory between data and latent space. Following Sanchez & Tsaftaris (2022); Sanchez et al. (2022a); Fontanella et al. (2024); Pérez-García et al. (2024); Rasal et al. (2025), we adopt this DDIM formulation for counterfactual generation, as summarized below; full algorithmic details are provided in App. C.

**Abduction.** We implement the abduction function $\mathbf{u} = f_\theta^{-1}(\mathbf{x}_0, \mathbf{pa})$ using the DDIM forward trajectory. Given an observed image $\mathbf{x}_0$ and conditioning $\mathbf{c}$ representing an embedding vector of semantic parents $\mathbf{pa}$, the latent $\mathbf{x}_T$ serves as a deterministic estimate of the exogenous noise $\mathbf{u}$:

$$\mathbf{x}_{t+1} = \sqrt{\alpha_{t+1}}\, \hat{\mathbf{x}}_0 + \sqrt{1 - \alpha_{t+1}}\, \boldsymbol{\epsilon}_\theta(\mathbf{x}_t, t, \mathbf{c}), \quad (4)$$
$$\hat{\mathbf{x}}_0 = \frac{1}{\sqrt{\alpha_t}} \left( \mathbf{x}_t - \sqrt{1 - \alpha_t}\, \boldsymbol{\epsilon}_\theta(\mathbf{x}_t, t, \mathbf{c}) \right),$$

for $t = 0, \ldots, T-1$, where $\hat{\mathbf{x}}_0$ denotes the model's estimate of the clean image at time $t$. The full abduction procedure is summarized in Algorithm 1.

**Action.** We apply an intervention to the semantic attributes $\mathbf{pa}$, e.g., `do(Male=1)`, and propagate the effect through the causal graph using invertible flows as in Pawlowski et al. (2020); Ribeiro et al. (2023). This yields the counterfactual attribute vector $\widetilde{\mathbf{pa}}$ and its embedding $\tilde{\mathbf{c}}$. The full action procedure is summarized in Algorithm 2.

**Prediction.** We implement the structural assignment $\tilde{\mathbf{x}} := f_\theta(\mathbf{u}, \widetilde{\mathbf{pa}})$ under the modified condition $\tilde{\mathbf{c}}$ using the DDIM reverse trajectory, with $\mathbf{u} = \mathbf{x}_T$ the exogenous noise estimated in Eq. (4):

$$\mathbf{x}_{t-1} = \sqrt{\alpha_{t-1}}\, \hat{\mathbf{x}}_0 + \sqrt{1 - \alpha_{t-1}}\, \boldsymbol{\epsilon}_\theta(\mathbf{x}_t, t, \tilde{\mathbf{c}}), \quad (5)$$

where $t = T, \ldots, 1$, $\hat{\mathbf{x}}_0$ is the predicted clean image, and the final output $\tilde{\mathbf{x}}_0$ is the predicted counterfactual $\tilde{\mathbf{x}}$. In practice, decoding under the counterfactual condition $\tilde{\mathbf{c}}$ using the conditional denoiser alone may be insufficient for producing effective counterfactuals. Additional guidance is often required to steer generation toward the desired intervention (Sanchez & Tsaftaris, 2022; Sanchez et al., 2022a; Komanduri et al., 2024; Fontanella et al., 2024; Weng et al., 2024; Song et al., 2024; Pérez-García et al., 2024; Rasal et al., 2025; Kumar et al., 2025). In line with previous work, we adopt classifier-free guidance to improve *intervention effectiveness*, by which we mean the extent to which the generated counterfactual reflects the intended target-attribute change, rather than a causal treatment effect.

## 2.2. Classifier-Free Guidance

Classifier-free guidance (CFG) (Ho & Salimans, 2022) is a widely adopted technique in conditional diffusion models. It enables conditional generation without requiring an external classifier by training a single denoising model to operate in both conditional and unconditional modes. During training, the same denoising network is exposed to both conditional and unconditional inputs by randomly replacing $\mathbf{c}$ with a null token $\varnothing$, thereby learning conditional and unconditional score estimates. At inference time, CFG biases the sampling process toward regions more consistent with the conditioning signal, which can be understood as sampling from a reweighted conditional distribution:

$$p^\omega(\mathbf{x}_t | \mathbf{c}) \propto p(\mathbf{x}_t) p(\mathbf{c} | \mathbf{x}_t)^\omega, \quad (6)$$

where $\omega \geq 0$ controls the guidance strength. This corresponds to interpolating between the unconditional and conditional scores:

$$\nabla_{\mathbf{x}_t} \log p^\omega(\mathbf{x}_t | \mathbf{c}) = \omega \nabla_{\mathbf{x}_t} \log p(\mathbf{x}_t | \mathbf{c}) + (1 - \omega) \nabla_{\mathbf{x}_t} \log p(\mathbf{x}_t). \quad (7)$$

In practice, this is implemented by combining the model's predictions with and without conditioning:

$$\boldsymbol{\epsilon}_{\text{CFG}}(\mathbf{x}_t, t, \mathbf{c}) = (1 - \omega) \cdot \boldsymbol{\epsilon}_\theta(\mathbf{x}_t, t, \varnothing) + \omega \cdot \boldsymbol{\epsilon}_\theta(\mathbf{x}_t, t, \mathbf{c}). \quad (8)$$

With CFG, the *abduction* and *action* steps remain unchanged. The only difference arises in the *prediction* step, where the conditional denoiser is replaced by the guided prediction $\boldsymbol{\epsilon}_{\text{CFG}}(\mathbf{x}_t, t, \tilde{\mathbf{c}})$ to improve intervention effectiveness (Rasal et al., 2025); see Algorithm 3 for details.

$$\mathbf{x}_{t-1} = \frac{\sqrt{\alpha_{t-1}}}{\sqrt{\alpha_t}} \left( \mathbf{x}_t - \sqrt{1 - \alpha_t}\, \boldsymbol{\epsilon}_{\text{CFG}}(\mathbf{x}_t, t, \tilde{\mathbf{c}}) \right)$$
$$+ \sqrt{1 - \alpha_{t-1}}\, \boldsymbol{\epsilon}_{\text{CFG}}(\mathbf{x}_t, t, \tilde{\mathbf{c}}). \quad (9)$$

Despite its effectiveness, CFG applies a single global guidance weight $\omega$ uniformly across the entire counterfactual embedding $\tilde{\mathbf{c}}$, which typically encodes multiple semantic attributes. In counterfactual generation, however, only the attributes affected by the intervention should be strengthened, while attributes outside the causal effect of the intervention should remain stable. Applying the same guidance strength to all components of $\tilde{\mathbf{c}}$ therefore couples intervention-relevant and intervention-invariant attributes during sampling. This can induce unintended changes to invariant attributes, a failure mode known as *attribute amplification* (Xia et al., 2024). Such off-target changes violate the intended relationships in the associated causal graph and undermine the axiomatic soundness of inferred counterfactuals (Monteiro et al., 2023).

To address this limitation, we propose a structured alternative that factors the guidance signal across semantically or causally defined attribute groups, allowing each group to be controlled with its own guidance weight.

# 3. Factored Classifier-Free Guidance

In this section, we present our *Factored Classifier-Free Guidance* (FCFG) for counterfactual image generation. We first propose a simple *attribute-split conditioning embedder* (Sec. 3.1) as a practical implementation that separates attributes in the embedding space to enable selective control. Building on this, we then describe our FCFG formulation in detail (Sec. 3.2), which allows distinct guidance strengths to be applied to different subsets of attributes within an assumed causal graph. Finally, we present how FCFG is integrated into DDIM-based counterfactual inference (Sec. 3.3), detailing its application across abduction, action, and prediction steps using causally defined attribute groupings.

## 3.1. Attribute-Split Conditioning Embedding

In practice, raw conditioning inputs such as discrete image labels or structured attributes (e.g., a patient's *sex*, *race*, or *disease status*) are not used directly in diffusion models but transformed into dense vectors using embedding functions, typically via multi-layer perceptrons (MLPs) (Dhariwal & Nichol, 2021), convolutional encoders (Zhang et al., 2023), or transformer-based text encoders (Ho & Salimans, 2022; Ramesh et al., 2022). These embeddings align semantic or categorical inputs with the model's internal representations, but conventional designs often entangle multiple attributes into a single conditioning vector, making it difficult to independently control attributes during sampling.

To address this, we introduce a simple *attribute-split conditioning embedding* technique that preserves the identity of each attribute in the embedding space. Let $pa_i$ denote the raw value of the $i$-th parent attribute (e.g., a binary indicator

or scalar). Each $pa_i$ is embedded independently via a dedicated MLP: $\mathcal{E}_i : \mathbb{R}^{d_i} \to \mathbb{R}^d$, and the final condition vector is formed by concatenating the outputs:

$$\mathbf{c} = \text{concat}\big(\mathcal{E}_1(pa_1), \mathcal{E}_2(pa_2), \ldots, \mathcal{E}_K(pa_K)\big), \quad (10)$$

where $\mathbf{c} \in \mathbb{R}^{Kd}$. This architecture provides a flexible representation where each attribute occupies a separate block of the conditioning vector. The embedders $\{\mathcal{E}_i\}_{i=1}^K$ are trained jointly with the denoising network end-to-end; they are not independently pretrained feature extractors. As a result, we can selectively null-tokenize or modulate individual attributes at inference time, enabling fine-grained control. Throughout the rest of the paper, we denote the semantic attribute vector as $\mathbf{pa}$ and the corresponding embedding vector as $\mathbf{c}$, as defined in Eq. (10).

## 3.2. Group-wise Formulation of FCFG

To overcome the limitations of CFG and enable more precise, causally aligned control in counterfactual image generation, we propose *Factored Classifier-Free Guidance* (FCFG). Let $\mathbf{pa} = (pa_1, \ldots, pa_K)$ denote the vector of semantic parent attributes. Rather than applying a single guidance weight uniformly to the entire conditioning vector, we partition semantic attributes $\mathbf{pa}$ into $M$ disjoint groups $\mathbf{pa}^{(1)}, \mathbf{pa}^{(2)}, \ldots, \mathbf{pa}^{(M)}$, and apply a separate guidance weight $\omega_m$ to each group.

**Proposition 3.1** (Proxy Posterior for FCFG). *Under the assumption that the groups $\mathbf{pa}^{(1)}, \ldots, \mathbf{pa}^{(M)}$ are conditionally independent given the latent variable $\mathbf{x}_t$, for any time $t$, that is: $p(\mathbf{pa}|\mathbf{x}_t) = \prod_{m=1}^M p(\mathbf{pa}^{(m)}|\mathbf{x}_t)$, we obtain the following factorized proxy posterior:*

$$p^\omega(\mathbf{x}_t \mid \mathbf{pa}) \propto p(\mathbf{x}_t) \prod_{m=1}^M p(\mathbf{pa}^{(m)} \mid \mathbf{x}_t)^{\omega_m}, \quad (11)$$

*where $\omega_m \geq 0$ controls the guidance strength for each group.*

A complete derivation and score-based justification for this proxy posterior is provided in App. B. The corresponding guided update used in score-based diffusion sampling is then given by:

$$
\begin{aligned}
\nabla_{\mathbf{x}_t} \log p^\omega(\mathbf{x}_t|\mathbf{pa}) = &\nabla \log p(\mathbf{x}_t) \\
&+ \sum_{m=1}^M \omega_m \big(\nabla \log p(\mathbf{x}_t|\mathbf{pa}^{(m)}) - \nabla \log p(\mathbf{x}_t)\big).
\end{aligned} \quad (12)
$$

In practice, we encode $\mathbf{pa}$ into a dense conditioning vector $\mathbf{c}$ using the attribute-split embedding described in Sec. 3.1. For each group $m$, we construct a masked embedding $\underline{\mathbf{c}}^{(m)}$ that retains only the embeddings for $\mathbf{pa}^{(m)}$ and replaces all

others with null tokens (represented here as zero vectors):

$$\underline{\mathbf{c}}^{(m)} = \text{concat}\left(\delta_1^{(m)} \cdot \mathcal{E}_1(pa_1), \ldots, \delta_K^{(m)} \cdot \mathcal{E}_K(pa_K)\right),$$

$$\delta_i^{(m)} = \begin{cases} 1, & \text{if } pa_i \in \mathbf{pa}^{(m)} \\ 0, & \text{otherwise} \end{cases}.$$

(13)

The final guided score used in the diffusion model is computed as follows:

$$\boldsymbol{\epsilon}_{\text{FCFG}}(\mathbf{x}_t, t, \mathbf{c}) = \boldsymbol{\epsilon}_\theta(\mathbf{x}_t, t, \varnothing)$$
$$+ \sum_{m=1}^{M} \omega_m \left( \boldsymbol{\epsilon}_\theta(\mathbf{x}_t, t, \underline{\mathbf{c}}^{(m)}) - \boldsymbol{\epsilon}_\theta(\mathbf{x}_t, t, \varnothing) \right).$$

(14)

We emphasize that the model is trained with standard classifier-free dropout, where the entire conditioning vector is replaced by the null token, rather than with all possible partially masked conditioning patterns. Thus, FCFG introduces a mild train–test mismatch at inference time by evaluating the denoiser on group-masked embeddings. Empirically, we do not observe systematic artifacts or instability from this mismatch, but it may become more pronounced with many groups, stronger guidance weights, or more severe masking patterns.

The proposed FCFG framework is highly flexible, as it allows arbitrary groupings of attributes, regardless of whether attributes within a group are mutually independent or not. Consequently, strongly entangled attributes should be placed in the same group rather than separated into different guidance groups. For the factorized proxy posterior in Theorem 3.1, the required assumption is conditional independence across groups given the latent variable $\mathbf{x}_t$. This flexibility enables a wide range of configurations. For instance, setting $M=1$ recovers standard global CFG, while increasing $M$ provides progressively finer-grained control, including per-attribute guidance ($M=K$) as an extreme case where we assume all attributes are independent of each other. Algorithm 4 summarizes the general group-wise FCFG counterfactual prediction procedure.

### 3.3. FCFG for Counterfactual Generation

We now detail how FCFG is straightforwardly integrated into DDIM-based counterfactual inference.

**Abduction.** The abduction step proceeds as in Eq. (4), where the conditioning vector $\mathbf{c}$ is obtained by embedding the semantic parent attributes $\mathbf{pa}$ using the attribute-split embedder defined in Eq. (10).

**Action.** As in previous setups, we apply a causal intervention to obtain a modified semantic vector $\widetilde{\mathbf{pa}}$. This is then embedded into the counterfactual conditioning vector $\tilde{\mathbf{c}}$ via

the attribute-split embedder:

$$\tilde{\mathbf{c}} = \text{concat}\left(\mathcal{E}_1(\widetilde{pa}_1), \ldots, \mathcal{E}_K(\widetilde{pa}_K)\right).$$

(15)

**Prediction.** The prediction step uses the FCFG-guided reverse DDIM trajectory:

$$\mathbf{x}_{t-1} = \sqrt{\alpha_{t-1}}\hat{\mathbf{x}}_0 + \sqrt{1-\alpha_{t-1}}\boldsymbol{\epsilon}_{\text{FCFG}}(\mathbf{x}_t, t, \tilde{\mathbf{c}}),$$
$$\text{where} \quad \hat{\mathbf{x}}_0 = \frac{1}{\sqrt{\alpha_t}}\left(\mathbf{x}_t - \sqrt{1-\alpha_t}\boldsymbol{\epsilon}_{\text{FCFG}}(\mathbf{x}_t, t, \tilde{\mathbf{c}})\right),$$

(16)

and $\boldsymbol{\epsilon}_{\text{FCFG}}(\mathbf{x}_t, t, \tilde{\mathbf{c}})$ is computed as in Eq. (14) using counterfactual conditioning embedding $\tilde{\mathbf{c}}$.

For counterfactual generation, we adopt a two-group partitioning of attributes based on the assumed causal graph. The *affected group* $\mathbf{pa}^{\text{aff}}$ contains attributes directly intervened upon and their descendants, while the *invariant group* $\mathbf{pa}^{\text{inv}}$ comprises attributes expected to remain unchanged. For the proxy-posterior interpretation in Theorem 3.1, these groups are assumed conditionally independent given the latent $\mathbf{x}_t$. In practice, this grouping is informed by the assumed causal graph: attributes directly intervened upon and their descendants are assigned to the affected group, while attributes outside the intervention's causal effect are assigned to the invariant group. Under this setup, Eq. (14) uses $M=2$ groups with separate guidance weights $\omega_{\text{aff}}$ and $\omega_{\text{inv}}$. The two-group counterfactual prediction procedure is summarized in Algorithm 5.

Note that the two-group partition is only one possible choice: guidance can also be separated at a finer-grained level, including the attribute level, provided conditional independence holds across groups. We present such extensions, including multi-attribute interventions and attribute-wise configurations, in App. I, which further demonstrate the generality and flexibility of FCFG. The proposed factoring framework can be integrated with recent CFG variants such as CFG++ and APG by applying the group-wise decomposition to their guided score formulations (see Apps. J and K).

## 4. Experiments

In this section, we demonstrate the benefits of the proposed approach across three public datasets. For each dataset, we train a diffusion model with the same architecture and training protocol, detailed in App. D. We first compare FCFG against the standard CFG baseline, and then further contextualize it against existing counterfactual generation methods, including HVAE, HVAE-soft, and SA-DCG. Unless otherwise specified, settings labeled as $\omega=X$ correspond to standard CFG with a global guidance weight. In contrast, configurations denoted by $\omega_{\text{aff}}=X$, $\omega_{\text{inv}}=Y$ represent the two-group FCFG, where separate guidance weights are

*Figure 1.* Comparison of $\Delta$ metrics under different interventions in CelebA-HQ. Left: Intervention on `Smiling`. Right: Intervention on `Young`. Both use baseline $\omega=1.0$. Under global CFG, increasing $\omega$ boosts the intended attribute but amplifies non-target ones. FCFG achieves similar improvements on the target attribute while mitigating amplification. See App. F.2 for full results.

*Figure 2.* Counterfactual generations in CelebA-HQ ($64 \times 64$). Each row compares global CFG (left) and FCFG (right) across guidance weights. Top: under `do(Male=no)`, global CFG also amplifies `Smiling`, although `Smiling` is not intervened upon. Middle: under `do(Young=no)`, global CFG produces increasingly strong age-related changes while also altering facial expression and identity beyond the intended age intervention. Bottom: under `do(Smiling=yes)`, global CFG makes the subject appear older, adds glasses, and alters identity. FCFG mitigates these unintended changes and better preserves invariant attributes. See App. F.3 for more visual results.

applied to the intervened and invariant attribute groups, respectively.

Following Monteiro et al. (2023); Melistas et al. (2024), we primarily evaluate counterfactual quality using two metrics. **Effectiveness** ($\Delta$): Measured by a pretrained classifier as the change in AUROC for target or invariant attributes relative to $\omega = 1.0$ (no CFG). Higher $\Delta$ on the intervened attribute indicates stronger target-attribute change, while large $\Delta$ on invariant attributes indicates unintended amplification. **Reversibility**: Assesses how well counterfactuals can be reversed to the original image using inverse interven-

tions. We report MAE and LPIPS; lower values indicate better identity preservation. See App. A.2 for details.

### 4.1. Main Results

**CelebA-HQ.** We begin our empirical evaluation of FCFG on the CelebA-HQ dataset (Karras et al., 2017), using `Smiling`, `Male`, and `Young` as independent binary attributes. We adopt this simplified setup to isolate inference-time failures of standard CFG in a controlled setting. Under this designed scenario, unintended changes in non-intervened attributes can be attributed to attribute ampli-

*Figure 3.* Reversibility analysis in CelebA-HQ ($64 \times 64$). Left: Quantitative evaluation of how well the original image is recovered after generating a counterfactual and mapping it back to the original condition under `do(Smiling)`. Right: A qualitative example showing a counterfactual generated under `do(Male)` and its reconstruction after reversing the intervention with CFG and our FCFG.

fication rather than valid causal effects. Refer to App. F.1 for more dataset details.

Fig. 1 presents the $\Delta$ metrics under different guidance strategies for two separate interventions, namely `do(Smiling)` and `do(Young)`. As the global guidance weight $\omega$ of CFG increases (left to right side of each plot), the $\Delta$ of the intervened attribute improves, but so do the $\Delta$ values of attributes that should remain invariant, indicating undesirable spurious amplification. In contrast, the right side of each plot shows results for FCFG, where distinct weights are applied to affected ($\omega_{\text{aff}}$) and invariant ($\omega_{\text{inv}}$) attribute groups. This decoupled formulation achieves comparable or stronger improvement on the intervened attribute while keeping the others stable, validating the ability of FCFG to produce more disentangled and effective counterfactuals purely at inference time.

Fig. 2 illustrates how global CFG can introduce unintended changes by uniformly amplifying all conditioning signals, even when only one attribute is meant to change. In the top row, applying `do(Male=no)` with increasing $\omega$ inadvertently amplifies `Smiling`; in the middle row, `do(Young=no)` changes non-target facial attributes and expression in addition to age; and in the bottom row, `do(Smiling=yes)` introduces changes to age, identity, and even adds glasses. These unintended shifts stem from global CFG treating all attributes equally. In contrast, FCFG applies decoupled guidance across attribute groups, assigning stronger weights to attributes affected by the intervention while preserving invariant factors. This results in counterfactuals that more faithfully reflect the intended change while better preserving identity and consistency in invariant attributes.

Fig. 3 evaluates the reversibility of counterfactuals in CelebA-HQ. The left panel shows MAE and LPIPS when recovering the original after applying and then reversing an intervention (e.g., `do(Smiling)`). With global CFG, errors grow as guidance strength increases, while FCFG yields consistently lower values for the same settings, improving recovery. The right panel shows a qualitative example under `do(Male)`, where global CFG amplifies non-intervened attributes (e.g., `Young`), making the reversed image appear

older. In contrast, FCFG applies strong guidance only to intervened attributes, mitigating amplification and producing more faithful, disentangled, and reversible counterfactuals. More reversibility results are provided in Apps. F.2 and F.3.

**EMEBD.** We evaluate FCFG on the EMBED (Jeong et al., 2023) breast mammography dataset. For our experiments, we define a binary `circle` attribute based on the presence of circular skin markers, and a binary breast `density` label, where categories A and B are grouped as `low` and categories C and D as `high`. See App. G.1 for dataset details.

Fig. 4 presents results for counterfactual generation on EMBED. The left bar plot reports $\Delta$ effectiveness metrics, measuring how classifier performance changes relative to the baseline. While global CFG improves effectiveness for the target attribute (`circle`), it also increases effectiveness on non-intervened attributes such as `density`, indicating unintended attribute amplfication. FCFG mitigates this by applying selective guidance, maintaining stable performance on non-target attributes. The figure on the right illustrates a key example: applying `do(density)` under global CFG unintentionally amplifies the presence of circular skin markers, as evidenced by the increased number of visible circles in both the counterfactual and reversed images. This is suppressed under FCFG, where `circle` features remain unchanged in both counterfactual and reversed images.

**MIMIC.** We evaluate our method on the MIMIC-CXR dataset (Johnson et al., 2019). We follow the dataset splits and filtering protocols from Ribeiro et al. (2023), and focus on the binary disease label of pleural effusion. The underlying causal graph in Ribeiro et al. (2023) includes four attributes: `race`, `sex`, `finding`, and `age`. We adopt this setup, but since our goal is to study *attribute amplification* caused by CFG, we focus on `sex`, `race`, and `finding`, which we assume to be mutually independent for the purposes of our analysis. Refer to App. H.1 for dataset details.

Fig. 5 presents an evaluation of counterfactual generation in MIMIC-CXR, highlighting the advantages of our proposed FCFG. The bar plot on the left shows $\Delta$ metrics that quantify the change in effectiveness relative to the baseline $\omega = 1.0$. While global CFG improves effectiveness for the

*Figure 4.* Evaluation of counterfactual generation on EMBED (192×192). Left: Δ metrics showing the effect of `do(circle)`. FCFG improves target intervention effectiveness while suppressing spurious shifts in non-intervened attributes. Right: A visual example showing the input image, the counterfactual under `do(density)`, the reversed image, and their difference maps (CF/Rev.−input). See App. G.2 for full quantitative results and App. G.3 for more visual results.

*Figure 5.* Evaluation of counterfactual generation on MIMIC (192×192). Left: Δ metrics showing the effect of `do(finding)`. FCFG improves target intervention effectiveness while suppressing spurious shifts in non-intervened attributes. Right: A visual example showing the input image, the counterfactual under `do(finding)`, the reversed image, and their difference maps (CF/Rev.−input). See App. H.2 for full quantitative results and App. H.3 for more qualitative results.

intervened variable (`finding`) as expected, it also introduces substantial spurious shifts in non-intervened attributes such as `race` and `sex`, revealing unwanted attribute amplification. In contrast, FCFG achieves comparable or higher intervened effectiveness while suppressing spurious amplification, demonstrating more precise and controlled generation. On the right, we show a qualitative example of a counterfactual generated under `do(finding)`, its reversed reconstruction, and their corresponding difference maps. Compared to global CFG, our method yields localized, clinically meaningful changes in counterfactuals and better identity preservation in the reversed image.

### 4.2. Comparison with Existing Counterfactual Methods

To further contextualize FCFG against existing counterfactual generation methods, we compare with HVAE (Ribeiro et al., 2023), HVAE-soft (Xia et al., 2024), and SA-DCG (Rasal et al., 2025); detailed numerical results are provided in App. E.1. These methods also rely on structured causal information, but incorporate it differently: HVAE and HVAE-soft use predictor-based counterfactual training objectives, while SA-DCG uses a diffusion-autoencoder framework with semantic abduction. In contrast, FCFG

modifies only the inference-time guidance rule of a trained conditional diffusion model.

Across CelebA-HQ and MIMIC-CXR, FCFG achieves a stronger target/off-target trade-off: it maintains or improves intervention effectiveness on the target attribute while substantially reducing off-target amplification. On CelebA-HQ under `do(Smiling)`, SA-DCG improves reversibility relative to standard CFG, but still exhibits off-target amplification under global guidance. At the stronger guidance setting, SA-DCG reaches a target AUC change of $+12.9$ but also increases the off-target Male AUC by $+3.0$, whereas FCFG reaches a slightly higher target change of $+13.1$ while reducing the off-target change to $-1.5$, with better reversibility measured by MAE and LPIPS.

On MIMIC-CXR, FCFG also achieves comparable or stronger target intervention than HVAE, HVAE-soft, and CFG, while reducing off-target change by a large margin. For example, under `do(sex)`, FCFG achieves a target AUC change of $+7.6$ with only $+0.6$ off-target change, compared with $+9.0$, $+5.7$, and $+10.7$ off-target change for HVAE, HVAE-soft, and CFG, respectively. Under `do(finding)`, FCFG achieves the strongest target

*Figure 6.* Qualitative results for `do(Smiling, Male, Young)`. Since all three attributes are intervened, no invariant attributes remain; therefore, global CFG and the two-group variant of FCFG are equivalent, as a single guidance weight is applied to all attributes. Attribute-wise FCFG further provides flexibility by enabling independent control over each attribute; setting $\omega_s = \omega_m = \omega_y = 2.5$ recovers the global/two-group configuration. See App. I.2 for more quantitative and qualitative results.

change of $+18.8$ while reducing off-target change to $+0.6$. These results support our central claim that the global guidance mechanism itself is an important source of attribute amplification, and that factorizing guidance provides a simple inference-time remedy.

We further evaluate general image quality using FID and analyze the practical cost of FCFG. On CelebA-HQ, FCFG consistently improves FID over global CFG across `do(Smiling)`, `do(Male)`, and `do(Young)`, suggesting that reducing off-target amplification also helps preserve image realism. FCFG also offers a favourable simplicity–control trade-off: unlike HVAE and HVAE-soft, it does not require an additional predictor-based counterfactual training objective, and unlike SA-DCG, it avoids a heavier diffusion-autoencoder framework and identity-preserving inference-time optimization. Full baseline comparisons, FID results, dropout ablations, and practical-cost comparisons are provided in App. E.

### 4.3. Generality and Compatibility of FCFG

**Multi-attribute intervention.** To illustrate that FCFG extends beyond the two-group setting used in our main experiments, we consider a three-attribute intervention `do(Smiling, Male, Young)`. In this setting, all attributes are directly intervened upon, and no invariant attributes remain; as a result, the standard two-group formulation collapses to global guidance, since a single guidance weight is applied to all attributes. As shown in Fig. 6, attribute-wise FCFG instead assigns independent guidance weights ($\omega_s, \omega_m, \omega_y$) to each attribute. While symmetric weights recover the global configuration, asymmetric settings demonstrate the additional flexibility of FCFG in selectively controlling individual semantic factors. Further discussion and results on multi-attribute interventions are provided in App. I.

**Compatibility with advanced guidance schemes.** FCFG is compatible with advanced guidance variants such as CFG++ (Chung et al., 2025) and APG (Sadat et al., 2025).

While both methods still suffer from attribute amplification in counterfactual generation, integrating FCFG mitigates this issue. Algorithmic descriptions and quantitative and qualitative results are provided in Apps. J and K.

## 5. Conclusion

We identify a key limitation of classifier-free guidance for counterfactual image generation: using a single global guidance scale for all conditioning attributes can spuriously amplify factors that should remain invariant under a causal intervention. To address this, we propose *Factored Classifier-Free Guidance* (FCFG), a factored guidance framework that applies separate guidance weights to semantically or causally defined groups of attributes. FCFG supports flexible groupings under a group-wise conditional independence assumption. In this work, we primarily consider a two-group partition that separates affected attributes, i.e., intervened attributes and their descendants, from invariant attributes. We also demonstrate that the framework naturally extends to more fine-grained settings, including multi-attribute and per-attribute configurations. Moreover, FCFG can be integrated with recent CFG refinements such as CFG++ and APG, addressing attribute amplification in these methods as well. We evaluate FCFG on CelebA-HQ, EMBED, and MIMIC-CXR, covering natural and medical image domains. Our results show that FCFG consistently reduces off-target attribute amplification while maintaining intervention effectiveness and improving identity preservation, particularly at higher guidance strengths. The main limitations are that FCFG assumes access to structured conditioning variables and a pre-specified causal or semantic grouping, and that guidance weights are currently chosen empirically. Important directions for future work include adaptively selecting guidance weights based on the input condition or diffusion timestep, and extending factored guidance to settings where the attribute grouping is uncertain or learned. Beyond counterfactual generation, the same principle may also extend to other conditional generation settings and to other diffusion or flow-matching models.

## Acknowledgements

T.X. is supported through the Imperial College London UKRI Impact Acceleration Account EP/X52556X/1. B.G. received support from the Royal Academy of Engineering as part of his Kheiron/RAEng Research Chair. B.G. and F.R. acknowledge the support of the UKRI AI programme, and the EPSRC, for CHAI - EPSRC Causality in Healthcare AI Hub (grant no. EP/Y028856/1). R.M. is supported by the European Union's Horizon Europe research and innovation programme under grant agreement 101080302. R.R. is supported by EPSRC through a Doctoral Training Partnerships PhD Scholarship. A.K. is supported by the EPSRC Doctoral Prize.

## Impact Statement

This paper aims to advance machine learning methods for counterfactual image generation by improving the controllability of classifier-free guidance. More reliable counterfactual generation could support scientific analysis, model auditing, and sensitivity studies in domains where understanding the effect of changing specific attributes is important. In medical imaging, such tools may help researchers study how generative models respond to changes in clinically relevant labels while preserving patient-specific factors that should remain unchanged.

At the same time, counterfactual images should not be interpreted as factual clinical outcomes or used directly for diagnosis, treatment planning, or decision-making without appropriate validation and expert oversight. As with other generative methods, there is also a risk that synthetic counterfactuals could be misused to create misleading visual evidence or to overstate causal conclusions when the assumed causal graph or conditioning variables are misspecified. Our method assumes access to structured attributes and a pre-specified causal or semantic grouping, and therefore does not remove the need for domain expertise when defining interventions and interpreting generated samples.

We believe these risks are best mitigated through careful validation, transparent reporting of assumptions, and restricting high-stakes use to research or decision-support settings with human oversight. Overall, the work is intended as a methodological contribution toward more controlled and interpretable generative modeling.

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

# A. Background

## A.1. Notation Summary

| Symbol | Description |
|--------|-------------|
| $\mathbf{x}_0$ (also denoted as $\mathbf{x}$) | Original observed image |
| $\mathbf{x}_t$ | Noisy image at diffusion timestep $t$ |
| $\mathbf{x}_T$ (i.e., $\mathbf{u}$) | Latent code after DDIM forward process (abduction; exogenous noise) |
| $\tilde{\mathbf{x}}$ | Generated counterfactual image |
| $\boldsymbol{\epsilon}_\theta(\mathbf{x}_t, t, \cdot)$ | Denoiser prediction given condition input |
| $\mathbf{pa}$ | Vector of semantic parent attributes (e.g., *sex*, *age*) |
| $\widetilde{\mathbf{pa}}$ | Counterfactual parent attributes after intervention |
| $pa_i$ | Raw value of the $i$-th semantic attribute |
| $\mathcal{E}_i$ | Embedding MLP for $pa_i$: $\mathbb{R}^{d_i} \to \mathbb{R}^d$ |
| $\mathbf{c}$ | Full conditioning vector from $\mathbf{pa}$ |
| $\tilde{\mathbf{c}}$ | Conditioning vector from counterfactual attributes $\widetilde{\mathbf{pa}}$ |
| $\varnothing$ | Null token for classifier-free guidance (unconditional input) |
| $\omega_m$ | CFG weight for attribute group $m$ |
| $\mathbf{pa}^{(m)}$ | Attributes in the $m$-th group |
| $\underline{\mathbf{c}}^{(m)}$ | Masked condition vector preserving only group $m$ |

*Table A.1.* Notation used throughout the paper. Tilde ($\sim$) indicates counterfactual quantities.

## A.2. Evaluating Counterfactuals

To evaluate the soundness of generated counterfactuals, we define a counterfactual image generation function $\mathcal{F}_\theta(\cdot)$, which produces counterfactuals according to

$$\tilde{\mathbf{x}} := \mathcal{F}_\theta(\mathbf{x}, \mathbf{pa}, \tilde{\mathbf{pa}}) = f_\theta(f_\theta^{-1}(\mathbf{x}, \mathbf{pa}), \tilde{\mathbf{pa}}). \tag{17}$$

We describe three key metrics used to assess counterfactual quality: composition, reversibility, and effectiveness (Monteiro et al., 2023; Melistas et al., 2024).

**Composition** evaluates how well the model reconstructs the original image under a null intervention, by computing a distance metric $d$ between the original image and its counterfactual:

$$\mathrm{Comp}(\mathbf{x}, \mathbf{pa}) := d(\mathbf{x}, \mathcal{F}_\theta(\mathbf{x}, \mathbf{pa}, \mathbf{pa})). \tag{18}$$

**Reversibility** measures the consistency of the counterfactual transformation by applying the reverse intervention and comparing the result to the original image:

$$\mathrm{Rev}(\mathbf{x}, \mathbf{pa}, \tilde{\mathbf{pa}}) := L_1(\mathbf{x}, \mathcal{F}_\theta(\mathcal{F}_\theta(\mathbf{x}, \mathbf{pa}, \tilde{\mathbf{pa}}), \tilde{\mathbf{pa}}, \mathbf{pa})). \tag{19}$$

**Effectiveness** quantifies whether the intended intervention has the desired causal effect. It compares the intervened value $\tilde{\mathbf{pa}}_k$ with the prediction obtained by an anti-causal model applied to the counterfactual image:

$$\mathrm{Eff}(\mathbf{x}, \mathbf{pa}, \tilde{\mathbf{pa}}) := L_k(\tilde{\mathbf{pa}}_k, \mathbf{Pa}_k(\mathcal{F}_\theta(\mathbf{x}, \mathbf{pa}, \tilde{\mathbf{pa}}))). \tag{20}$$

**A note on Composition.** We do not report the **Composition** metric in our evaluation, as it is ill-defined in the context of CFG and FCFG. Since both methods use the same trained diffusion model, applying a null intervention (i.e., $\tilde{\mathbf{pa}} = \mathbf{pa}$) does not meaningfully differentiate between them. If reconstruction is performed without guidance, CFG and FCFG are equivalent, reducing to standard decoding. If guidance is applied, it becomes unclear how to split attributes into invariant ($\mathbf{pa}_{\mathrm{inv}}$) and intervened ($\mathbf{pa}_{\mathrm{aff}}$) groups during null intervention. For instance, assigning all attributes to $\mathbf{pa}_{\mathrm{inv}}$ with $\omega_{\mathrm{inv}} = 1$ would effectively disable guidance, making the comparison trivial and uninformative. For this reason, we focus on **Effectiveness** and **Reversibility**, which better capture the behavior of guided sampling under interventions.

**Effectiveness Classifier.** To evaluate effectiveness, we train a classifier with a ResNet-18 backbone for each dataset, using the split as in Table A.2. The classifier predicts the intervened attribute from generated counterfactuals, and AUROC is used to quantify intervention success. On CelebA, the classifier achieves AUROC scores of 0.974 (`Smiling`), 0.992 (`Male`), and 0.828 (`Young`). On EMBED, the AUROC is 0.935 for `density` and 0.908 for `circle`. On MIMIC-CXR, AUROC scores are 0.864 for `race`, 0.991 for `sex`, and 0.938 for `finding`.

**Reversibility Metrics.** We use Mean Absolute Error (MAE) and LPIPS (Zhang et al., 2018) to evaluate reversibility. These metrics quantify the pixel-level and perceptual similarity, respectively, between the original image and its reversed counterfactual.

# B. Factored Classifier-Free Guidance

We provide a theoretical justification for the Factored Classifier-Free Guidance formulation presented in Proposition 3.1, interpreting it as gradient ascent on a sharpened proxy posterior under a group-level conditional independence assumption.

**Proof of Proposition 3.1.**    We begin by assuming that the semantic attributes are partitioned into $M$ disjoint groups:

$$\mathbf{pa} = (\mathbf{pa}^{(1)}, \ldots, \mathbf{pa}^{(M)}). \tag{21}$$

Under the assumption that these groups are conditionally independent given $\mathbf{x}_t$, we have:

$$p(\mathbf{pa} \mid \mathbf{x}_t) = \prod_{m=1}^{M} p(\mathbf{pa}^{(m)} \mid \mathbf{x}_t). \tag{22}$$

Applying Bayes' rule:

$$p(\mathbf{x}_t \mid \mathbf{pa}) = \frac{p(\mathbf{pa} \mid \mathbf{x}_t) \cdot p(\mathbf{x}_t)}{p(\mathbf{pa})} \propto p(\mathbf{pa} \mid \mathbf{x}_t) \cdot p(\mathbf{x}_t). \tag{23}$$

so the posterior can be factorized as:

$$p(\mathbf{x}_t \mid \mathbf{pa}) \propto p(\mathbf{x}_t) \cdot \prod_{m=1}^{M} p(\mathbf{pa}^{(m)} \mid \mathbf{x}_t). \tag{24}$$

Applying group-level guidance weights $\omega_m$ yields the sharpened proxy posterior:

$$p^{\omega}(\mathbf{x}_t \mid \mathbf{pa}) \propto p(\mathbf{x}_t) \cdot \prod_{m=1}^{M} p(\mathbf{pa}^{(m)} \mid \mathbf{x}_t)^{\omega_m}. \tag{25}$$

**Gradient of the log proxy posterior.**    Starting from the group-weighted proxy posterior in Eq. (25), taking logs yields

$$\log p^{\omega}(\mathbf{x}_t \mid \mathbf{pa}) = \log p(\mathbf{x}_t) + \sum_{m=1}^{M} \omega_m \log p(\mathbf{pa}^{(m)} \mid \mathbf{x}_t) + \text{const}. \tag{26}$$

Differentiating w.r.t. $\mathbf{x}_t$ gives

$$\nabla_{\mathbf{x}_t} \log p^{\omega}(\mathbf{x}_t \mid \mathbf{pa}) = \nabla_{\mathbf{x}_t} \log p(\mathbf{x}_t) + \sum_{m=1}^{M} \omega_m \nabla_{\mathbf{x}_t} \log p(\mathbf{pa}^{(m)} \mid \mathbf{x}_t). \tag{27}$$

To relate $\nabla_{\mathbf{x}_t} \log p(\mathbf{pa}^{(m)} \mid \mathbf{x}_t)$ to conditional scores, we apply Bayes' rule: $p(\mathbf{pa}^{(m)} \mid \mathbf{x}_t) = \frac{p(\mathbf{x}_t \mid \mathbf{pa}^{(m)}) p(\mathbf{pa}^{(m)})}{p(\mathbf{x}_t)}$. Taking logs and differentiating yields: (noting that $p(\mathbf{pa}^{(m)})$ is constant w.r.t. $\mathbf{x}_t$)

$$\nabla_{\mathbf{x}_t} \log p(\mathbf{pa}^{(m)} \mid \mathbf{x}_t) = \nabla_{\mathbf{x}_t} \log p(\mathbf{x}_t \mid \mathbf{pa}^{(m)}) - \nabla_{\mathbf{x}_t} \log p(\mathbf{x}_t). \tag{28}$$

Substituting Eq. (28) into Eq. (27) gives

$$\nabla_{\mathbf{x}_t} \log p^{\omega}(\mathbf{x}_t \mid \mathbf{pa}) = \nabla_{\mathbf{x}_t} \log p(\mathbf{x}_t) + \sum_{m=1}^{M} \omega_m \Big( \nabla_{\mathbf{x}_t} \log p(\mathbf{x}_t \mid \mathbf{pa}^{(m)}) - \nabla_{\mathbf{x}_t} \log p(\mathbf{x}_t) \Big), \tag{29}$$

where $\mathbf{pa}^{(m)}$ is the $m$-th group of attributes, and $\omega_m$ is the guidance weight for that group.

**From scores to $\epsilon$-parameterization.**    In $\epsilon$-prediction diffusion models, scores are implemented by the denoiser $\epsilon_\theta(\mathbf{x}_t, t, \cdot)$; we therefore approximate each conditional score by evaluating the model with a masked conditioning input that retains only group $m$. This yields the practical guided noise prediction:

$$\boldsymbol{\epsilon}_{\mathrm{CFG}} = \boldsymbol{\epsilon}_\theta(\mathbf{x}_t, t, \varnothing) + \sum_{m=1}^{M} \omega_m \cdot \Big( \boldsymbol{\epsilon}_\theta(\mathbf{x}_t, t, \underline{\mathbf{c}}^{(m)}) - \boldsymbol{\epsilon}_\theta(\mathbf{x}_t, t, \varnothing) \Big), \tag{30}$$

where $\underline{\mathbf{c}}^{(m)}$ denotes the masked condition vector in which only group $m$ is retained and all others are null-tokenized, as defined in Eq. (13).

# C. DDIM-based Diffusion Counterfactual Generation.

**Overview.** Given an observed image $\mathbf{x}_0$ and its associated semantic attributes $\mathbf{pa}$, counterfactual generation proceeds as follows: (i) in the *abduction* step, the observed image is inverted into a latent noise variable via a deterministic DDIM inversion conditioned on the factual attributes; (ii) in the *action* step, a causal intervention is applied in the semantic attribute space to obtain counterfactual attributes; (iii) in the *prediction* step, a guided reverse diffusion trajectory is run to generate the counterfactual image. The abduction and action steps are shared across all methods considered in this work, while different guidance formulations affect only the prediction step.

**Abduction.** We first describe the *abduction* step, which aims to infer the latent exogenous noise variable $\mathbf{u}$ corresponding to an observed image $\mathbf{x}_0$ under its factual semantic attributes $\mathbf{pa}$. Following prior diffusion-based counterfactual methods (Sanchez & Tsaftaris, 2022; Rasal et al., 2025), we implement abduction via a *conditional deterministic DDIM inversion*, as summarised in Algorithm 1.

---

**Algorithm 1** Abduction via Deterministic DDIM Inversion (shared across all methods)

---

**Input:** observed image $\mathbf{x}_0$, factual attributes $\mathbf{pa}$, attribute-split embedder $\{\mathcal{E}_i\}_{i=1}^{K}$, DDIM schedule $\{\alpha_t\}_{t=0}^{T}$
**Embed factual attributes.**
  1: $\mathbf{c} \leftarrow \text{concat}\left(\mathcal{E}_1(pa_1), \ldots, \mathcal{E}_K(pa_K)\right)$
**Abduction (DDIM inversion):**
  1: **for** $t = 0$ to $T-1$ **do**
  2:     $\hat{\mathbf{x}}_0 \leftarrow \frac{1}{\sqrt{\alpha_t}}\left(\mathbf{x}_t - \sqrt{1-\alpha_t}\,\boldsymbol{\epsilon}_\theta(\mathbf{x}_t, t, \mathbf{c})\right)$
  3:     $\mathbf{x}_{t+1} \leftarrow \sqrt{\alpha_{t+1}}\,\hat{\mathbf{x}}_0 + \sqrt{1-\alpha_{t+1}}\,\boldsymbol{\epsilon}_\theta(\mathbf{x}_t, t, \mathbf{c})$
  4: **end for**
**Output:** exogenous noise estimate $\mathbf{u} \leftarrow \mathbf{x}_T$

---

**Action.** We apply an intervention to the semantic attributes $\mathbf{pa}$ (e.g., `do(Male=1)`) and propagate its effect through the causal graph using invertible flow-based models, following prior work on flow-based structural causal models (Pawlowski et al., 2020; Ribeiro et al., 2023; Rasal et al., 2025). This produces a counterfactual attribute vector $\widetilde{\mathbf{pa}}$, whose embedding $\tilde{\mathbf{c}}$ is subsequently used for counterfactual prediction. The action step is summarized in Algorithm 2.

---

**Algorithm 2** Action via Attribute Intervention (shared across all methods)

---

**Input:** factual attributes $\mathbf{pa}$, intervention index set $\mathcal{I} \subseteq \{1, \ldots, K\}$, intervention values $\mathbf{pa}'_{\mathcal{I}}$, attribute-split embedder $\{\mathcal{E}_i\}_{i=1}^{K}$
**Action (Invertible Normalizing Flow-based Structural Causal Models):**
  1: $\widetilde{\mathbf{pa}} \leftarrow \text{do}\left(\mathbf{pa}_{\mathcal{I}} \leftarrow \mathbf{pa}'_{\mathcal{I}}\right)$
  2: $\tilde{\mathbf{c}} \leftarrow \text{concat}\left(\mathcal{E}_1(\widetilde{pa}_1), \ldots, \mathcal{E}_K(\widetilde{pa}_K)\right)$
**Output:** intervened conditioning $\tilde{\mathbf{c}}$

---

**Counterfactual prediction with CFG.** Counterfactual prediction evaluates the structural assignment $\tilde{\mathbf{x}} := f_\theta(\mathbf{u}, \widetilde{\mathbf{pa}})$ under the intervened semantic attributes. Given the exogenous noise estimate $\mathbf{u}$ obtained during abduction and the counterfactual conditioning embedding $\tilde{\mathbf{c}}$ produced by the action step, the counterfactual image is generated by running the deterministic reverse DDIM trajectory conditioned on $\tilde{\mathbf{c}}$. Following prior work (Rasal et al., 2025; Sanchez et al., 2022b; Komanduri et al., 2024), classifier-free guidance (CFG) is applied during the reverse process. The full procedure is summarized in Algorithm 3.

**Counterfactual prediction with FCFG.** Counterfactual prediction with FCFG follows the same reverse DDIM decoding procedure as standard CFG, but replaces the global guidance with group-wise guidance defined over masked conditioning embeddings. Given the exogenous noise estimate $\mathbf{u}$ from abduction and the counterfactual conditioning embedding $\tilde{\mathbf{c}}$ from the action step, counterfactual samples are generated by running the deterministic reverse DDIM trajectory using the FCFG-guided score defined in Eq. (14). The full procedure is summarized in Algorithm 4.

---

**Algorithm 3** Counterfactual Prediction with CFG

---

**Input:** exogenous noise estimate $\mathbf{u}$ (from *abduction*), counterfactual conditioning $\tilde{\mathbf{c}}$ (from *prediction*), guidance weight $\omega \geq 0$, DDIM schedule $\{\alpha_t\}_{t=0}^{T}$, null conditioning token $\varnothing$

**Counterfactual Prediction:**

1: $\mathbf{x}_T \leftarrow \mathbf{u}$
2: **for** $t = T$ down to 1 **do**
3: $\quad \boldsymbol{\epsilon}_{\text{CFG}} \leftarrow \boldsymbol{\epsilon}_\theta(\mathbf{x}_t, t, \varnothing) + \omega\big(\boldsymbol{\epsilon}_\theta(\mathbf{x}_t, t, \tilde{\mathbf{c}}) - \boldsymbol{\epsilon}_\theta(\mathbf{x}_t, t, \varnothing)\big)$
4: $\quad \hat{\mathbf{x}}_0 \leftarrow \frac{1}{\sqrt{\alpha_t}}\big(\mathbf{x}_t - \sqrt{1 - \alpha_t}\,\boldsymbol{\epsilon}_{\text{CFG}}\big)$
5: $\quad \mathbf{x}_{t-1} \leftarrow \sqrt{\alpha_{t-1}}\,\hat{\mathbf{x}}_0 + \sqrt{1 - \alpha_{t-1}}\,\boldsymbol{\epsilon}_{\text{CFG}}$
6: **end for**

**Output:** counterfactual image $\tilde{\mathbf{x}} \leftarrow \mathbf{x}_0$

---

**Algorithm 4** Counterfactual Prediction with FCFG (Ours; General)

---

**Input:** exogenous noise estimate $\mathbf{u}$ (from *abduction*), counterfactual conditioning embedding $\tilde{\mathbf{c}}$ (from *action*), masked group embeddings $\{\tilde{\underline{\mathbf{c}}}^{(m)}\}_{m=1}^{M}$, group-wise guidance weights $\{\omega_m\}_{m=1}^{M}$, DDIM schedule $\{\alpha_t\}_{t=0}^{T}$, null conditioning token $\varnothing$

**Counterfactual Prediction:**

1: $\mathbf{x}_T \leftarrow \mathbf{u}$
2: **for** $t = T$ down to 1 **do**
3: $\quad \boldsymbol{\epsilon}_{\text{FCFG}} \leftarrow \boldsymbol{\epsilon}_\theta(\mathbf{x}_t, t, \varnothing)$
4: $\quad$ **for** $m = 1$ to $M$ **do**
5: $\quad\quad \boldsymbol{\epsilon}_{\text{FCFG}} \leftarrow \boldsymbol{\epsilon}_{\text{FCFG}} + \omega_m\Big(\boldsymbol{\epsilon}_\theta(\mathbf{x}_t, t, \tilde{\underline{\mathbf{c}}}^{(m)}) - \boldsymbol{\epsilon}_\theta(\mathbf{x}_t, t, \varnothing)\Big)$
6: $\quad$ **end for**
7: $\quad \hat{\mathbf{x}}_0 \leftarrow \frac{1}{\sqrt{\alpha_t}}\big(\mathbf{x}_t - \sqrt{1 - \alpha_t}\,\boldsymbol{\epsilon}_{\text{FCFG}}\big)$
8: $\quad \mathbf{x}_{t-1} \leftarrow \sqrt{\alpha_{t-1}}\,\hat{\mathbf{x}}_0 + \sqrt{1 - \alpha_{t-1}}\,\boldsymbol{\epsilon}_{\text{FCFG}}$
9: **end for**

**Output:** counterfactual image $\tilde{\mathbf{x}} \leftarrow \mathbf{x}_0$

---

**Two-group instantiation.** In most of our experiments, we adopt a two-group partition with *affected* and *invariant* attributes ($M{=}2$), refer to Sec. 3.3 for more details. This can be summarized in Algorithm 5.

---

**Algorithm 5** Counterfactual Prediction with FCFG (Ours; Two-group partition)

---

**Input:** exogenous noise estimate $\mathbf{u}$ (from *abduction*), counterfactual conditioning embedding $\tilde{\mathbf{c}}$ (from *action*), masked embeddings $\tilde{\underline{\mathbf{c}}}^{\text{aff}}$, $\tilde{\underline{\mathbf{c}}}^{\text{inv}}$, guidance weights $\omega_{\text{aff}}, \omega_{\text{inv}}$, DDIM schedule $\{\alpha_t\}_{t=0}^{T}$, null conditioning token $\varnothing$

**Counterfactual Prediction:**

1: $\mathbf{x}_T \leftarrow \mathbf{u}$
2: **for** $t = T$ down to 1 **do**
3: $\quad \boldsymbol{\epsilon}_{\text{FCFG}} \leftarrow \boldsymbol{\epsilon}_\theta(\mathbf{x}_t, t, \varnothing) + \omega_{\text{aff}}\big(\boldsymbol{\epsilon}_\theta(\mathbf{x}_t, t, \tilde{\underline{\mathbf{c}}}^{\text{aff}}) - \boldsymbol{\epsilon}_\theta(\mathbf{x}_t, t, \varnothing)\big) + \omega_{\text{inv}}\big(\boldsymbol{\epsilon}_\theta(\mathbf{x}_t, t, \tilde{\underline{\mathbf{c}}}^{\text{inv}}) - \boldsymbol{\epsilon}_\theta(\mathbf{x}_t, t, \varnothing)\big)$
4: $\quad \hat{\mathbf{x}}_0 \leftarrow \frac{1}{\sqrt{\alpha_t}}\big(\mathbf{x}_t - \sqrt{1 - \alpha_t}\,\boldsymbol{\epsilon}_{\text{FCFG}}\big)$
5: $\quad \mathbf{x}_{t-1} \leftarrow \sqrt{\alpha_{t-1}}\,\hat{\mathbf{x}}_0 + \sqrt{1 - \alpha_{t-1}}\,\boldsymbol{\epsilon}_{\text{FCFG}}$
6: **end for**

**Output:** counterfactual image $\tilde{\mathbf{x}} \leftarrow \mathbf{x}_0$

---

**Extensions of FCFG.** The core idea of FCFG, namely factoring the guidance signal across causally or semantically defined attribute groups, can be integrated with guidance variants beyond standard classifier-free guidance. In this work, we extend FCFG to CFG++ (Chung et al., 2025) and to adaptive projection guidance (APG) (Sadat et al., 2025) by applying the same factoring mechanism to their respective guided score formulations. The resulting methods, FCFG++ and FAPG, are described in App. J and App. K, respectively, where we present quantitative and qualitative results showing that both CFG++ and APG exhibit attribute amplification, and that integrating FCFG with these methods effectively mitigates this issue.

# D. Implementation Details

**Architecture.**   We adopt the commonly used U-net backbone (Dhariwal & Nichol, 2021) for all diffusion models in this work. We modify it to support CFG and FCFG. Each conditioning attribute is projected via a dedicated MLP embedder (Sec. 3.1), and the resulting embeddings are concatenated with the timestep embedding. During training, we apply exponential moving average (EMA) to model weights for improved stability. All images are normalized to the range [-1,1]. The complete architecture and training configurations for each dataset are summarized in Table A.2.

*Table A.2.* Training and architecture of diffusion U-Net configurations used in our experiments. *We did not evaluate on the whole test set due to the high computational cost of diffusion sampling.

| PARAMETER | CELEBA-HQ | EMBED | MIMIC-CXR |
|---|---|---|---|
| TRAIN SET SIZE | 24,000 | 151,948 | 62,336 |
| VALIDATION SET SIZE | 3,000 | 7,156 | 9,968 |
| TEST SET SIZE* | 3,000 | 43,669 | 30,535 |
| RESOLUTION | $64 \times 64 \times 3$ | $192 \times 192 \times 1$ | $192 \times 192 \times 1$ |
| BATCH SIZE | 128 | 48 | 48 |
| TRAINING EPOCHS | 6000 | 5000 | 5000 |
| BASE CHANNELS (U-NET) | 64 | 64 | 32 |
| CHANNEL MULTIPLIERS | [1,2,4,8] | [1,1,2,2,4,4] | [1,2,3,4,5,6] |
| ATTENTION RESOLUTIONS | [16] | - | - |
| RESNET BLOCKS | 2 | 2 | 2 |
| DROPOUT RATE | 0.1 | 0.0 | 0.0 |
| NUM. CONDITIONING ATTRS | 3 | 2 | 4 |
| COND. EMBEDDING DIM | $3 \times 32$ | $2 \times 32$ | $4 \times 32$ |
| NOISE SCHEDULE | LINEAR | COSINE | LINEAR |
| LEARNING RATE | 1e-4 | | |
| OPTIMISER | ADAM (WD 1e-4) | | |
| EMA DECAY | 0.9999 | | |
| TRAINING STEPS $T$ | 1000 | | |
| LOSS | MSE (noise prediction) | | |

**Training Procedure for FCFG.**   The training of our proposed FCFG follows the same setup as standard Classifier-Free Guidance (CFG). Specifically, we apply classifier-free dropout (Ho & Salimans, 2022) by replacing the entire conditioning vector with a null token (i.e., zero vector) with probability $p_\varnothing = 0.5$. Unlike the typical choice of $p_\varnothing = 0.2$, we found that using $p_\varnothing = 0.5$ better preserves identity, which is particularly important for counterfactual generation tasks. Note that we apply dropout to all attributes jointly, rather than selectively masking subsets. One could alternatively consider group-wise dropout—nullifying only a random subset of attribute groups—but such partial masking may encourage the model to over-rely on the remaining visible attributes, making the resulting guidance less disentangled and less robust. We leave this as an interesting direction for future exploration.

**Computation Resources.**   All experiments were conducted on servers equipped with multiple NVIDIA GPUs, including L40S and similar models, each with approximately 48GB of memory. Training each model typically takes around one week on one GPU. Due to the high computational cost of diffusion-based sampling, generating counterfactuals for each intervention (`do(key)`) and each guidance configuration takes approximately one day for the MIMIC-CXR and EMBED datasets, and around 7 hours for CelebA-HQ.

**Evaluation.**   Due to the computational cost of diffusion sampling, we evaluate on fixed, balanced subsets rather than the full test sets. For CelebA-HQ and EMBED, we use 1,000 samples each, selected to ensure an even distribution across the conditioning attributes. For MIMIC-CXR, we evaluate on 1,500 samples, stratified to balance race groups. These fixed subsets are reused across all experiments to enable fair and consistent comparisons. To generate counterfactuals, we use DDIM sampling with 1,000 time steps, as we find this setting achieves stronger identity preservation compared to shorter schedules—an essential property for counterfactual evaluation. This setup allows us to assess attribute-specific phenomena such as amplification and reversibility while keeping the sampling cost manageable.

**Scalability of Attribute-Split Embeddings.** Using a separate MLP for each conditioning attribute may appear unscalable at first glance. However, in our implementation, each per-attribute MLP is intentionally lightweight. Specifically, we use a $1 \rightarrow 32 \rightarrow 32$ architecture. The resulting 32-dimensional embeddings are concatenated to form a $32 \times K$ conditioning vector, where $K$ denotes the number of attributes, and the per-attribute embedding dimension can be adjusted if needed. Even with a relatively large number of attributes (e.g., $K{=}40$), this design results in only 44,800 parameters in total, which is negligible relative to the overall diffusion model size. In practice, our experiments involve far fewer attributes (typically $K{=}2$–3), making the overhead even smaller. By contrast, encoding the same attributes using a single joint MLP that produces a $32 \times K$-dimensional conditioning vector would require a substantially larger network. For example, a $40 \rightarrow 1280 \rightarrow 1280$ MLP contains approximately 1.69 million parameters, making it significantly less parameter-efficient than the attribute-split design. The proposed approach therefore achieves improved parameter efficiency while preserving explicit attribute-level structure in the conditioning space.

# E. Additional Baseline and Ablation Results

This appendix provides additional results complementing the main experiments, including comparisons with existing counterfactual generation methods, image-quality evaluation using FID, an ablation on the classifier-free dropout probability $p_\emptyset$, and a practical cost comparison. These results further support the conclusion that FCFG improves the target/off-target trade-off while preserving image quality and maintaining a favourable computational profile.

## E.1. Comparison with Existing Counterfactual Generation Methods

We compare FCFG with HVAE (Ribeiro et al., 2023), HVAE-soft (Xia et al., 2024), and SA-DCG (Rasal et al., 2025). These methods use structured causal information but incorporate it differently. HVAE and HVAE-soft use predictor-based counterfactual training objectives, while SA-DCG uses a diffusion-autoencoder framework with semantic abduction. In contrast, FCFG modifies only the inference-time guidance rule of a trained conditional diffusion model.

Table A.3 summarizes the comparison. On CelebA-HQ, SA-DCG improves reversibility relative to standard CFG but still exhibits off-target amplification under global guidance. FCFG reduces this amplification while also improving reversibility. On MIMIC-CXR, FCFG achieves comparable or stronger target intervention than HVAE, HVAE-soft, and CFG, while reducing off-target change by a large margin.

*Table A.3.* **Comparison with existing counterfactual generation methods.** On CelebA-HQ, results are reported under `do(Smiling)`, with off-target AUC measured on Male. On MIMIC-CXR, off-target AUC denotes the average change over non-intervened attributes. FCFG preserves target intervention effectiveness while substantially reducing off-target amplification. Higher target AUC is better; lower off-target AUC, MAE, and LPIPS are better.

| Dataset | Method | Guidance configuration | Target AUC ↑ | Off-target AUC ↓ | MAE ↓ | LPIPS ↓ |
|---|---|---|---|---|---|---|
| CelebA-HQ, `do(Smiling)` | CFG (Ho & Salimans, 2022) | $\omega = 2.0$ | +11.2 | +2.7 | 0.203 | 0.127 |
| | SA-DCG (Rasal et al., 2025) | $\omega = 2.0$ | +12.7 | +3.0 | 0.178 | 0.111 |
| | FCFG (Ours) | $\omega_{\text{aff}} = 2.0, \omega_{\text{inv}} = 1.2$ | +10.5 | +0.4 | **0.146** | **0.098** |
| | CFG (Ho & Salimans, 2022) | $\omega = 3.0$ | +12.3 | +3.0 | 0.263 | 0.155 |
| | SA-DCG (Rasal et al., 2025) | $\omega = 3.0$ | +12.9 | +3.0 | 0.221 | 0.145 |
| | FCFG (Ours) | $\omega_{\text{aff}} = 3.0, \omega_{\text{inv}} = 1.2$ | **+13.1** | **−1.5** | **0.177** | **0.122** |
| MIMIC-CXR, `do(sex)` | HVAE (Ribeiro et al., 2023) | – | +7.4 | +9.0 | – | – |
| | HVAE-soft (Xia et al., 2024) | – | +7.3 | +5.7 | – | – |
| | CFG (Ho & Salimans, 2022) | $\omega = 3.0$ | +7.5 | +10.7 | – | – |
| | FCFG (Ours) | $\omega_{\text{aff}} = 3.0, \omega_{\text{inv}} = 1.2$ | **+7.6** | **+0.6** | – | – |
| MIMIC-CXR, `do(finding)` | HVAE (Ribeiro et al., 2023) | – | +17.1 | +5.1 | – | – |
| | HVAE-soft (Xia et al., 2024) | – | +17.3 | +5.0 | – | – |
| | CFG (Ho & Salimans, 2022) | $\omega = 3.0$ | +17.8 | +5.3 | – | – |
| | FCFG (Ours) | $\omega_{\text{aff}} = 3.0, \omega_{\text{inv}} = 1.2$ | **+18.8** | **+0.6** | – | – |

## E.2. Image Quality Evaluation

We evaluate general image quality using FID on CelebA-HQ. Although FID is not a complete counterfactual metric, since a valid counterfactual should intentionally differ from the source image along the intervened attribute, it provides a useful complementary measure of realism. As shown in Table A.4, FCFG consistently achieves lower FID than global CFG across all three interventions and guidance strengths. This suggests that reducing off-target amplification also helps preserve overall image quality.

*Table A.4.* **FID comparison on CelebA-HQ.** Lower is better. FCFG consistently improves FID over global CFG across interventions.

| Method | do(Smiling) | do(Male) | do(Young) |
|---|---|---|---|
| CFG, $\omega = 2.0$ | 19.0 | 19.3 | 19.8 |
| FCFG, $\omega_{\text{aff}} = 2.0, \omega_{\text{inv}} = 1.2$ | **18.3** | **18.8** | **19.2** |
| CFG, $\omega = 3.0$ | 21.3 | 21.2 | 22.2 |
| FCFG, $\omega_{\text{aff}} = 3.0, \omega_{\text{inv}} = 1.2$ | **19.6** | **19.4** | **20.3** |

### E.3. Ablation on Classifier-Free Dropout

Our main experiments use classifier-free dropout probability $p_\emptyset = 0.5$. To justify this choice, we compare it against the commonly used $p_\emptyset = 0.2$ on CelebA-HQ under do(Smiling). Table A.5 reports reversibility results for CFG, FCFG, CFG++, FCFG++, APG, and FAPG. Across all guidance formulations, $p_\emptyset = 0.5$ yields lower MAE and LPIPS than $p_\emptyset = 0.2$, indicating improved identity preservation. We therefore use $p_\emptyset = 0.5$ in our main experiments.

*Table A.5.* **Ablation on classifier-free dropout $p_\emptyset$ under do(Smiling) on CelebA-HQ.** Lower MAE and LPIPS indicate better reversibility and identity preservation. Across all guidance variants, $p_\emptyset = 0.5$ improves reversibility compared with $p_\emptyset = 0.2$.

| Guidance | MAE, $p_\emptyset = 0.2$ | MAE, $p_\emptyset = 0.5$ | LPIPS, $p_\emptyset = 0.2$ | LPIPS, $p_\emptyset = 0.5$ |
|---|---|---|---|---|
| CFG | 0.309 | **0.263** | 0.180 | **0.155** |
| FCFG | 0.206 | **0.177** | 0.140 | **0.122** |
| CFG++ | 0.298 | **0.251** | 0.177 | **0.154** |
| FCFG++ | 0.212 | **0.175** | 0.149 | **0.121** |
| APG | 0.280 | **0.269** | 0.187 | **0.166** |
| FAPG | 0.231 | **0.195** | 0.161 | **0.136** |

### E.4. Practical Cost and Sampling Overhead

FCFG also offers a favourable simplicity–control trade-off. Unlike HVAE and HVAE-soft, it does not require an additional predictor-based counterfactual training objective or auxiliary pretrained predictors. Unlike SA-DCG, it avoids a heavier diffusion-autoencoder framework and additional identity-preserving inference-time optimization.

The main computational overhead of FCFG occurs at sampling time rather than training time. Standard CFG requires two denoiser evaluations per timestep, one unconditional and one conditional. In contrast, FCFG with $M$ groups requires $M+1$ denoiser evaluations per timestep: one unconditional evaluation and one masked conditional evaluation for each group. In our main two-group setting, this corresponds to three denoiser evaluations per timestep. Thus, the sampling cost increases linearly with the number of guidance groups, while the training-time parameter and memory overhead of the attribute-specific MLP embedders remains negligible relative to the diffusion backbone.

Table A.6 compares FCFG with SA-DCG on CelebA-HQ. FCFG is lighter and faster in this setting while achieving a better target/off-target trade-off and improved reversibility.

*Table A.6.* **Practical comparison with SA-DCG on CelebA-HQ.** Results are reported under do(Smiling) with the stronger guidance setting. FCFG is lighter and faster while achieving a better target/off-target trade-off and improved reversibility.

| Method | Params | Time | Target AUC $\uparrow$ | Off-target AUC $\downarrow$ | MAE $\downarrow$ | LPIPS $\downarrow$ |
|---|---|---|---|---|---|---|
| SA-DCG (Rasal et al., 2025) | 72.6M | 180s | +12.9 | +3.0 | 0.221 | 0.145 |
| FCFG (Ours) | 60.5M | 140s | **+13.1** | **−1.5** | **0.177** | **0.122** |

# F. CelebA-HQ

## F.1. Dataset details

For the CelebA-HQ dataset (Karras et al., 2017), we select `Smiling`, `Male`, and `Young` as three independent binary variables. These are among the most reliably annotated attributes in CelebA, each achieving over 95% consistency in manual labeling (Wu et al., 2023). Moreover, they exhibit low inconsistency across duplicate face pairs (e.g., `Male`: 0.005; `Smiling`: 0.077), suggesting minimal label noise. We assume these variables to be independent, as our goal is to isolate and analyze attribute amplification under global classifier-free guidance (CFG), which is more interpretable with uncorrelated factors. As shown in Fig. A.1, the Pearson correlation matrix confirms weak pairwise correlations among these attributes. Although a moderate negative correlation is observed between `Male` and `Young` ($\rho = -0.33$), we attribute this to dataset bias rather than a true causal dependency, and proceed by modeling them as independent.

*Figure A.1.* Pearson correlation matrix of CelebA-HQ attributes: `Smiling`, `Male`, and `Young`. While a moderate negative correlation is observed between `Male` and `Young` ($\rho = -0.33$), we regard this as a spurious correlation likely stemming from dataset bias rather than a meaningful causal relationship. Therefore, for the purposes of our analysis, we assume these attributes to be independent.

## F.2. Extra quantitative result for Celeba-HQ

*Table A.7.* CelebA-HQ: Effectiveness (ROC-AUC ↑) and reversibility (MAE, LPIPS ↓) when varying $\omega_{\mathrm{aff}}$. Compared to global CFG, FCFG achieves strong intervention effectiveness on intervened attributes while mitigating amplification on invariant attributes. For larger $\omega_{\mathrm{aff}}$, we set $\omega_{\mathrm{inv}}{=}1.2$ to prevent degradation of invariant attributes. Across all interventions, FCFG maintains better reversibility than global CFG with matched guidance strength.

| do(key) | Guidance | Guidance configurations | Smiling AUC/Δ | Male AUC/Δ | Young AUC/Δ | MAE | LPIPS |
|---|---|---|---|---|---|---|---|
| do(Smiling) | CFG (Ho & Salimans, 2022) | $\omega = 1.0$ | 86.5 / +0.0 | 96.9 / +0.0 | 78.6 / +0.0 | 0.113 | 0.082 |
| | | $\omega = 1.2$ | 91.7 / +5.2 | 98.5 / +1.6 | 80.9 / +2.3 | 0.133 | 0.091 |
| | | $\omega = 1.5$ | 95.5 / +9.0 | 99.2 / +2.3 | 84.3 / +5.7 | 0.163 | 0.111 |
| | | $\omega = 1.7$ | 96.7 / +10.2 | 99.4 / +2.5 | 85.7 / +7.1 | 0.179 | 0.119 |
| | | $\omega = 2.0$ | 97.7 / +11.2 | 99.6 / +2.7 | 88.3 / +9.7 | 0.203 | 0.127 |
| | | $\omega = 2.5$ | 98.6 / +12.1 | 99.7 / +2.8 | 89.3 / +10.7 | 0.234 | 0.142 |
| | | $\omega = 3.0$ | 98.8 / +12.3 | 99.9 / +3.0 | 90.3 / +11.7 | 0.263 | 0.155 |
| | FCFG (Ours) | $\omega_{\mathrm{aff}} = 1.2, \omega_{\mathrm{inv}} = 1.0$ | 89.9 / +3.4 | 95.8 / -1.1 | 77.8 / -0.8 | 0.128 | 0.093 |
| | | $\omega_{\mathrm{aff}} = 1.5, \omega_{\mathrm{inv}} = 1.2$ | 93.9 / +7.4 | 97.5 / +0.6 | 79.7 / +1.1 | 0.136 | 0.092 |
| | | $\omega_{\mathrm{aff}} = 1.7, \omega_{\mathrm{inv}} = 1.2$ | 95.4 / +8.9 | 97.5 / +0.6 | 79.9 / +1.3 | 0.141 | 0.095 |
| | | $\omega_{\mathrm{aff}} = 2.0, \omega_{\mathrm{inv}} = 1.2$ | 97.0 / +10.5 | 97.3 / +0.4 | 79.7 / +1.1 | 0.146 | 0.098 |
| | | $\omega_{\mathrm{aff}} = 2.5, \omega_{\mathrm{inv}} = 1.2$ | 98.9 / +12.4 | 96.1 / -0.8 | 77.8 / -0.8 | 0.164 | 0.112 |
| | | $\omega_{\mathrm{aff}} = 3.0, \omega_{\mathrm{inv}} = 1.2$ | 99.6 / +13.1 | 95.4 / -1.5 | 77.6 / -1.0 | 0.177 | 0.122 |
| do(Male) | CFG (Ho & Salimans, 2022) | $\omega = 1.0$ | 86.6 / +0.0 | 91.8 / +0.0 | 79.8 / +0.0 | 0.115 | 0.079 |
| | | $\omega = 1.2$ | 90.1 / +3.5 | 95.1 / +3.3 | 80.8 / +1.0 | 0.127 | 0.088 |
| | | $\omega = 1.5$ | 93.3 / +6.7 | 97.2 / +5.4 | 82.0 / +2.2 | 0.158 | 0.111 |
| | | $\omega = 1.7$ | 94.7 / +8.1 | 97.5 / +5.7 | 83.7 / +3.9 | 0.175 | 0.123 |
| | | $\omega = 2.0$ | 96.0 / +9.4 | 97.9 / +6.1 | 85.0 / +5.2 | 0.202 | 0.139 |
| | | $\omega = 2.5$ | 97.6 / +11.0 | 98.4 / +6.6 | 87.5 / +7.7 | 0.238 | 0.156 |
| | | $\omega = 3.0$ | 98.2 / +11.6 | 99.2 / +7.4 | 90.2 / +10.4 | 0.267 | 0.171 |
| | FCFG (Ours) | $\omega_{\mathrm{aff}} = 1.2, \omega_{\mathrm{inv}} = 1.0$ | 85.1 / -1.5 | 91.3 / -0.5 | 79.0 / -0.8 | 0.137 | 0.097 |
| | | $\omega_{\mathrm{aff}} = 1.5, \omega_{\mathrm{inv}} = 1.2$ | 88.3 / +1.7 | 93.8 / +2.0 | 78.9 / -0.9 | 0.149 | 0.101 |
| | | $\omega_{\mathrm{aff}} = 1.7, \omega_{\mathrm{inv}} = 1.2$ | 88.1 / +1.5 | 95.8 / +4.0 | 77.5 / -2.3 | 0.151 | 0.103 |
| | | $\omega_{\mathrm{aff}} = 2.0, \omega_{\mathrm{inv}} = 1.2$ | 88.0 / +1.4 | 97.8 / +6.0 | 77.0 / -2.8 | 0.158 | 0.109 |
| | | $\omega_{\mathrm{aff}} = 2.5, \omega_{\mathrm{inv}} = 1.2$ | 87.4 / +0.8 | 99.4 / +7.6 | 76.2 / -3.6 | 0.171 | 0.118 |
| | | $\omega_{\mathrm{aff}} = 3.0, \omega_{\mathrm{inv}} = 1.2$ | 87.4 / +0.8 | 99.7 / +7.9 | 75.9 / -3.9 | 0.188 | 0.130 |
| do(Young) | CFG (Ho & Salimans, 2022) | $\omega = 1.0$ | 87.5 / +0.0 | 95.7 / +0.0 | 62.3 / +0.0 | 0.115 | 0.085 |
| | | $\omega = 1.2$ | 90.8 / +3.3 | 97.4 / +1.7 | 64.5 / +2.2 | 0.130 | 0.088 |
| | | $\omega = 1.5$ | 95.6 / +8.1 | 99.3 / +3.6 | 66.3 / +4.0 | 0.166 | 0.110 |
| | | $\omega = 1.7$ | 96.7 / +9.2 | 99.4 / +3.7 | 67.8 / +5.5 | 0.183 | 0.119 |
| | | $\omega = 2.0$ | 97.7 / +10.2 | 99.6 / +3.9 | 69.5 / +7.2 | 0.204 | 0.130 |
| | | $\omega = 2.5$ | 98.3 / +10.8 | 99.8 / +4.1 | 73.5 / +11.2 | 0.234 | 0.146 |
| | | $\omega = 3.0$ | 98.5 / +11.0 | 99.9 / +4.2 | 77.7 / +15.4 | 0.261 | 0.160 |
| | FCFG (Ours) | $\omega_{\mathrm{aff}} = 1.2, \omega_{\mathrm{inv}} = 1.0$ | 87.4 / -0.1 | 95.0 / -0.7 | 63.2 / +0.9 | 0.129 | 0.095 |
| | | $\omega_{\mathrm{aff}} = 1.5, \omega_{\mathrm{inv}} = 1.2$ | 90.0 / +2.5 | 96.7 / +1.0 | 67.4 / +5.1 | 0.147 | 0.100 |
| | | $\omega_{\mathrm{aff}} = 1.7, \omega_{\mathrm{inv}} = 1.2$ | 89.2 / +1.7 | 96.1 / +0.4 | 71.3 / +9.0 | 0.150 | 0.103 |
| | | $\omega_{\mathrm{aff}} = 2.0, \omega_{\mathrm{inv}} = 1.2$ | 87.9 / +0.4 | 94.5 / -1.2 | 75.6 / +13.3 | 0.157 | 0.110 |
| | | $\omega_{\mathrm{aff}} = 2.5, \omega_{\mathrm{inv}} = 1.2$ | 88.5 / +1.0 | 91.9 / -3.8 | 81.8 / +19.5 | 0.172 | 0.125 |
| | | $\omega_{\mathrm{aff}} = 3.0, \omega_{\mathrm{inv}} = 1.2$ | 86.7 / -0.8 | 90.0 / -5.7 | 87.6 / +25.3 | 0.188 | 0.136 |

*Table A.8.* CelebA-HQ: Effectiveness (ROC-AUC ↑) and reversibility (MAE, LPIPS ↓) when varying $\omega_{\mathrm{inv}}$. Increasing $\omega_{\mathrm{inv}}$ consistently increases effectiveness on invariant variables, while degrading intervention effectiveness. When $\omega_{\mathrm{inv}}{=}2.5$, the amplification on invariant attributes becomes comparable to that of the global CFG setting with $\omega{=}2.5$.

| do(key) | Guidance | Guidance configuration | Smiling AUC/Δ | Male AUC/Δ | Young AUC/Δ | MAE | LPIPS |
|---|---|---|---|---|---|---|---|
| do(Smiling) | CFG (Ho & Salimans, 2022) | $\omega = 1.0$ | 86.5 / +0.0 | 96.9 / +0.0 | 78.6 / +0.0 | 0.113 | 0.082 |
| | | $\omega = 2.5$ | 98.6 / +12.1 | 99.7 / +2.8 | 89.3 / +10.7 | 0.234 | 0.142 |
| | FCFG (Ours) | $\omega_{\mathrm{aff}} = 2.5, \omega_{\mathrm{inv}} = 1.0$ | 99.1 / +12.6 | 92.6 / -4.3 | 75.2 / -3.4 | 0.165 | 0.118 |
| | | $\omega_{\mathrm{aff}} = 2.5, \omega_{\mathrm{inv}} = 1.2$ | 98.9 / +12.4 | 96.1 / -0.8 | 77.8 / -0.8 | 0.164 | 0.112 |
| | | $\omega_{\mathrm{aff}} = 2.5, \omega_{\mathrm{inv}} = 1.5$ | 98.3 / +11.8 | 98.2 / +1.3 | 82.6 / +4.0 | 0.177 | 0.118 |
| | | $\omega_{\mathrm{aff}} = 2.5, \omega_{\mathrm{inv}} = 1.7$ | 98.1 / +11.6 | 98.8 / +1.9 | 84.0 / +5.4 | 0.189 | 0.123 |
| | | $\omega_{\mathrm{aff}} = 2.5, \omega_{\mathrm{inv}} = 2.0$ | 97.5 / +11.0 | 99.3 / +2.4 | 87.0 / +8.4 | 0.209 | 0.131 |
| | | $\omega_{\mathrm{aff}} = 2.5, \omega_{\mathrm{inv}} = 2.5$ | 96.3 / +9.8 | 99.5 / +2.6 | 88.7 / +10.1 | 0.236 | 0.143 |
| do(Male) | CFG (Ho & Salimans, 2022) | $\omega = 1.0$ | 86.6 / +0.0 | 91.8 / +0.0 | 79.8 / +0.0 | 0.115 | 0.079 |
| | | $\omega = 2.5$ | 97.6 / +11.0 | 98.4 / +6.6 | 87.5 / +7.7 | 0.238 | 0.156 |
| | FCFG (Ours) | $\omega_{\mathrm{aff}} = 2.5, \omega_{\mathrm{inv}} = 1.0$ | 83.4 / -3.2 | 99.4 / +7.6 | 68.9 / -10.9 | 0.173 | 0.122 |
| | | $\omega_{\mathrm{aff}} = 2.5, \omega_{\mathrm{inv}} = 1.2$ | 87.4 / +0.8 | 99.4 / +7.6 | 71.1 / -8.7 | 0.171 | 0.118 |
| | | $\omega_{\mathrm{aff}} = 2.5, \omega_{\mathrm{inv}} = 1.5$ | 92.0 / +5.4 | 99.3 / +7.5 | 74.0 / -5.8 | 0.182 | 0.119 |
| | | $\omega_{\mathrm{aff}} = 2.5, \omega_{\mathrm{inv}} = 1.7$ | 93.5 / +6.9 | 98.9 / +7.1 | 76.8 / -3.0 | 0.189 | 0.123 |
| | | $\omega_{\mathrm{aff}} = 2.5, \omega_{\mathrm{inv}} = 2.0$ | 95.3 / +8.7 | 98.7 / +6.9 | 80.2 / +0.4 | 0.207 | 0.135 |
| | | $\omega_{\mathrm{aff}} = 2.5, \omega_{\mathrm{inv}} = 2.5$ | 97.2 / +10.6 | 97.7 / +5.9 | 87.5 / +7.7 | 0.242 | 0.158 |
| do(Young) | CFG (Ho & Salimans, 2022) | $\omega = 1.0$ | 87.5 / +0.0 | 95.7 / +0.0 | 62.3 / +0.0 | 0.115 | 0.085 |
| | | $\omega = 2.5$ | 98.3 / +10.8 | 99.8 / +4.1 | 73.5 / +11.2 | 0.234 | 0.146 |
| | FCFG (Ours) | $\omega_{\mathrm{aff}} = 2.5, \omega_{\mathrm{inv}} = 1.0$ | 83.4 / -4.1 | 85.9 / -9.8 | 85.1 / +22.8 | 0.169 | 0.127 |
| | | $\omega_{\mathrm{aff}} = 2.5, \omega_{\mathrm{inv}} = 1.2$ | 88.5 / +1.0 | 91.9 / -3.8 | 81.8 / +19.5 | 0.172 | 0.125 |
| | | $\omega_{\mathrm{aff}} = 2.5, \omega_{\mathrm{inv}} = 1.5$ | 92.4 / +4.9 | 96.4 / +0.7 | 78.5 / +16.2 | 0.187 | 0.127 |
| | | $\omega_{\mathrm{aff}} = 2.5, \omega_{\mathrm{inv}} = 1.7$ | 94.3 / +6.8 | 97.8 / +2.1 | 75.3 / +13.0 | 0.199 | 0.133 |
| | | $\omega_{\mathrm{aff}} = 2.5, \omega_{\mathrm{inv}} = 2.0$ | 97.1 / +9.6 | 99.0 / +3.3 | 73.3 / +11.0 | 0.215 | 0.139 |
| | | $\omega_{\mathrm{aff}} = 2.5, \omega_{\mathrm{inv}} = 2.5$ | 99.3 / +11.8 | 99.7 / +4.0 | 68.5 / +6.2 | 0.238 | 0.147 |

## F.3. Extra qualitative results for CelebA-HQ

*Figure A.2.* **Additional qualitative results for do(Smiling) on CelebA-HQ.** Each row shows the original image followed by counterfactuals generated with global CFG ($\omega$) and FCFG ($\omega_{\text{aff}}, \omega_{\text{inv}}$). FCFG better preserves *invariant* attributes and identity while effectively reflecting the intervention.

*Figure A.3.* **Additional qualitative results for do(Male) on CelebA-HQ.** Each row shows the original image followed by counterfactuals generated with global CFG ($\omega$) and FCFG ($\omega_{\text{aff}}, \omega_{\text{inv}}$). FCFG better preserves *invariant* attributes and identity while effectively reflecting the intervention.

*Figure A.4.* **Additional qualitative results for do(Young) on CelebA-HQ.** Each row shows the original image followed by counterfactuals generated with global CFG ($\omega$) and FCFG ($\omega_{\text{aff}}, \omega_{\text{inv}}$). FCFG better preserves *invariant* attributes and identity while effectively reflecting the intervention.

| Smiling: no
Male: no
Young: no | $\omega = 2.5$
do(Smiling=yes) | $\omega = 2.5$
Reversed | $\omega_{\text{aff}} = 2.5, \omega_{\text{inv}} = 1.0$
do(Smiling=yes) | $\omega_{\text{aff}} = 2.5, \omega_{\text{inv}} = 1.0$
Reversed |

| Smiling: no
Male: no
Young: yes | $\omega = 2.5$
do(Smiling=yes) | $\omega = 2.5$
Reversed | $\omega_{\text{aff}} = 2.5, \omega_{\text{inv}} = 1.0$
do(Smiling=yes) | $\omega_{\text{aff}} = 2.5, \omega_{\text{inv}} = 1.0$
Reversed |

| Smiling: no
Male: yes
Young: yes | $\omega = 2.5$
do(Smiling=yes) | $\omega = 2.5$
Reversed | $\omega_{\text{aff}} = 2.5, \omega_{\text{inv}} = 1.0$
do(Smiling=yes) | $\omega_{\text{aff}} = 2.5, \omega_{\text{inv}} = 1.0$
Reversed |

| Smiling: no
Male: no
Young: yes | $\omega = 2.5$
do(Smiling=yes) | $\omega = 2.5$
Reversed | $\omega_{\text{aff}} = 2.5, \omega_{\text{inv}} = 1.0$
do(Smiling=yes) | $\omega_{\text{aff}} = 2.5, \omega_{\text{inv}} = 1.0$
Reversed |

| Smiling: yes
Male: yes
Young: yes | $\omega = 2.5$
do(Smiling=no) | $\omega = 2.5$
Reversed | $\omega_{\text{aff}} = 2.5, \omega_{\text{inv}} = 1.0$
do(Smiling=no) | $\omega_{\text{aff}} = 2.5, \omega_{\text{inv}} = 1.0$
Reversed |

*Figure A.5.* **Reversibility analysis for** `do(Similing)` **on CelebA-HQ.** Each row shows the original image, followed by counterfactuals generated using global CFG ($\omega$) and our proposed FCFG ($\omega_{\text{int}}, \omega_{\text{inv}}$), along with their respective reversed generations. FCFG more faithfully preserves non-intervened attributes, resulting in visually and semantically more consistent reversals. This highlights the benefit of FCFG in enhancing both targeted editability and reversibility.

| Smiling: no
Male: yes
Young: yes | $\omega = 2.5$
do(Male=no) | $\omega = 2.5$
Reversed | $\omega_{aff} = 2.5, \omega_{inv} = 1.0$
do(Male=no) | $\omega_{aff} = 2.5, \omega_{inv} = 1.0$
Reversed |
|---|---|---|---|---|

| Smiling: no
Male: yes
Young: no | $\omega = 2.5$
do(Male=no) | $\omega = 2.5$
Reversed | $\omega_{aff} = 2.5, \omega_{inv} = 1.0$
do(Male=no) | $\omega_{aff} = 2.5, \omega_{inv} = 1.0$
Reversed |
|---|---|---|---|---|

| Smiling: yes
Male: yes
Young: yes | $\omega = 2.5$
do(Male=no) | $\omega = 2.5$
Reversed | $\omega_{aff} = 2.5, \omega_{inv} = 1.0$
do(Male=no) | $\omega_{aff} = 2.5, \omega_{inv} = 1.0$
Reversed |
|---|---|---|---|---|

| Smiling: no
Male: yes
Young: yes | $\omega = 2.5$
do(Male=no) | $\omega = 2.5$
Reversed | $\omega_{aff} = 2.5, \omega_{inv} = 1.0$
do(Male=no) | $\omega_{aff} = 2.5, \omega_{inv} = 1.0$
Reversed |
|---|---|---|---|---|

| Smiling: no
Male: yes
Young: no | $\omega = 2.5$
do(Male=no) | $\omega = 2.5$
Reversed | $\omega_{aff} = 2.5, \omega_{inv} = 1.0$
do(Male=no) | $\omega_{aff} = 2.5, \omega_{inv} = 1.0$
Reversed |
|---|---|---|---|---|

*Figure A.6.* **Reversibility analysis for do(Male) on CelebA-HQ.** Each row shows the original image, followed by counterfactuals generated using global CFG ($\omega$) and our proposed FCFG ($\omega_{int}, \omega_{inv}$), along with their respective reversed generations. FCFG more faithfully preserves non-intervened attributes, resulting in visually and semantically more consistent reversals. This highlights the benefit of FCFG in enhancing both targeted editability and reversibility.

| Smiling: no
Male: yes
Young: yes | $\omega = 2.5$
do(Young=no) | $\omega = 2.5$
Reversed | $\omega_{aff} = 2.5$, $\omega_{inv} = 1.0$
do(Young=no) | $\omega_{aff} = 2.5$, $\omega_{inv} = 1.0$
Reversed |

| Smiling: no
Male: yes
Young: no | $\omega = 2.5$
do(Young=yes) | $\omega = 2.5$
Reversed | $\omega_{aff} = 2.5$, $\omega_{inv} = 1.0$
do(Young=yes) | $\omega_{aff} = 2.5$, $\omega_{inv} = 1.0$
Reversed |

| Smiling: yes
Male: yes
Young: yes | $\omega = 2.5$
do(Young=no) | $\omega = 2.5$
Reversed | $\omega_{aff} = 2.5$, $\omega_{inv} = 1.0$
do(Young=no) | $\omega_{aff} = 2.5$, $\omega_{inv} = 1.0$
Reversed |

| Smiling: yes
Male: yes
Young: no | $\omega = 2.5$
do(Young=yes) | $\omega = 2.5$
Reversed | $\omega_{aff} = 2.5$, $\omega_{inv} = 1.0$
do(Young=yes) | $\omega_{aff} = 2.5$, $\omega_{inv} = 1.0$
Reversed |

| Smiling: yes
Male: yes
Young: yes | $\omega = 2.5$
do(Young=no) | $\omega = 2.5$
Reversed | $\omega_{aff} = 2.5$, $\omega_{inv} = 1.0$
do(Young=no) | $\omega_{aff} = 2.5$, $\omega_{inv} = 1.0$
Reversed |

*Figure A.7.* **Reversibility analysis for `do(Young)` on CelebA-HQ.** Each row shows the original image, followed by counterfactuals generated using global CFG ($\omega$) and our proposed FCFG ($\omega_{int}, \omega_{inv}$), along with their respective reversed generations. FCFG more faithfully preserves non-intervened attributes, resulting in visually and semantically more consistent reversals. This highlights the benefit of FCFG in enhancing both targeted editability and reversibility.

*Figure A.8.* **Effect of $\omega_{\text{inv}}$ on CelebA-HQ counterfactuals.** Each row shows the original image followed by counterfactuals generated using global CFG ($\omega$=2.5) and our proposed FCFG with fixed intervention guidance ($\omega_{\text{aff}}$=2.5) and varying invariant guidance $\omega_{\text{inv}} \in \{1.0, 1.5, 2.0, 2.5\}$. As $\omega_{\text{inv}}$ increases, amplification of invariant attributes becomes more pronounced, and at $\omega_{\text{inv}}$=2.5, FCFG effectively reproduces the same over-editing behavior as global CFG. This shows that $\omega_{\text{inv}}$ modulates the degree of guidance applied to invariant attributes and should be carefully calibrated to maintain identity and disentanglement.

# G. EMEBD

## G.1. Dataset details

We use the EMory BrEast imaging Dataset (EMBED) (Jeong et al., 2023) for our experiments. Schueppert et al. (2024) manually labeled 22,012 images with circular markers and trained a classifier on this subset, which was then applied to the full dataset to infer circle annotations. We adopt this preprocessing pipeline and extract the `circle` attribute from their predictions. To define the `density` label, we binarize the original four-category breast density annotations by grouping categories A and B as `low` density, and categories C and D as `high` density. While the full dataset comprises 151,948 training, 7,156 validation, and 43,669 test samples, we use only 1,000 test samples in this work due to the high computational cost of diffusion models. As shown in Fig. A.9, the Pearson correlation matrix reveals that `density` and `circle` are nearly uncorrelated, supporting our assumption of their independence.

*Figure A.9.* **Pearson correlation matrix of EMBED attributes:** `density` and `circle`. The correlation between these two variables is negligible ($\rho = -0.04$), suggesting that they can be reasonably treated as independent for the purposes of our analysis.

## G.2. Extra quantitative results for EMBED

*Table A.9.* EMBED: Effectiveness (ROC-AUC ↑) and reversibility (MAE, LPIPS ↓) when varying $\omega_{\text{aff}}$. Compared to global CFG (i.e., $\omega$), FCFG achieves strong intervention effectiveness on the intervened variable while mitigating amplification on invariant variables. For higher $\omega_{\text{aff}}$, we set $\omega_{\text{inv}}=1.2$ to prevent degradation of invariant attributes. While reversibility deteriorates with increasing $\omega_{\text{aff}}$, FCFG consistently maintains better reversibility than global CFG with $\omega=\omega_{\text{aff}}$.

| do(key) | Guidance type | Guidance configuration | Density AUC/Δ | Circle AUC/Δ | MAE | LPIPS |
|---|---|---|---|---|---|---|
| do(density) | CFG (Ho & Salimans, 2022) | $\omega = 1.0$ | 63.4 / +0.0 | 92.9 / +0.0 | 0.027 | 0.033 |
| | | $\omega = 1.2$ | 70.5 / +7.1 | 94.5 / +1.6 | 0.033 | 0.038 |
| | | $\omega = 1.5$ | 79.0 / +15.6 | 95.9 / +3.0 | 0.035 | 0.047 |
| | | $\omega = 1.7$ | 84.3 / +20.9 | 96.7 / +3.8 | 0.032 | 0.055 |
| | | $\omega = 2.0$ | 89.6 / +26.2 | 97.5 / +4.6 | 0.034 | 0.064 |
| | | $\omega = 2.5$ | 95.2 / +31.8 | 97.7 / +4.8 | 0.042 | 0.076 |
| | | $\omega = 3.0$ | 97.8 / +34.4 | 98.2 / +5.3 | 0.045 | 0.086 |
| | FCFG (Ours) | $\omega_{\text{aff}} = 1.2, \omega_{\text{inv}} = 1.0$ | 73.1 / +9.7 | 92.8 / -0.1 | 0.028 | 0.038 |
| | | $\omega_{\text{aff}} = 1.5, \omega_{\text{inv}} = 1.0$ | 81.6 / +18.2 | 92.2 / -0.7 | 0.029 | 0.043 |
| | | $\omega_{\text{aff}} = 1.7, \omega_{\text{inv}} = 1.0$ | 86.2 / +22.8 | 91.6 / -1.3 | 0.031 | 0.048 |
| | | $\omega_{\text{aff}} = 2.0, \omega_{\text{inv}} = 1.0$ | 91.6 / +28.2 | 90.7 / -2.2 | 0.032 | 0.053 |
| | | $\omega_{\text{aff}} = 2.5, \omega_{\text{inv}} = 1.2$ | 96.6 / +33.2 | 92.2 / -0.7 | 0.036 | 0.064 |
| | | $\omega_{\text{aff}} = 3.0, \omega_{\text{inv}} = 1.2$ | 98.6 / +35.2 | 91.6 / -1.3 | 0.038 | 0.071 |
| do(circle) | CFG (Ho & Salimans, 2022) | $\omega = 1.0$ | 92.6 / +0.0 | 90.6 / +0.0 | 0.023 | 0.026 |
| | | $\omega = 1.2$ | 94.7 / +2.1 | 92.1 / +1.5 | 0.029 | 0.024 |
| | | $\omega = 1.5$ | 96.8 / +4.2 | 93.9 / +3.3 | 0.030 | 0.027 |
| | | $\omega = 1.7$ | 97.9 / +5.3 | 95.2 / +4.6 | 0.027 | 0.035 |
| | | $\omega = 2.0$ | 98.8 / +6.2 | 96.4 / +5.8 | 0.030 | 0.040 |
| | | $\omega = 2.5$ | 99.7 / +7.1 | 97.8 / +7.2 | 0.038 | 0.043 |
| | | $\omega = 3.0$ | 99.9 / +7.3 | 98.4 / +7.8 | 0.042 | 0.051 |
| | FCFG (Ours) | $\omega_{\text{aff}} = 1.2, \omega_{\text{inv}} = 1.0$ | 93.3 / +0.7 | 92.6 / +2.0 | 0.024 | 0.028 |
| | | $\omega_{\text{aff}} = 1.5, \omega_{\text{inv}} = 1.0$ | 93.2 / +0.6 | 94.4 / +3.8 | 0.025 | 0.030 |
| | | $\omega_{\text{aff}} = 1.7, \omega_{\text{inv}} = 1.0$ | 93.2 / +0.6 | 95.7 / +5.1 | 0.025 | 0.032 |
| | | $\omega_{\text{aff}} = 2.0, \omega_{\text{inv}} = 1.0$ | 92.9 / +0.3 | 97.2 / +6.6 | 0.026 | 0.034 |
| | | $\omega_{\text{aff}} = 2.5, \omega_{\text{inv}} = 1.2$ | 94.5 / +1.9 | 98.5 / +7.9 | 0.027 | 0.038 |
| | | $\omega_{\text{aff}} = 3.0, \omega_{\text{inv}} = 1.2$ | 94.0 / +1.4 | 98.9 / +8.3 | 0.029 | 0.042 |

*Table A.10.* EMBED: Effectiveness (ROC-AUC ↑) and reversibility (MAE, LPIPS ↓) when varying $\omega_{\text{inv}}$. Increasing $\omega_{\text{inv}}$ consistently increases effectiveness on invariant variables, while degrading intervention effectiveness. When $\omega_{\text{inv}} = 2.5$, the amplification on invariant attributes becomes comparable to the global CFG setting with $\omega = 2.5$.

| do(key) | Guidance type | Guidance configuration | Density AUC/Δ | Circle AUC/Δ | MAE | LPIPS |
|---|---|---|---|---|---|---|
| do(density) | CFG (Ho & Salimans, 2022) | $\omega = 1.0$ | 63.4 / +0.0 | 92.9 / +0.0 | 0.027 | 0.033 |
| | | $\omega = 2.5$ | 95.2 / +31.8 | 97.7 / +4.8 | 0.042 | 0.076 |
| | FCFG (Ours) | $\omega_{\text{aff}} = 2.5, \omega_{\text{inv}} = 1.0$ | 96.7 / +33.3 | 89.5 / -3.4 | 0.035 | 0.063 |
| | | $\omega_{\text{aff}} = 2.5, \omega_{\text{inv}} = 1.2$ | 96.6 / +33.2 | 92.2 / -0.7 | 0.036 | 0.064 |
| | | $\omega_{\text{aff}} = 2.5, \omega_{\text{inv}} = 1.5$ | 96.6 / +33.2 | 94.6 / +1.7 | 0.036 | 0.067 |
| | | $\omega_{\text{aff}} = 2.5, \omega_{\text{inv}} = 1.7$ | 96.6 / +33.2 | 95.7 / +2.8 | 0.037 | 0.070 |
| | | $\omega_{\text{aff}} = 2.5, \omega_{\text{inv}} = 2.0$ | 96.5 / +33.1 | 96.6 / +3.7 | 0.038 | 0.073 |
| | | $\omega_{\text{aff}} = 2.5, \omega_{\text{inv}} = 2.5$ | 96.4 / +33.0 | 97.6 / +4.7 | 0.039 | 0.080 |
| do(circle) | CFG (Ho & Salimans, 2022) | $\omega = 1.0$ | 92.6 / +0.0 | 90.6 / +0.0 | 0.023 | 0.026 |
| | | $\omega = 2.5$ | 99.7 / +7.1 | 97.8 / +7.2 | 0.038 | 0.043 |
| | FCFG (Ours) | $\omega_{\text{aff}} = 2.5, \omega_{\text{inv}} = 1.0$ | 92.2 / -0.4 | 98.5 / +7.9 | 0.028 | 0.039 |
| | | $\omega_{\text{aff}} = 2.5, \omega_{\text{inv}} = 1.2$ | 94.5 / +1.9 | 98.5 / +7.9 | 0.027 | 0.038 |
| | | $\omega_{\text{aff}} = 2.5, \omega_{\text{inv}} = 1.5$ | 97.2 / +4.6 | 98.3 / +7.7 | 0.028 | 0.039 |
| | | $\omega_{\text{aff}} = 2.5, \omega_{\text{inv}} = 1.7$ | 98.1 / +5.5 | 98.2 / +7.6 | 0.029 | 0.040 |
| | | $\omega_{\text{aff}} = 2.5, \omega_{\text{inv}} = 2.0$ | 99.0 / +6.4 | 98.2 / +7.6 | 0.031 | 0.043 |
| | | $\omega_{\text{aff}} = 2.5, \omega_{\text{inv}} = 2.5$ | 99.8 / +7.2 | 98.0 / +7.4 | 0.035 | 0.050 |

## G.3. Extra qualitative results for EMBED

*Figure A.10.* **Additional qualitative results for `do(density)` on EMBED.** Each row shows the original image followed by counterfactuals generated with global CFG ($\omega$) and FCFG ($\omega_{\text{aff}}, \omega_{\text{inv}}$). FCFG better preserves *invariant* attributes and identity while effectively reflecting the intervention. Notably, under global CFG, increasing $\omega$ leads to spurious changes in circle count, whereas FCFG mitigates such amplification.

*Figure A.11.* **Additional qualitative results for `do(circle)` on EMBED.** Each row shows the original image followed by counterfactuals generated with global CFG ($\omega$) and FCFG ($\omega_{\text{aff}}, \omega_{\text{inv}}$) and the difference map (CF-input). FCFG better preserves *invariant* attributes and identity while effectively reflecting the intervention. Notably, under global CFG, increasing $\omega$ leads to spurious changes in density, whereas FCFG mitigates such amplification.

*Figure A.12.* **Reversibility analysis for do(`density`) on EMBED.** Each row shows the original image, the counterfactual generated using global CFG ($\omega$) or FCFG ($\omega_{int}$, $\omega_{inv}$), their corresponding reversed generations, and the associated difference maps (counterfactual - input, and reversed - input). FCFG more faithfully preserves non-intervened attributes and leads to smaller residuals in the difference maps, indicating better identity preservation.

*Figure A.13.* **Reversibility analysis for do(`circle`) on EMBED.** Each row shows the original image, the counterfactual generated using global CFG ($\omega$) or FCFG ($\omega_{int}$, $\omega_{inv}$), their corresponding reversed generations, and the associated difference maps (counterfactual - input, and reversed - input). FCFG more faithfully preserves non-intervened attributes and leads to smaller residuals in the difference maps, indicating better identity preservation.

# H. MIMIC

## H.1. Dataset details

We use the MIMIC-CXR dataset (Johnson et al., 2019) in our experiments. Following the dataset splits and filtering protocols of (Ribeiro et al., 2023; Glocker et al., 2023), we focus on the binary disease label for pleural effusion. We adopt the same causal graph (DAG) as proposed in Ribeiro et al. (2023), in which `age` is modeled as a parent of `finding`. While we include `age` as part of the conditioning variables, we do not intervene on it. Instead, our primary goal is to study amplification of unintervened variables caused by CFG. For this purpose, we focus on `race`, `sex`, and `finding`, which we assume to be mutually independent. Fig. A.14 shows the Pearson correlation matrix of these three attributes, where all pairwise correlations are small (e.g., $\rho=0.12$ between `race` and `sex`, and $\rho=-0.15$ between `race` and `finding`), supporting the validity of the independence assumption in our counterfactual modeling.

*Figure A.14.* **Pearson correlation matrix of MIMIC attributes:** `race`, `sex`, and `finding`. All pairwise correlations are low (e.g., $\rho=0.12$ between `race` and `sex`, and $\rho=-0.15$ between `race` and `finding`), suggesting that these variables can be reasonably treated as independent for the purposes of our counterfactual analysis.

## H.2. Extra quantitative results for MIMIC-CXR

*Table A.11.* MIMIC: Effectiveness (ROC-AUC ↑) and reversibility (MAE, LPIPS ↓) when varying $\omega_{\mathrm{aff}}$. Compared to global CFG, FCFG achieves strong intervention effectiveness on the intervened variable while mitigating amplification on invariant variables. For higher $\omega_{\mathrm{aff}}$, we fix $\omega_{\mathrm{inv}}=1.2$ to prevent degradation of invariant attributes.

| do(key) | Guidance type | Guidance configuration | Sex AUC/Δ | Race AUC/Δ | Finding AUC/Δ | MAE | LPIPS |
|---|---|---|---|---|---|---|---|
| do(sex) | CFG (Ho & Salimans, 2022) | $\omega = 1.0$ | 92.4 / +0.0 | 75.6 / +0.0 | 88.8 / +0.0 | 0.146 | 0.202 |
| | | $\omega = 1.2$ | 95.2 / +2.8 | 79.3 / +3.7 | 92.8 / +4.0 | 0.151 | 0.206 |
| | | $\omega = 1.5$ | 97.7 / +5.3 | 82.4 / +6.8 | 95.7 / +6.9 | 0.171 | 0.226 |
| | | $\omega = 1.7$ | 98.5 / +6.1 | 84.7 / +9.1 | 97.0 / +8.2 | 0.186 | 0.239 |
| | | $\omega = 2.0$ | 99.3 / +6.9 | 87.4 / +11.8 | 98.0 / +9.2 | 0.207 | 0.258 |
| | | $\omega = 2.5$ | 99.8 / +7.4 | 90.5 / +14.9 | 99.0 / +10.2 | 0.239 | 0.284 |
| | | $\omega = 3.0$ | 99.9 / +7.5 | 92.9 / +17.3 | 99.5 / +10.7 | 0.266 | 0.305 |
| | FCFG (Ours) | $\omega_{\mathrm{aff}} = 1.2, \omega_{\mathrm{inv}} = 1.0$ | 96.4 / +4.0 | 74.6 / -1.0 | 89.1 / +0.3 | 0.158 | 0.217 |
| | | $\omega_{\mathrm{aff}} = 1.5, \omega_{\mathrm{inv}} = 1.0$ | 98.4 / +6.0 | 74.1 / -1.5 | 88.5 / -0.3 | 0.167 | 0.227 |
| | | $\omega_{\mathrm{aff}} = 1.7, \omega_{\mathrm{inv}} = 1.2$ | 99.2 / +6.8 | 76.9 / +1.3 | 91.5 / +2.7 | 0.174 | 0.233 |
| | | $\omega_{\mathrm{aff}} = 2.0, \omega_{\mathrm{inv}} = 1.2$ | 99.5 / +7.1 | 75.8 / +0.2 | 90.9 / +2.1 | 0.183 | 0.243 |
| | | $\omega_{\mathrm{aff}} = 2.5, \omega_{\mathrm{inv}} = 1.2$ | 99.9 / +7.5 | 74.9 / -0.7 | 90.1 / +1.3 | 0.199 | 0.260 |
| | | $\omega_{\mathrm{aff}} = 3.0, \omega_{\mathrm{inv}} = 1.2$ | 100.0 / +7.6 | 74.5 / -1.1 | 89.4 / +0.6 | 0.216 | 0.276 |
| do(race) | CFG (Ho & Salimans, 2022) | $\omega = 1.0$ | 95.1 / +0.0 | 65.4 / +0.0 | 90.4 / +0.0 | 0.135 | 0.191 |
| | | $\omega = 1.2$ | 97.6 / +2.5 | 69.9 / +4.5 | 93.3 / +2.9 | 0.135 | 0.190 |
| | | $\omega = 1.5$ | 98.9 / +3.8 | 73.9 / +8.5 | 96.2 / +5.8 | 0.155 | 0.209 |
| | | $\omega = 1.7$ | 99.3 / +4.2 | 76.3 / +10.9 | 97.5 / +7.1 | 0.171 | 0.223 |
| | | $\omega = 2.0$ | 99.6 / +4.5 | 80.5 / +15.1 | 98.2 / +7.8 | 0.193 | 0.242 |
| | | $\omega = 2.5$ | 99.7 / +4.6 | 86.1 / +20.7 | 99.2 / +8.8 | 0.229 | 0.271 |
| | | $\omega = 3.0$ | 99.8 / +4.7 | 90.1 / +24.7 | 99.4 / +9.0 | 0.256 | 0.292 |
| | FCFG (Ours) | $\omega_{\mathrm{aff}} = 1.2, \omega_{\mathrm{inv}} = 1.0$ | 95.4 / +0.3 | 69.6 / +4.2 | 90.3 / -0.1 | 0.141 | 0.198 |
| | | $\omega_{\mathrm{aff}} = 1.5, \omega_{\mathrm{inv}} = 1.0$ | 94.8 / -0.3 | 75.5 / +10.1 | 90.0 / -0.4 | 0.147 | 0.203 |
| | | $\omega_{\mathrm{aff}} = 1.7, \omega_{\mathrm{inv}} = 1.2$ | 97.2 / +2.1 | 78.3 / +12.9 | 92.4 / +2.0 | 0.153 | 0.208 |
| | | $\omega_{\mathrm{aff}} = 2.0, \omega_{\mathrm{inv}} = 1.2$ | 96.4 / +1.3 | 82.8 / +17.4 | 92.0 / +1.6 | 0.160 | 0.215 |
| | | $\omega_{\mathrm{aff}} = 2.5, \omega_{\mathrm{inv}} = 1.2$ | 95.6 / +0.5 | 89.0 / +23.6 | 91.7 / +1.3 | 0.178 | 0.231 |
| | | $\omega_{\mathrm{aff}} = 3.0, \omega_{\mathrm{inv}} = 1.2$ | 94.0 / -1.1 | 92.7 / +27.3 | 91.7 / +1.3 | 0.197 | 0.249 |
| do(finding) | CFG (Ho & Salimans, 2022) | $\omega = 1.0$ | 94.6 / +0.0 | 78.3 / +0.0 | 80.8 / +0.0 | 0.134 | 0.193 |
| | | $\omega = 1.2$ | 97.2 / +2.6 | 81.2 / +2.9 | 85.7 / +4.9 | 0.136 | 0.194 |
| | | $\omega = 1.5$ | 98.9 / +4.3 | 83.8 / +5.5 | 90.6 / +9.8 | 0.153 | 0.210 |
| | | $\omega = 1.7$ | 99.5 / +4.9 | 85.7 / +7.4 | 92.9 / +12.1 | 0.165 | 0.222 |
| | | $\omega = 2.0$ | 99.7 / +5.1 | 88.5 / +10.2 | 95.0 / +14.2 | 0.184 | 0.239 |
| | | $\omega = 2.5$ | 99.8 / +5.2 | 91.9 / +13.6 | 97.7 / +16.9 | 0.215 | 0.264 |
| | | $\omega = 3.0$ | 99.9 / +5.3 | 93.8 / +15.5 | 98.6 / +17.8 | 0.244 | 0.287 |
| | FCFG (Ours) | $\omega_{\mathrm{aff}} = 1.2, \omega_{\mathrm{inv}} = 1.0$ | 95.1 / +0.5 | 78.1 / -0.2 | 85.6 / +4.8 | 0.142 | 0.202 |
| | | $\omega_{\mathrm{aff}} = 1.5, \omega_{\mathrm{inv}} = 1.0$ | 94.9 / +0.3 | 77.8 / -0.5 | 92.0 / +11.2 | 0.141 | 0.201 |
| | | $\omega_{\mathrm{aff}} = 1.7, \omega_{\mathrm{inv}} = 1.2$ | 97.1 / +2.5 | 80.6 / +2.3 | 93.5 / +12.7 | 0.150 | 0.209 |
| | | $\omega_{\mathrm{aff}} = 2.0, \omega_{\mathrm{inv}} = 1.2$ | 96.9 / +2.3 | 80.2 / +1.9 | 96.6 / +15.8 | 0.149 | 0.209 |
| | | $\omega_{\mathrm{aff}} = 2.5, \omega_{\mathrm{inv}} = 1.2$ | 96.4 / +1.8 | 80.1 / +1.8 | 98.8 / +18.0 | 0.151 | 0.212 |
| | | $\omega_{\mathrm{aff}} = 3.0, \omega_{\mathrm{inv}} = 1.2$ | 95.2 / +0.6 | 79.3 / +1.0 | 99.6 / +18.8 | 0.154 | 0.216 |

*Table A.12.* **MIMIC: Effectiveness (ROC-AUC ↑) and reversibility (MAE, LPIPS ↓) when varying $\omega_{\mathrm{inv}}$.** Increasing $\omega_{\mathrm{inv}}$ increases effectiveness on invariant variables while degrading intervention effectiveness. When $\omega_{\mathrm{inv}} = 2.5$, amplification on invariant attributes becomes comparable to the global CFG setting with $\omega = 2.5$.

| do(key) | Guidance type | Guidance configuration | Sex AUC/Δ | Race AUC/Δ | Finding AUC/Δ | MAE | LPIPS |
|---|---|---|---|---|---|---|---|
| do(sex) | CFG (Ho & Salimans, 2022) | $\omega = 1.0$ | 92.4 / +0.0 | 75.6 / +0.0 | 88.8 / +0.0 | 0.146 | 0.202 |
| | | $\omega = 2.5$ | 99.8 / +7.4 | 90.5 / +14.9 | 99.0 / +10.2 | 0.239 | 0.284 |
| | FCFG (Ours) | $\omega_{\mathrm{aff}} = 2.5, \omega_{\mathrm{inv}} = 1.0$ | 99.9 / +7.5 | 71.3 / -4.3 | 86.2 / -2.6 | 0.200 | 0.261 |
| | | $\omega_{\mathrm{aff}} = 2.5, \omega_{\mathrm{inv}} = 1.2$ | 99.9 / +7.5 | 74.9 / -0.7 | 90.1 / +1.3 | 0.199 | 0.260 |
| | | $\omega_{\mathrm{aff}} = 2.5, \omega_{\mathrm{inv}} = 1.5$ | 99.8 / +7.4 | 80.1 / +4.5 | 94.2 / +5.4 | 0.207 | 0.264 |
| | | $\omega_{\mathrm{aff}} = 2.5, \omega_{\mathrm{inv}} = 1.7$ | 99.7 / +7.3 | 83.2 / +7.6 | 95.9 / +7.1 | 0.214 | 0.269 |
| | | $\omega_{\mathrm{aff}} = 2.5, \omega_{\mathrm{inv}} = 2.0$ | 99.7 / +7.3 | 86.7 / +11.1 | 97.5 / +8.7 | 0.227 | 0.278 |
| | | $\omega_{\mathrm{aff}} = 2.5, \omega_{\mathrm{inv}} = 2.5$ | 99.6 / +7.2 | 90.3 / +14.7 | 98.9 / +10.1 | 0.249 | 0.293 |
| do(race) | CFG (Ho & Salimans, 2022) | $\omega = 1.0$ | 95.1 / +0.0 | 65.4 / +0.0 | 90.4 / +0.0 | 0.135 | 0.191 |
| | | $\omega = 2.5$ | 99.7 / +4.6 | 86.1 / +20.7 | 99.2 / +8.8 | 0.229 | 0.271 |
| | FCFG (Ours) | $\omega_{\mathrm{aff}} = 2.5, \omega_{\mathrm{inv}} = 1.0$ | 91.3 / -3.8 | 89.5 / +24.1 | 88.4 / -2.0 | 0.181 | 0.237 |
| | | $\omega_{\mathrm{aff}} = 2.5, \omega_{\mathrm{inv}} = 1.2$ | 95.6 / +0.5 | 89.0 / +23.6 | 91.7 / +1.3 | 0.178 | 0.231 |
| | | $\omega_{\mathrm{aff}} = 2.5, \omega_{\mathrm{inv}} = 1.5$ | 98.5 / +3.4 | 87.9 / +22.5 | 95.2 / +4.8 | 0.185 | 0.236 |
| | | $\omega_{\mathrm{aff}} = 2.5, \omega_{\mathrm{inv}} = 1.7$ | 99.2 / +4.1 | 87.4 / +22.0 | 96.8 / +6.4 | 0.191 | 0.242 |
| | | $\omega_{\mathrm{aff}} = 2.5, \omega_{\mathrm{inv}} = 2.0$ | 99.6 / +4.5 | 86.3 / +20.9 | 98.0 / +7.6 | 0.205 | 0.253 |
| | | $\omega_{\mathrm{aff}} = 2.5, \omega_{\mathrm{inv}} = 2.5$ | 99.8 / +4.7 | 85.6 / +20.2 | 99.1 / +8.7 | 0.231 | 0.274 |
| do(finding) | CFG (Ho & Salimans, 2022) | $\omega = 1.0$ | 94.6 / +0.0 | 78.3 / +0.0 | 80.8 / +0.0 | 0.134 | 0.193 |
| | | $\omega = 2.5$ | 99.8 / +5.2 | 91.9 / +13.6 | 97.7 / +16.9 | 0.215 | 0.264 |
| | FCFG (Ours) | $\omega_{\mathrm{aff}} = 2.5, \omega_{\mathrm{inv}} = 1.0$ | 93.0 / -1.6 | 77.0 / -1.3 | 99.0 / +18.2 | 0.143 | 0.206 |
| | | $\omega_{\mathrm{aff}} = 2.5, \omega_{\mathrm{inv}} = 1.2$ | 96.4 / +1.8 | 80.1 / +1.8 | 98.8 / +18.0 | 0.151 | 0.212 |
| | | $\omega_{\mathrm{aff}} = 2.5, \omega_{\mathrm{inv}} = 1.5$ | 98.5 / +3.9 | 84.1 / +5.8 | 98.3 / +17.5 | 0.166 | 0.224 |
| | | $\omega_{\mathrm{aff}} = 2.5, \omega_{\mathrm{inv}} = 1.7$ | 99.2 / +4.6 | 86.2 / +7.9 | 97.8 / +17.0 | 0.180 | 0.236 |
| | | $\omega_{\mathrm{aff}} = 2.5, \omega_{\mathrm{inv}} = 2.0$ | 99.6 / +5.0 | 89.4 / +11.1 | 97.1 / +16.3 | 0.202 | 0.254 |
| | | $\omega_{\mathrm{aff}} = 2.5, \omega_{\mathrm{inv}} = 2.5$ | 99.9 / +5.3 | 92.6 / +14.3 | 95.8 / +15.0 | 0.239 | 0.282 |

## H.3. Extra visual results for MIMIC

*Figure A.15.* **Additional qualitative results for `do(finding)` on MIMIC.** Each row shows the original image followed by counterfactuals generated using global CFG ($\omega$) and FCFG ($\omega_{\text{aff}}$, $\omega_{\text{inv}}$). FCFG better preserves *invariant* attributes and identity while accurately applying the intended intervention. Compared to standard CFG, FCFG produces counterfactuals with more localized changes and stronger identity preservation.

*Figure A.16.* **Additional qualitative results for `do(sex)` on MIMIC.** Each row shows the original image followed by counterfactuals generated using global CFG ($\omega$) and FCFG ($\omega_{\text{aff}}$, $\omega_{\text{inv}}$). FCFG better preserves *invariant* attributes and identity while accurately applying the intended intervention on `sex`. Compared to standard CFG, which tends to amplify unrelated features such as disease (i.e. `finding`), FCFG produces counterfactuals with more localized, semantically aligned changes and stronger identity preservation.

*Figure A.17.* **Additional qualitative results for `do(race)` on MIMIC.** Each row shows the original image followed by counterfactuals generated using global CFG ($\omega$) and FCFG ($\omega_{\text{aff}}, \omega_{\text{inv}}$). While race interventions correspond to relatively subtle visual changes, standard CFG often amplifies unrelated features such as disease appearance (e.g., `finding`). In contrast, FCFG better preserves *invariant* attributes and identity, producing counterfactuals that are more localized, semantically aligned, and faithful to the intended intervention.

*Figure A.18.* **Reversibility analysis on MIMIC.** Each row shows the original image, the counterfactual generated using global CFG ($\omega$) or FCFG ($\omega_{\text{int}}, \omega_{\text{inv}}$), their corresponding reversed generations, and the associated difference maps (counterfactual - input, and reversed - input). FCFG more faithfully preserves non-intervened attributes and leads to smaller residuals in the difference maps, indicating better identity preservation.

# I. Multi-Attribute Interventions

To demonstrate the generality of the proposed FCFG, we conduct experiments with multi-attribute interventions, i.e., interventions that involve modifying multiple attributes simultaneously. Such interventions can be handled under the two-group partition defined in section 3.3. We also explore an attribute-wise guidance scheme to further highlight the flexibility and generality of FCFG.

Recall that Proposition 3.1 only requires that different groups are mutually independent given the latent variable. In the case of CelebA, the attributes Smiling, Male, and Young are assumed to be conditionally independent of each other (see Section F.1). This independence allows us to treat each attribute as its own group, thereby extending the two-group partition introduced in Section 3.3 to an attribute-wise setting. In this scheme, each attribute is assigned its own guidance weight (e.g., $\omega_s$ for Smiling, $\omega_m$ for Male, and $\omega_y$ for Young), enabling fine-grained and disentangled control over multi-attribute interventions. However, attribute-wise FCFG is more computationally demanding, as evident from eq. 14, which requires evaluating $\epsilon_\theta$ once for the unconditional case and once per group. This results in $M+1$ forward passes (where $M$ is the number of groups), compared to 2 for global CFG and 3 for the two-group FCFG. In the following, we present experimental results with two-attribute interventions in App. I.1 and with three-attribute interventions in App. I.2.

## I.1. Two-Attribute Interventions

We begin with two-attribute interventions, where two of the variables Smiling, Male, and Young are intervened upon simultaneously. Tables A.13, A.14, and A.15 report the Effectiveness (AUC) and Reversibility (MAE, LPIPS) metrics. Across all pairs, global guidance ($\omega$=2.5) yields high effectiveness for the intervened attributes but also amplifies the non-intervened one. Two-group FCFG ($\omega_{\text{aff}}$=2.5, $\omega_{\text{inv}}$=1.0) consistently suppresses such spurious changes while maintaining high effectiveness on the intervened attributes. FCFG further demonstrates its flexibility and generality through the attribute-wise configuration, where each attribute receives its own guidance weight. This allows selective adjustment of individual attributes, while symmetric settings (e.g., $\omega_s$=$\omega_y$=2.5, $\omega_m$=1.0) recover the group-wise performance. Qualitative examples in Figs. A.19, A.20 and A.21 support these findings, showing that attribute-wise FCFG allows finer control over the intervened attributes and reproduces the outcomes of two-group FCFG under symmetric configurations.

*Table A.13.* CelebA: Effectiveness (AUC ↑) and reversibility (MAE, LPIPS ↓) for do(Smiling, Male). Global CFG ($\omega$=2.5) achieves high effectiveness on both intervened attributes but amplifies the non-intervened attribute Young. Two-group and attribute-wise variants of FCFG mitigate this amplification while maintaining strong intervention effectiveness; adjusting a single attribute weight selectively affects the corresponding attribute, and setting $\omega_s$=$\omega_m$=2.5 with $\omega_y = 1.0$ recovers the group-wise configuration ($\omega_{\text{aff}} = 2.5$, $\omega_{\text{inv}} = 1.0$).

| Guidance type | Guidance configuration | Smiling AUC/$\Delta$ | Male AUC/$\Delta$ | Young AUC/$\Delta$ | MAE | LPIPS |
|---|---|---|---|---|---|---|
| CFG (Ho & Salimans, 2022) | $\omega = 1.0$ | 83.3 / +0.0 | 90.7 / +0.0 | 79.3 / +0.0 | 0.117 | 0.082 |
| | $\omega = 2.5$ | 97.7 / +14.4 | 99.5 / +8.8 | 87.7 / +8.4 | 0.227 | 0.155 |
| FCFG (Ours), two-group | $\omega_{\text{aff}} = 2.5, \omega_{\text{inv}} = 1.0$ | 98.9 / +15.6 | 99.0 / +8.3 | 72.9 / -6.4 | 0.189 | 0.123 |
| FCFG (Ours), attribute-wise | $\omega_s = 1.0, \omega_m = 1.0, \omega_y = 1.0$ | 82.1 / -1.2 | 85.5 / -5.2 | 81.1 / +1.8 | 0.144 | 0.102 |
| | $\omega_s = 1.0, \omega_m = 2.5, \omega_y = 1.0$ | 79.5 / -3.8 | 99.4 / +8.7 | 76.1 / -3.2 | 0.171 | 0.120 |
| | $\omega_s = 2.5, \omega_m = 1.0, \omega_y = 1.0$ | 99.3 / +16.0 | 82.4 / -8.3 | 77.3 / -2.0 | 0.172 | 0.119 |
| | $\omega_s = 2.5, \omega_m = 2.0, \omega_y = 1.0$ | 98.7 / +15.4 | 96.3 / +5.6 | 74.6 / -4.7 | 0.175 | 0.114 |
| | $\omega_s = 2.5, \omega_m = 2.5, \omega_y = 1.0$ | 98.4 / +15.1 | 98.7 / +8.0 | 72.4 / -6.9 | 0.186 | 0.121 |
| | $\omega_s = 2.5, \omega_m = 3.0, \omega_y = 1.0$ | 97.6 / +14.3 | 99.5 / +8.8 | 71.0 / -8.3 | 0.198 | 0.128 |

*Table A.14.* CelebA: Effectiveness (AUC ↑) and reversibility (MAE, LPIPS ↓) for do(Smiling, Young). Global CFG ($\omega = 2.5$) achieves high effectiveness on both intervened attributes but amplifies the non-intervened attribute Male. Two-group and attribute-wise variants of FCFG mitigate this amplification while maintaining strong intervention effectiveness; adjusting a single attribute weight selectively affects the corresponding attribute, and setting $\omega_s = \omega_y = 2.5$ with $\omega_m = 1.0$ recovers the two-group configuration ($\omega_{\text{aff}} = 2.5$, $\omega_{\text{inv}} = 1.0$).

| Guidance type | Guidance configuration | Smiling AUC/Δ | Male AUC/Δ | Young AUC/Δ | MAE | LPIPS |
|---|---|---|---|---|---|---|
| CFG (Ho & Salimans, 2022) | $\omega = 1.0$ | 83.6 / +0.0 | 94.6 / +0.0 | 60.1 / +0.0 | 0.123 | 0.094 |
| | $\omega = 2.5$ | 96.8 / +13.2 | 99.8 / +5.2 | 75.7 / +15.6 | 0.221 | 0.138 |
| FCFG (Ours), two-group | $\omega_{\text{aff}} = 2.5$, $\omega_{\text{inv}} = 1.0$ | 97.5 / +13.9 | 85.5 / -9.1 | 79.0 / +18.9 | 0.204 | 0.148 |
| FCFG (Ours), attribute-wise | $\omega_s = 1.0, \omega_m = 1.0, \omega_y = 1.0$ | 82.1 / -1.5 | 93.8 / -0.8 | 58.1 / -2.0 | 0.139 | 0.107 |
| | $\omega_s = 1.0, \omega_m = 1.0, \omega_y = 2.5$ | 77.7 / -5.9 | 85.6 / -9.0 | 84.2 / +24.1 | 0.176 | 0.137 |
| | $\omega_s = 2.5, \omega_m = 1.0, \omega_y = 1.0$ | 98.9 / +15.3 | 91.5 / -3.1 | 54.9 / -5.2 | 0.173 | 0.125 |
| | $\omega_s = 2.5, \omega_m = 1.0, \omega_y = 2.0$ | 97.9 / +14.3 | 87.2 / -7.4 | 71.0 / +10.9 | 0.189 | 0.138 |
| | $\omega_s = 2.5, \omega_m = 1.0, \omega_y = 2.5$ | 97.0 / +13.4 | 86.1 / -8.5 | 77.9 / +17.8 | 0.201 | 0.147 |
| | $\omega_s = 2.5, \omega_m = 1.0, \omega_y = 3.0$ | 96.1 / +12.5 | 83.7 / -10.9 | 84.4 / +24.3 | 0.212 | 0.154 |

*Table A.15.* CelebA: Effectiveness (AUC ↑) and reversibility (MAE, LPIPS ↓) for do(Male, Young). Global CFG ($\omega = 2.5$) achieves high effectiveness on both intervened attributes but amplifies the non-intervened attribute Smiling. Two-group and attribute-wise variants of FCFG mitigate this amplification while maintaining strong intervention effectiveness; adjusting a single attribute weight selectively affects the corresponding attribute, and setting $\omega_m = \omega_y = 2.5$ with $\omega_s = 1.0$ recovers the two-group configuration ($\omega_{\text{aff}} = 2.5$, $\omega_{\text{inv}} = 1.0$).

| Guidance type | Guidance configuration | Smiling AUC/Δ | Male AUC/Δ | Young AUC/Δ | MAE | LPIPS |
|---|---|---|---|---|---|---|
| CFG (Ho & Salimans, 2022) | $\omega = 1.0$ | 82.5 / +0.0 | 89.1 / +0.0 | 63.1 / +0.0 | 0.122 | 0.088 |
| | $\omega = 2.5$ | 98.5 / +16.0 | 99.6 / +10.5 | 79.8 / +16.7 | 0.216 | 0.143 |
| FCFG (Ours), two-group | $\omega_{\text{aff}} = 2.5$, $\omega_{\text{inv}} = 1.0$ | 80.0 / -2.5 | 99.2 / +10.1 | 83.9 / +20.8 | 0.198 | 0.144 |
| FCFG (Ours), attribute-wise | $\omega_s = 1.0, \omega_m = 1.0, \omega_y = 1.0$ | 85.2 / +2.7 | 88.1 / -1.0 | 63.4 / +0.3 | 0.154 | 0.114 |
| | $\omega_s = 1.0, \omega_m = 1.0, \omega_y = 2.5$ | 82.8 / +0.3 | 86.5 / -2.6 | 86.1 / +23.0 | 0.183 | 0.137 |
| | $\omega_s = 1.0, \omega_m = 2.5, \omega_y = 1.0$ | 83.1 / +0.6 | 99.4 / +10.3 | 65.6 / +2.5 | 0.177 | 0.126 |
| | $\omega_s = 1.0, \omega_m = 2.5, \omega_y = 2.0$ | 80.5 / -2.0 | 99.3 / +10.2 | 76.5 / +13.4 | 0.183 | 0.128 |
| | $\omega_s = 1.0, \omega_m = 2.5, \omega_y = 2.5$ | 80.3 / -2.2 | 98.8 / +9.7 | 81.9 / +18.8 | 0.199 | 0.141 |
| | $\omega_s = 1.0, \omega_m = 2.5, \omega_y = 3.0$ | 82.0 / -0.5 | 98.4 / +9.3 | 85.0 / +21.9 | 0.208 | 0.147 |

*Figure A.19.* **Qualitative results for do(Smiling, Male) on CelebA-HQ.** Each row shows the original image followed by counterfactuals generated with two-group FCFG ($\omega_{\text{aff}}, \omega_{\text{inv}}$) and with attribute-wise FCFG ($\omega_{\text{s}}$ for Smiling, $\omega_{\text{m}}$ for Male, and $\omega_{\text{y}}$ for Young). Attribute-wise FCFG provides more flexible configurations, allowing selective control of individual attributes while recovering the two-group FCFG results under symmetric settings.

*Figure A.20.* **Qualitative results for do(Smiling, Young) on CelebA-HQ.** Each row shows the original image followed by counterfactuals generated with two-group FCFG ($\omega_{\mathrm{aff}}, \omega_{\mathrm{inv}}$) and with attribute-wise FCFG ($\omega_{\mathrm{s}}$ for Smiling, $\omega_{\mathrm{m}}$ for Male, and $\omega_{\mathrm{y}}$ for Young). Attribute-wise FCFG provides more flexible configurations, allowing selective control of individual attributes while recovering the two-group FCFG results under symmetric settings.

*Figure A.21.* **Qualitative results for do(Male, Young) on CelebA-HQ.** Each row shows the original image followed by counterfactuals generated with two-group FCFG ($\omega_{\text{aff}}$, $\omega_{\text{inv}}$) and with attribute-wise FCFG ($\omega_{\text{s}}$ for Smiling, $\omega_{\text{m}}$ for Male, and $\omega_{\text{y}}$ for Young). Attribute-wise FCFG provides more flexible configurations, allowing selective control of individual attributes while recovering the two-group FCFG results under symmetric settings.

## I.2. Three-Attribute Interventions

We then move to three-attribute interventions, where all of `Smiling`, `Male`, and `Young` are intervened simultaneously. Table A.16 reports the corresponding Effectiveness (AUC) and Reversibility (MAE, LPIPS) metrics. Notably, in this setting all attributes are intervened, which makes global CFG and two-group FCFG identical, as no invariant attributes remain. In this setting, attribute-wise FCFG provides additional flexibility: it enables selective control of the guidance strength across attributes, while symmetric settings (e.g., $\omega_s = \omega_m = \omega_y = 2.5$) recover the outcomes of global CFG/two-group FCFG ($\omega = 2.5$). Qualitative examples in Fig. A.22 further illustrate this flexibility, showing that attribute-wise FCFG can selectively control each attribute under the all-attribute intervention.

*Table A.16.* **CelebA: Effectiveness (AUC ↑) and reversibility (MAE, LPIPS ↓) for do(`Smiling`, `Male`, `Young`).** Since all three attributes are intervened, no invariant attributes remain; consequently, global CFG and the two-group variant of FCFG are equivalent, as the same guidance weight is applied to all attributes. Attribute-wise FCFG further demonstrates flexibility by enabling independent control over each attribute; setting $\omega_s = \omega_m = \omega_y = 2.5$ recovers the global/two-group configuration.

| Guidance type | Guidance configuration | Smiling AUC/Δ | Male AUC/Δ | Young AUC/Δ | MAE | LPIPS |
|---|---|---|---|---|---|---|
| CFG (Ho & Salimans, 2022) | $\omega = 1.0$ | 86.1 / +0.0 | 88.4 / +0.0 | 64.0 / +0.0 | 0.124 | 0.093 |
| | $\omega = 2.5^1$ | 98.1 / +12.0 | 99.0 / +10.6 | 84.7 / +20.7 | 0.207 | 0.138 |
| FCFG (Ours), attribute-wise | $\omega_s = 1.0, \omega_m = 1.0, \omega_y = 1.0$ | 84.1 / -2.0 | 88.6 / +0.2 | 66.4 / +2.4 | 0.151 | 0.114 |
| | $\omega_s = 2.5, \omega_m = 1.0, \omega_y = 1.0$ | 99.3 / +13.2 | 86.8 / -1.6 | 66.3 / +2.3 | 0.176 | 0.123 |
| | $\omega_s = 1.0, \omega_m = 2.5, \omega_y = 1.0$ | 80.9 / -5.2 | 99.3 / +10.9 | 64.8 / +0.8 | 0.183 | 0.130 |
| | $\omega_s = 1.0, \omega_m = 1.0, \omega_y = 2.5$ | 79.1 / -7.0 | 86.6 / -1.8 | 88.5 / +24.5 | 0.181 | 0.141 |
| | $\omega_s = 1.0, \omega_m = 2.5, \omega_y = 2.5$ | 79.0 / -7.1 | 99.2 / +10.8 | 83.8 / +19.8 | 0.189 | 0.142 |
| | $\omega_s = 2.5, \omega_m = 1.0, \omega_y = 2.5$ | 97.8 / +11.7 | 86.1 / -2.3 | 85.9 / +21.9 | 0.188 | 0.147 |
| | $\omega_s = 2.5, \omega_m = 2.5, \omega_y = 1.0$ | 98.4 / +12.3 | 98.4 / +10.0 | 67.7 / +3.7 | 0.191 | 0.126 |
| | $\omega_s = 2.5, \omega_m = 2.5, \omega_y = 2.0$ | 97.9 / +11.8 | 98.5 / +10.1 | 77.6 / +13.6 | 0.198 | 0.133 |
| | $\omega_s = 2.5, \omega_m = 2.5, \omega_y = 2.5$ | 97.7 / +11.6 | 98.3 / +9.9 | 81.6 / +17.6 | 0.210 | 0.139 |
| | $\omega_s = 2.5, \omega_m = 2.5, \omega_y = 3.0$ | 97.6 / +11.5 | 98.6 / +10.2 | 84.8 / +20.8 | 0.220 | 0.151 |

---

[1]When all attributes are intervened, two-group FCFG is equivalent to global CFG, as the same guidance weight is applied to every attribute.

*Figure A.22.* **Qualitative results for do(Smiling, Male, Young) on CelebA-HQ.** The first column shows the original image, followed by counterfactuals generated with global/two-group FCFG ($\omega$=2.5) and with attribute-wise FCFG ($\omega_{\mathrm{s}}, \omega_{\mathrm{m}}, \omega_{\mathrm{y}}$). Attribute-wise FCFG enables selective control of the three attributes (e.g., raising only $\omega_{\mathrm{s}}$ or $\omega_{\mathrm{y}}$) while symmetric settings (e.g., $\omega_{\mathrm{s}}$=$\omega_{\mathrm{m}}$=$\omega_{\mathrm{y}}$=2.5) reproduce the outcomes of the global/two-group configuration.

## J. Compatibility with CFG++.

**Counterfactual prediction with CFG++.** Classifier-Free Guidance++ (CFG++) (Chung et al., 2025) modifies DDIM sampling by interpolating between unconditional and conditional noise predictions using a mixing parameter $\lambda \in [0, 1]$. In the standard global setting, the guided noise prediction is defined as

$$\epsilon_{\text{CFG++}}(\mathbf{x}_t, t, \mathbf{c}) = \epsilon_\theta(\mathbf{x}_t, t, \varnothing) + \lambda\Big(\epsilon_\theta(\mathbf{x}_t, t, \mathbf{c}) - \epsilon_\theta(\mathbf{x}_t, t, \varnothing)\Big), \tag{31}$$

where $\epsilon_\theta(\mathbf{x}_t, t, \varnothing)$ and $\epsilon_\theta(\mathbf{x}_t, t, \mathbf{c})$ denote the unconditional and conditional noise predictions, respectively. In CFG++, this guided noise prediction is used only to compute the denoised estimate $\hat{\mathbf{x}}_0$, while the reverse DDIM update direction is taken from the unconditional prediction $\epsilon_\theta(\mathbf{x}_t, t, \varnothing)$. The resulting reverse diffusion procedure for counterfactual prediction is summarized in Algorithm 6.

---

**Algorithm 6** Counterfactual Prediction with CFG++

---

**Input:** exogenous noise estimate $\mathbf{u}$ (from *abduction*), counterfactual conditioning embedding $\tilde{\mathbf{c}}$ (from *action*), CFG++ interpolation parameter $\lambda \in [0, 1]$, DDIM schedule $\{\alpha_t\}_{t=0}^T$, null conditioning token $\varnothing$

**Counterfactual prediction:**

1: $\mathbf{x}_T \leftarrow \mathbf{u}$
2: **for** $t = T$ down to $1$ **do**
3: $\quad \epsilon_{\text{CFG++}} \leftarrow \epsilon_\theta(\mathbf{x}_t, t, \varnothing) + \lambda\Big(\epsilon_\theta(\mathbf{x}_t, t, \tilde{\mathbf{c}}) - \epsilon_\theta(\mathbf{x}_t, t, \varnothing)\Big)$
4: $\quad \hat{\mathbf{x}}_0 \leftarrow \frac{1}{\sqrt{\alpha_t}}\left(\mathbf{x}_t - \sqrt{1 - \alpha_t}\,\epsilon_{\text{CFG++}}\right)$
5: $\quad \mathbf{x}_{t-1} \leftarrow \sqrt{\alpha_{t-1}}\,\hat{\mathbf{x}}_0 + \sqrt{1 - \alpha_{t-1}}\,\epsilon_\theta(\mathbf{x}_t, t, \varnothing)$
6: **end for**

**Output:** counterfactual image $\tilde{\mathbf{x}} \leftarrow \mathbf{x}_0$

---

**Counterfactual prediction with FCFG++.** The key idea of Factored Classifier-Free Guidance is to decompose the conditional signal into multiple group-specific components defined over semantically or causally related attributes (see Sec. 3.2). For the two-group (affected/invariant) setting used in counterfactual generation (see Sec. 3.3), we extend CFG++ by applying separate interpolation weights to the corresponding masked conditioning embeddings. We term this variant *FCFG++*. Concretely, the guided noise prediction in FCFG++ is defined as

$$\epsilon_{\text{FCFG++}}(\mathbf{x}_t, t, \mathbf{c}) = \epsilon_\theta(\mathbf{x}_t, t, \varnothing) + \lambda_{\text{aff}}\Big(\epsilon_\theta(\mathbf{x}_t, t, \underline{\mathbf{c}}^{(\text{aff})}) - \epsilon_\theta(\mathbf{x}_t, t, \varnothing)\Big) + \lambda_{\text{inv}}\Big(\epsilon_\theta(\mathbf{x}_t, t, \underline{\mathbf{c}}^{(\text{inv})}) - \epsilon_\theta(\mathbf{x}_t, t, \varnothing)\Big), \tag{32}$$

where $\underline{\mathbf{c}}^{(\text{aff})}$ and $\underline{\mathbf{c}}^{(\text{inv})}$ denote the masked conditioning embeddings corresponding to the affected attributes $\mathbf{pa}^{\text{aff}}$ and invariant attributes $\mathbf{pa}^{\text{inv}}$, respectively (see Eq. (13)). As in CFG++, the guided prediction is used to compute $\hat{\mathbf{x}}_0$, while the reverse DDIM update direction remains unconditional. The reverse diffusion procedure FCFG++ is summarized in Algorithm 7.

---

**Algorithm 7** Counterfactual Prediction FCFG++ (Ours; two-group partition)

---

**Input:** exogenous noise estimate $\mathbf{u}$ (from *abduction*), counterfactual conditioning embedding $\tilde{\mathbf{c}}$ (from *action*), masked embeddings $\underline{\tilde{\mathbf{c}}}^{\text{aff}}$, $\underline{\tilde{\mathbf{c}}}^{\text{inv}}$, guidance weights $\lambda_{\text{aff}}, \lambda_{\text{inv}} \in [0, 1]$, DDIM schedule $\{\alpha_t\}_{t=0}^T$, null conditioning token $\varnothing$

**Counterfactual prediction:**

1: $\mathbf{x}_T \leftarrow \mathbf{u}$
2: **for** $t = T$ down to $1$ **do**
3: $\quad \epsilon_{\text{FCFG++}} \leftarrow \epsilon_\theta(\mathbf{x}_t, t, \varnothing) + \lambda_{\text{aff}}\Big(\epsilon_\theta(\mathbf{x}_t, t, \underline{\tilde{\mathbf{c}}}^{\text{aff}}) - \epsilon_\theta(\mathbf{x}_t, t, \varnothing)\Big) + \lambda_{\text{inv}}\Big(\epsilon_\theta(\mathbf{x}_t, t, \underline{\tilde{\mathbf{c}}}^{\text{inv}}) - \epsilon_\theta(\mathbf{x}_t, t, \varnothing)\Big)$
4: $\quad \hat{\mathbf{x}}_0 \leftarrow \frac{1}{\sqrt{\alpha_t}}\left(\mathbf{x}_t - \sqrt{1 - \alpha_t}\,\epsilon_{\text{FCFG++}}\right)$
5: $\quad \mathbf{x}_{t-1} \leftarrow \sqrt{\alpha_{t-1}}\,\hat{\mathbf{x}}_0 + \sqrt{1 - \alpha_{t-1}}\,\epsilon_\theta(\mathbf{x}_t, t, \varnothing)$
6: **end for**

**Output:** counterfactual image $\tilde{\mathbf{x}} \leftarrow \mathbf{x}_0$

---

*Table A.17.* CelebA: Effectiveness (ROC-AUC ↑) and reversibility (MAE, LPIPS ↓) of counterfactual generation under different guidance settings. Effectiveness is measured by the ROC-AUC of downstream attribute classifiers on generated counterfactual images, with Δ indicating the change relative to the baseline CFG ($\omega = 1.0$), while reversibility evaluates the reconstruction fidelity of reversed generations. CFG and CFG++ denote global classifier-free guidance baselines; CFG++, like CFG, exhibits attribute amplification, i.e., increasing effectiveness on unintervened attributes as the guidance strength $\lambda$ increases. FCFG and FCFG++ correspond to the proposed group-wise guidance variants applied to CFG and CFG++, respectively, demonstrating that FCFG is compatible with both samplers and can mitigate unintended attribute amplification while preserving intervention effectiveness.

| do(key) | Guidance | Guidance Configurations | Smiling AUC/Δ | Male AUC/Δ | Young AUC/Δ | MAE | LPIPS |
|---|---|---|---|---|---|---|---|
| do(Smiling) | CFG (Ho & Salimans, 2022) | $\omega = 1.0$ | 86.5 / +0.0 | 96.9 / +0.0 | 78.6 / +0.0 | 0.113 | 0.082 |
| | | $\omega = 1.5$ | 95.5 / +9.0 | 99.2 / +2.3 | 84.3 / +5.7 | 0.163 | 0.111 |
| | | $\omega = 3.0$ | 98.8 / +12.3 | 99.9 / +3.0 | 90.3 / +11.7 | 0.263 | 0.155 |
| | FCFG (Ours) | $\omega_{\text{aff}} = 1.5, \omega_{\text{inv}} = 1.2$ | 93.9 / +7.4 | 97.5 / +0.6 | 79.7 / +1.1 | 0.136 | 0.092 |
| | | $\omega_{\text{aff}} = 3.0, \omega_{\text{inv}} = 1.2$ | 99.6 / +13.1 | 95.4 / -1.5 | 77.6 / -1.0 | 0.177 | 0.122 |
| | CFG++ (Chung et al., 2025) | $\lambda = 0.01$ | 96.0 / +9.5 | 99.1 / +2.2 | 83.9 / +5.3 | 0.159 | 0.108 |
| | | $\lambda = 0.02$ | 99.2 / +12.7 | 99.8 / +2.9 | 90.1 / +11.5 | 0.251 | 0.154 |
| | FCFG++ (Ours) | $\lambda_{\text{aff}} = 0.01, \lambda_{\text{inv}} = 0.008$ | 95.0 / +8.5 | 97.4 / +0.5 | 80.4 / +1.8 | 0.138 | 0.095 |
| | | $\lambda_{\text{aff}} = 0.02, \lambda_{\text{inv}} = 0.008$ | 99.7 / +13.2 | 95.5 / -1.4 | 77.7 / -0.9 | 0.175 | 0.121 |
| do(Male) | CFG (Ho & Salimans, 2022) | $\omega = 1.0$ | 86.6 / +0.0 | 91.8 / +0.0 | 79.8 / +0.0 | 0.115 | 0.079 |
| | | $\omega = 1.5$ | 93.3 / +6.7 | 97.2 / +5.4 | 82.0 / +2.2 | 0.158 | 0.111 |
| | | $\omega = 3.0$ | 98.2 / +11.6 | 99.2 / +7.4 | 90.2 / +10.4 | 0.267 | 0.171 |
| | FCFG (Ours) | $\omega_{\text{aff}} = 1.5, \omega_{\text{inv}} = 1.2$ | 88.3 / +1.7 | 93.8 / +2.0 | 78.9 / -0.9 | 0.149 | 0.101 |
| | | $\omega_{\text{aff}} = 3.0, \omega_{\text{inv}} = 1.2$ | 87.4 / +0.8 | 99.7 / +7.9 | 75.9 / -3.9 | 0.188 | 0.130 |
| | CFG++ (Chung et al., 2025) | $\lambda = 0.01$ | 94.4 / +7.8 | 97.1 / +5.3 | 82.7 / +2.9 | 0.154 | 0.107 |
| | | $\lambda = 0.02$ | 98.4 / +11.8 | 99.3 / +7.5 | 88.7 / +8.9 | 0.259 | 0.170 |
| | FCFG++ (Ours) | $\lambda_{\text{aff}} = 0.01, \lambda_{\text{inv}} = 0.008$ | 90.1 / +3.5 | 94.2 / +2.4 | 78.2 / -1.6 | 0.146 | 0.100 |
| | | $\lambda_{\text{aff}} = 0.02, \lambda_{\text{inv}} = 0.008$ | 89.0 / +2.4 | 99.7 / +7.9 | 76.1 / -3.7 | 0.186 | 0.127 |
| do(Young) | CFG (Ho & Salimans, 2022) | $\omega = 1.0$ | 87.5 / +0.0 | 95.7 / +0.0 | 62.3 / +0.0 | 0.115 | 0.085 |
| | | $\omega = 1.5$ | 95.6 / +8.1 | 99.3 / +3.6 | 66.3 / +4.0 | 0.166 | 0.110 |
| | | $\omega = 3.0$ | 98.5 / +11.0 | 99.9 / +4.2 | 77.7 / +15.4 | 0.261 | 0.160 |
| | FCFG (Ours) | $\omega_{\text{aff}} = 1.5, \omega_{\text{inv}} = 1.2$ | 90.0 / +2.5 | 96.7 / +1.0 | 67.4 / +5.1 | 0.147 | 0.100 |
| | | $\omega_{\text{aff}} = 3.0, \omega_{\text{inv}} = 1.2$ | 86.7 / -0.8 | 90.0 / -5.7 | 87.6 / +25.3 | 0.188 | 0.136 |
| | CFG++ (Chung et al., 2025) | $\lambda = 0.01$ | 95.1 / +7.6 | 99.3 / +3.6 | 66.0 / +3.7 | 0.159 | 0.110 |
| | | $\lambda = 0.02$ | 99.0 / +11.5 | 99.9 / +4.2 | 78.1 / +15.8 | 0.257 | 0.165 |
| | FCFG++ (Ours) | $\lambda_{\text{aff}} = 0.01, \lambda_{\text{inv}} = 0.008$ | 91.4 / +3.9 | 97.4 / +1.7 | 67.1 / +4.8 | 0.145 | 0.104 |
| | | $\lambda_{\text{aff}} = 0.02, \lambda_{\text{inv}} = 0.008$ | 88.2 / +0.7 | 90.3 / -5.4 | 87.0 / +24.7 | 0.192 | 0.147 |

**Experiments.** We conduct experiments on CelebA-HQ to evaluate counterfactual generation under CFG++ and its compatibility with the proposed FCFG. As in counterfactual generation with CFG and FCFG, the *abduction* and *action* steps follow Algorithm 1 and Algorithm 2, respectively. The *counterfactual prediction* step follows Algorithm 6 for CFG++ and Algorithm 7 for FCFG++. Our results confirm that CFG++ suffers from attribute amplification, and demonstrate that FCFG remains effective when combined with CFG++, mitigating unintended changes to non-intervened attributes. Detailed quantitative and qualitative results, together with further discussion, are reported below.

Table A.17 reports the effectiveness and reversibility of counterfactual generation on CelebA under different guidance strategies. For the global baselines, both CFG and CFG++ exhibit a monotonic increase in intervention effectiveness as the guidance strength ($\omega$ or $\lambda$) increases. However, these gains are consistently accompanied by amplified responses in *non-intervened* attributes, indicating that CFG++ inherits the attribute amplification behaviour observed in standard CFG. Notably, CFG++ with $\lambda = 0.02$ achieves intervention performance comparable to CFG with $\omega = 3.0$, while inducing a similar degree of unintended amplification on invariant attributes.

In contrast, the proposed FCFG and FCFG++ explicitly decouple the guidance applied to intervened and invariant attribute groups. Across all three interventions, FCFG++ preserves strong improvements on the targeted attribute while substantially reducing unintended amplification on non-intervened attributes, compared to both CFG and CFG++. In particular, FCFG++ with $\lambda_{\text{aff}} = 0.02$ and $\lambda_{\text{inv}} = 0.008$ achieves intervention effectiveness comparable to FCFG with $\omega_{\text{aff}} = 3.0$ and $\omega_{\text{inv}} = 1.2$, with both configurations exhibiting markedly weaker amplification on uninvolved attributes. This comparison shows that FCFG++ maintains strong intervention strength without inheriting the amplification effects intrinsic to global guidance.

*Figure A.23.* **Qualitative comparison of CFG++ and FCFG++ for counterfactual generation and reversibility on CelebA.** For each row, we show the original image (left), the counterfactual generated under a single-attribute intervention (e.g., `do(Smiling)`, `do(Male)`, `do(Young)`), and the corresponding reversed image obtained by applying the inverse intervention. While CFG++ achieves strong intervention effects, it exhibits attribute amplification similar to standard CFG, resulting in degraded reversibility and unintended changes in non-intervened attributes in the reversed images. In contrast, FCFG++ preserves the intended intervention strength while more faithfully maintaining non-intervened attributes, producing visually and semantically more consistent reversals across different interventions. These results highlight the benefit of integrating FCFG with CFG++, improving both targeted editability and reversibility.

Fig. A.23 provides qualitative comparisons between CFG++ and FCFG++ for counterfactual generation and reversibility on CelebA across multiple single-attribute interventions. While CFG++ successfully enforces the intended attribute changes (e.g., smiling, gender, or age), the reversed images frequently exhibit identity drift and unintended modifications to non-intervened attributes, reflecting the attribute amplification behaviour observed quantitatively. In contrast, FCFG++ produces counterfactuals with comparable intervention strength while yielding substantially more faithful reversals: non-intervened attributes are better preserved, and the reversed images remain visually and semantically closer to the original inputs across different intervention types.

In summary, these quantitative and qualitative results demonstrate that the observed behaviour is consistent across both guidance formulations (CFG and CFG++), confirming that FCFG is fully compatible with CFG++ and can be integrated without modifying the underlying diffusion update. Moreover, reversibility metrics (MAE and LPIPS), together with the qualitative evidence, indicate that FCFG and FCFG++ achieve comparable or improved reconstruction fidelity relative to their global counterparts. Overall, these results establish FCFG++ as an effective extension of CFG++ for counterfactual generation with improved attribute control, mitigating unintended attribute amplification without sacrificing targeted editability or reversibility.

# K. Compatibility with APG.

**Counterfactual prediction with APG.**   Adaptive Projected Guidance (APG) (Sadat et al., 2025) is a refinement of classifier-free guidance that modifies the denoised estimate at each diffusion step by selectively filtering the discrepancy between conditional and unconditional predictions.

Let $\mathbf{x}_t$ denote the latent variable at timestep $t$. We define the DDIM *denoiser operator* at timestep $t$ as the mapping from a noise prediction $\boldsymbol{\epsilon}$ to the corresponding denoised estimate $\hat{\mathbf{x}}_0$:

$$\mathcal{D}_t(\mathbf{x}_t; \boldsymbol{\epsilon}) \triangleq \frac{1}{\sqrt{\alpha_t}} \Big( \mathbf{x}_t - \sqrt{1 - \alpha_t}\,\boldsymbol{\epsilon} \Big). \tag{33}$$

Given unconditional and conditional noise predictions $\boldsymbol{\epsilon}_\theta(\mathbf{x}_t, t, \varnothing)$ and $\boldsymbol{\epsilon}_\theta(\mathbf{x}_t, t, \mathbf{c})$, respectively, the corresponding denoised predictions are then

$$\hat{\mathbf{x}}_0(\mathbf{x}_t, t, \varnothing) = \mathcal{D}_t(\mathbf{x}_t; \boldsymbol{\epsilon}_\theta(\mathbf{x}_t, t, \varnothing)), \tag{34}$$

$$\hat{\mathbf{x}}_0(\mathbf{x}_t, t, \mathbf{c}) = \mathcal{D}_t(\mathbf{x}_t; \boldsymbol{\epsilon}_\theta(\mathbf{x}_t, t, \mathbf{c})). \tag{35}$$

APG operates on the difference between the conditional and unconditional denoised predictions:

$$\Delta\hat{\mathbf{x}}_0 = \hat{\mathbf{x}}_0(\mathbf{x}_t, t, \mathbf{c}) - \hat{\mathbf{x}}_0(\mathbf{x}_t, t, \varnothing). \tag{36}$$

This difference is decomposed into components parallel and orthogonal to the conditional denoised prediction $\hat{\mathbf{x}}_0(\mathbf{x}_t, t, \mathbf{c})$. The parallel component is given by

$$\Delta\hat{\mathbf{x}}_0^{\parallel} = \frac{\langle \Delta\hat{\mathbf{x}}_0,\ \hat{\mathbf{x}}_0(\mathbf{x}_t, t, \mathbf{c}) \rangle}{\|\hat{\mathbf{x}}_0(\mathbf{x}_t, t, \mathbf{c})\|_2^2}\, \hat{\mathbf{x}}_0(\mathbf{x}_t, t, \mathbf{c}), \tag{37}$$

and the orthogonal component by

$$\Delta\hat{\mathbf{x}}_0^{\perp} = \Delta\hat{\mathbf{x}}_0 - \Delta\hat{\mathbf{x}}_0^{\parallel}. \tag{38}$$

We define the *APG operator* as

$$\mathrm{APG}\big(\hat{\mathbf{x}}_0(\mathbf{x}_t, t, \mathbf{c}), \hat{\mathbf{x}}_0(\mathbf{x}_t, t, \varnothing); \eta\big) = \Delta\hat{\mathbf{x}}_0^{\perp} + \eta\,\Delta\hat{\mathbf{x}}_0^{\parallel}, \tag{39}$$

---

**Algorithm 8** Counterfactual Prediction with APG

**Input:** exogenous noise estimate $\mathbf{u}$ (from *abduction*), counterfactual conditioning embedding $\tilde{\mathbf{c}}$ (from *action*), guidance weight $\omega \geq 0$, APG parameter $\eta \in [0, 1]$, DDIM schedule $\{\alpha_t\}_{t=0}^{T}$, null conditioning token $\varnothing$
**Counterfactual prediction:**
1: $\mathbf{x}_T \leftarrow \mathbf{u}$
2: **for** $t = T$ down to $1$ **do**
3:     Compute unconditional and conditional denoised predictions (Eq. (33)):

$$\hat{\mathbf{x}}_0^{\varnothing} \leftarrow \mathcal{D}_t(\mathbf{x}_t; \boldsymbol{\epsilon}_\theta(\mathbf{x}_t, t, \varnothing)), \quad \hat{\mathbf{x}}_0^{\tilde{\mathbf{c}}} \leftarrow \mathcal{D}_t(\mathbf{x}_t; \boldsymbol{\epsilon}_\theta(\mathbf{x}_t, t, \tilde{\mathbf{c}}))$$

4:     Apply APG projection (Eq. (39)):
$$\hat{\mathbf{x}}_0^{\mathrm{APG}} \leftarrow \hat{\mathbf{x}}_0^{\varnothing} + \omega \cdot \mathrm{APG}\big(\hat{\mathbf{x}}_0^{\tilde{\mathbf{c}}}, \hat{\mathbf{x}}_0^{\varnothing}; \eta\big)$$

5:     Convert APG-modified denoised prediction to noise:

$$\boldsymbol{\epsilon}^{\mathrm{APG}} \leftarrow \frac{\mathbf{x}_t - \sqrt{\alpha_t}\,\hat{\mathbf{x}}_0^{\mathrm{APG}}}{\sqrt{1 - \alpha_t}}$$

6:     Deterministic DDIM update:

$$\mathbf{x}_{t-1} \leftarrow \sqrt{\alpha_{t-1}}\,\hat{\mathbf{x}}_0^{\mathrm{APG}} + \sqrt{1 - \alpha_{t-1}}\,\boldsymbol{\epsilon}^{\mathrm{APG}}$$

7: **end for**
**Output:** counterfactual image $\tilde{\mathbf{x}} \leftarrow \mathbf{x}_0$

---

where $\Delta\hat{\mathbf{x}}_0^{\parallel}$ is obtained via Eq. (37), $\Delta\hat{\mathbf{x}}_0^{\perp}$ is obtained via Eq. (38), and $\eta \in [0, 1]$ controls the contribution of the parallel component. Following Sadat et al. (2025), we set $\eta = 0$ in all experiments and adopt the recommended APG hyperparameters of Sadat et al. (2025), using reverse momentum $\beta = -0.75$ and norm-threshold rescaling with radius $r = 2.5$. For simplicity, we omit the explicit formulation of APG reverse momentum and norm-threshold rescaling in our algorithms.

The final APG-guided denoised prediction is then written compactly as

$$\hat{\mathbf{x}}_0^{\text{APG}} = \hat{\mathbf{x}}_0(\mathbf{x}_t, t, \varnothing) + \omega \cdot \text{APG}\big(\hat{\mathbf{x}}_0(\mathbf{x}_t, t, \mathbf{c}), \hat{\mathbf{x}}_0(\mathbf{x}_t, t, \varnothing); \eta\big), \tag{40}$$

where $\omega \geq 0$ denotes the guidance strength.

This APG-modified denoised prediction is converted back to a noise estimate and used in the standard deterministic DDIM update, as summarized in Algorithm 8.

**Counterfactual prediction with FAPG.** The core idea of FCFG is to decompose the conditional signal used for counterfactual generation into semantically distinct groups and to control their influence independently during sampling. This idea can be naturally integrated with APG.

Concretely, given the exogenous noise estimate $\mathbf{u}$ from the *abduction* step and the counterfactual conditioning embedding $\tilde{\mathbf{c}}$ from the *action* step, we first compute the unconditional denoised prediction $\hat{\mathbf{x}}_0^{\varnothing} = \mathcal{D}_t(\mathbf{x}_t; \boldsymbol{\epsilon}_\theta(\mathbf{x}_t, t, \varnothing))$. We then construct a collection of group-wise conditional denoised predictions $\{\hat{\mathbf{x}}_0^{(m)}\}_{m=1}^M$ using masked conditioning embeddings $\{\underline{\tilde{\mathbf{c}}}^{(m)}\}_{m=1}^M$ defined in Eq. (13), via

$$\hat{\mathbf{x}}_0^{(m)} = \mathcal{D}_t\Big(\mathbf{x}_t; \boldsymbol{\epsilon}_\theta(\mathbf{x}_t, t, \underline{\tilde{\mathbf{c}}}^{(m)})\Big).$$

---

**Algorithm 9** Counterfactual Prediction with FAPG (Ours; two-group partition)

---

**Input:** exogenous noise estimate $\mathbf{u}$ (from *abduction*), counterfactual conditioning embedding $\tilde{\mathbf{c}}$ (from *action*), masked embeddings $\underline{\tilde{\mathbf{c}}}^{\text{aff}}, \underline{\tilde{\mathbf{c}}}^{\text{inv}}$, group-wise guidance weights $\omega_{\text{aff}}, \omega_{\text{inv}} \geq 0$, APG parameter $\eta=0$, DDIM schedule $\{\alpha_t\}_{t=0}^T$, null conditioning token $\varnothing$

**Counterfactual prediction:**

1: $\mathbf{x}_T \leftarrow \mathbf{u}$
2: **for** $t = T$ down to 1 **do**
3:     Compute unconditional denoised prediction (Eq. 33):

$$\hat{\mathbf{x}}_0^{\varnothing} \leftarrow \mathcal{D}_t(\mathbf{x}_t; \boldsymbol{\epsilon}_\theta(\mathbf{x}_t, t, \varnothing))$$

4:     Compute group-wise conditional denoised predictions (Eq. (33)):

$$\hat{\mathbf{x}}_0^{\text{aff}} \leftarrow \mathcal{D}_t\big(\mathbf{x}_t; \boldsymbol{\epsilon}_\theta(\mathbf{x}_t, t, \underline{\tilde{\mathbf{c}}}^{\text{aff}})\big), \quad \hat{\mathbf{x}}_0^{\text{inv}} \leftarrow \mathcal{D}_t\big(\mathbf{x}_t; \boldsymbol{\epsilon}_\theta(\mathbf{x}_t, t, \underline{\tilde{\mathbf{c}}}^{\text{inv}})\big)$$

5:     Apply APG projection to each group (Eq. (39)):

$$\Delta\hat{\mathbf{x}}_0^{\text{APG},m} \leftarrow \text{APG}\big(\hat{\mathbf{x}}_0^m, \hat{\mathbf{x}}_0^{\varnothing}; \eta\big), \quad m \in \{\text{aff}, \text{inv}\}$$

6:     Combine group-wise APG updates:

$$\hat{\mathbf{x}}_0^{\text{FAPG}} \leftarrow \hat{\mathbf{x}}_0^{\varnothing} + \omega_{\text{aff}} \, \Delta\hat{\mathbf{x}}_0^{\text{APG,aff}} + \omega_{\text{inv}} \, \Delta\hat{\mathbf{x}}_0^{\text{APG,inv}}$$

7:     Convert APG-modified denoised prediction to noise:

$$\boldsymbol{\epsilon}^{\text{FAPG}} \leftarrow \frac{\mathbf{x}_t - \sqrt{\alpha_t} \, \hat{\mathbf{x}}_0^{\text{FAPG}}}{\sqrt{1 - \alpha_t}}$$

8:     Deterministic DDIM update:

$$\mathbf{x}_{t-1} \leftarrow \sqrt{\alpha_{t-1}} \, \hat{\mathbf{x}}_0^{\text{FAPG}} + \sqrt{1 - \alpha_{t-1}} \, \boldsymbol{\epsilon}^{\text{FAPG}}$$

9: **end for**
**Output:** counterfactual image $\tilde{\mathbf{x}} \leftarrow \mathbf{x}_0$

---

Rather than applying APG to a single aggregated conditional prediction, we apply APG *independently to each group*. Specifically, for each group $m$, we apply the APG operator (Eq. (39)) to the difference between the group-wise and unconditional denoised predictions:

$$\Delta\hat{\mathbf{x}}_0^{\text{APG},(m)} = \text{APG}\left(\hat{\mathbf{x}}_0^{(m)}, \hat{\mathbf{x}}_0^{\varnothing}; \eta\right).$$

The final APG-guided denoised prediction is then obtained by combining the group-wise APG updates using guidance weights $\{\omega_m\}_{m=1}^{M}$:

$$\hat{\mathbf{x}}_0^{\text{FAPG}} = \hat{\mathbf{x}}_0^{\varnothing} + \sum_{m=1}^{M} \omega_m \, \Delta\hat{\mathbf{x}}_0^{\text{APG},(m)}.$$

This APG-guided denoised prediction is converted back to a noise estimate and used in the deterministic reverse DDIM update to obtain $\mathbf{x}_{t-1}$. For clarity, we refer to the integration of FCFG into APG as *FAPG*. The full counterfactual prediction procedure is summarized in Algorithm 9.

**Experiments.**   We conduct experiments on CelebA-HQ to evaluate counterfactual generation under APG and its compatibility with the proposed FCFG. As in counterfactual generation with CFG and FCFG, the *abduction* and *action* steps follow Algorithm 1 and Algorithm 2, respectively. The *counterfactual prediction* step follows the APG formulation, while FAPG applies the proposed factored guidance mechanism on top of APG.

Our results show that APG, despite producing strong intervention effects, still suffers from attribute amplification, leading to unintended changes in non-intervened attributes and degraded reversibility. In contrast, FAPG remains effective when combined with APG, substantially mitigating unintended attribute amplification while preserving intervention strength and reversibility. Detailed quantitative and qualitative results, together with further discussion, are reported below.

Table A.18 reports the effectiveness and reversibility of counterfactual generation on CelebA under APG-based guidance. Similar to CFG, APG improves intervention effectiveness as the guidance strength $\omega$ increases, but these gains are consistently accompanied by amplified responses in *invariant* attributes, indicating that APG inherits the attribute amplification behaviour of global guidance. This effect is reflected both quantitatively, through increased ROC-AUC on uninvolved attributes, and qualitatively, through noticeable identity drift and semantic inconsistency in reversed generations (Fig. A.24).

In contrast, the proposed FCFG explicitly factors the guidance signal into intervened and invariant attribute groups, enabling independent control over their respective contributions. When combined with APG, FAPG preserves strong improvements on the targeted attribute while substantially suppressing unintended amplification on non-intervened attributes across all three interventions. As shown in Table A.18 and the qualitative examples in Fig. A.24, FAPG yields more faithful reversals and improved identity preservation compared to APG, without sacrificing intervention effectiveness. These results demonstrate that the proposed factored guidance framework is fully compatible with APG and effectively mitigates attribute amplification in both quantitative metrics and visual outcomes.

*Table A.18.* CelebA: Effectiveness (ROC-AUC ↑) and reversibility (MAE, LPIPS ↓) of counterfactual generation under different guidance settings. Effectiveness is measured by the ROC-AUC of downstream attribute classifiers evaluated on generated counterfactual images, with $\Delta$ denoting the change relative to the baseline CFG ($\omega = 1.0$). Reversibility evaluates the reconstruction identity preservation of reversed generations. Both CFG and APG exhibit attribute amplification, leading to unintended changes in non-intervened attributes as guidance strength increases. In contrast, FCFG and FAPG mitigate this effect by decoupling guidance applied to intervened and invariant attribute groups. Compared to APG, FAPG achieves improved control over unintended attribute amplification while preserving intervention effectiveness and reversibility, demonstrating that the proposed factored guidance framework is compatible with APG.

| do(key) | Guidance | Guidance Configuration | Smiling AUC/$\Delta$ | Male AUC/$\Delta$ | Young AUC/$\Delta$ | MAE | LPIPS |
|---|---|---|---|---|---|---|---|
| do(Smiling) | CFG | $\omega = 1.0$ | 86.5 / +0.0 | 96.9 / +0.0 | 78.6 / +0.0 | 0.113 | 0.082 |
| | | $\omega = 1.5$ | 95.5 / +9.0 | 99.2 / +2.3 | 84.3 / +5.7 | 0.163 | 0.111 |
| | | $\omega = 3.0$ | 98.8 / +12.3 | 99.9 / +3.0 | 90.3 / +11.7 | 0.263 | 0.155 |
| | FCFG (Ours) | $\omega_{\mathrm{aff}} = 1.5$, $\omega_{\mathrm{inv}} = 1.2$ | 93.9 / +7.4 | 97.5 / +0.6 | 79.7 / +1.1 | 0.136 | 0.092 |
| | | $\omega_{\mathrm{aff}} = 3.0$, $\omega_{\mathrm{inv}} = 1.2$ | 99.6 / +13.1 | 95.4 / -1.5 | 77.6 / -1.0 | 0.177 | 0.122 |
| | APG (Sadat et al., 2025) | $\omega = 1.5$ | 95.0 / +8.5 | 99.0 / +2.1 | 83.6 / +5.0 | 0.173 | 0.120 |
| | | $\omega = 3.0$ | 98.9 / +12.4 | 99.7 / +2.8 | 90.5 / +11.9 | 0.269 | 0.166 |
| | FAPG (Ours) | $\omega_{\mathrm{aff}} = 1.5$, $\omega_{\mathrm{inv}} = 1.2$ | 93.8 / +7.3 | 97.4 / +0.5 | 79.3 / +0.7 | 0.150 | 0.101 |
| | | $\omega_{\mathrm{aff}} = 3.0$, $\omega_{\mathrm{inv}} = 1.2$ | 99.5 / +13.0 | 94.9 / -2.0 | 76.2 / -2.4 | 0.195 | 0.136 |
| do(Male) | CFG | $\omega = 1.0$ | 86.6 / +0.0 | 91.8 / +0.0 | 79.8 / +0.0 | 0.115 | 0.079 |
| | | $\omega = 1.5$ | 93.3 / +6.7 | 97.2 / +5.4 | 82.0 / +2.2 | 0.158 | 0.111 |
| | | $\omega = 3.0$ | 98.2 / +11.6 | 99.2 / +7.4 | 90.2 / +10.4 | 0.267 | 0.171 |
| | FCFG (Ours) | $\omega_{\mathrm{aff}} = 1.5$, $\omega_{\mathrm{inv}} = 1.2$ | 88.3 / +1.7 | 93.8 / +2.0 | 78.9 / -0.9 | 0.149 | 0.101 |
| | | $\omega_{\mathrm{aff}} = 3.0$, $\omega_{\mathrm{inv}} = 1.2$ | 87.4 / +0.8 | 99.7 / +7.9 | 75.9 / -3.9 | 0.188 | 0.130 |
| | APG (Sadat et al., 2025) | $\omega = 1.5$ | 93.3 / +6.7 | 97.1 / +5.3 | 82.9 / +3.1 | 0.173 | 0.122 |
| | | $\omega = 3.0$ | 98.7 / +12.1 | 99.6 / +7.8 | 82.0 / +8.2 | 0.273 | 0.181 |
| | FAPG (Ours) | $\omega_{\mathrm{aff}} = 1.5$, $\omega_{\mathrm{inv}} = 1.2$ | 88.4 / +1.8 | 93.6 / +1.8 | 79.6 / -0.2 | 0.159 | 0.110 |
| | | $\omega_{\mathrm{aff}} = 3.0$, $\omega_{\mathrm{inv}} = 1.2$ | 87.5 / +0.9 | 99.7 / +7.9 | 75.0 / -3.8 | 0.206 | 0.146 |
| do(Young) | CFG | $\omega = 1.0$ | 87.5 / +0.0 | 95.7 / +0.0 | 62.3 / +0.0 | 0.115 | 0.085 |
| | | $\omega = 1.5$ | 95.6 / +8.1 | 99.3 / +3.6 | 66.3 / +4.0 | 0.166 | 0.110 |
| | | $\omega = 3.0$ | 98.5 / +11.0 | 99.9 / +4.2 | 77.7 / +15.4 | 0.261 | 0.160 |
| | FCFG (Ours) | $\omega_{\mathrm{aff}} = 1.5$, $\omega_{\mathrm{inv}} = 1.2$ | 90.0 / +2.5 | 96.7 / +1.0 | 67.4 / +5.1 | 0.147 | 0.100 |
| | | $\omega_{\mathrm{aff}} = 3.0$, $\omega_{\mathrm{inv}} = 1.2$ | 86.7 / -0.8 | 90.0 / -5.7 | 87.6 / +25.3 | 0.188 | 0.136 |
| | APG (Sadat et al., 2025) | $\omega = 1.5$ | 95.4 / +7.9 | 99.1 / +3.4 | 64.4 / +2.1 | 0.168 | 0.112 |
| | | $\omega = 3.0$ | 98.9 / +11.4 | 99.9 / +4.2 | 77.8 / +15.5 | 0.261 | 0.163 |
| | FAPG (Ours) | $\omega_{\mathrm{aff}} = 1.5$, $\omega_{\mathrm{inv}} = 1.2$ | 90.4 / +2.9 | 96.8 / +1.1 | 66.3 / +4.0 | 0.159 | 0.110 |
| | | $\omega_{\mathrm{aff}} = 3.0$, $\omega_{\mathrm{inv}} = 1.2$ | 85.6 / -1.9 | 90.0 / -5.7 | 85.7 / +23.4 | 0.207 | 0.149 |

*Figure A.24.* **Qualitative comparison of APG and FAPG for counterfactual generation and reversibility on CelebA.** For each row, we show the original image (left), the counterfactual generated under a single-attribute intervention (e.g., `do(Smiling)`, `do(Male)`, `do(Young)`), and the corresponding reversed image obtained by applying the inverse intervention. APG produces strong intervention effects but introduces unintended changes to non-intervened attributes, leading to degraded reversibility. In contrast, FAPG factors the guidance signal into intervened and invariant attribute groups, preserving the intended intervention strength while maintaining non-intervened attributes. As a result, FAPG yields more consistent reversed images across different interventions, demonstrating that the proposed factored guidance framework is compatible with APG and improves reversibility without intervention effectiveness.

