# OpenReview forum: "Factored Classifier-Free Guidance"
_ICML.cc/2026/Conference — ICML 2026 regular_

### Official Review · Reviewer_ZHkR · 2026-03-10

**Soundness:** 3
**Presentation:** 3
**Significance:** 3
**Originality:** 3
**Overall Recommendation:** 5
**Confidence:** 3

**Summary:**

This paper is about counterfactual image generation using classifier-free guidance in diffusion models. The authors show that standard CFG leads to poor counterfactual generation, due to the phenomenon of 'attribute amplification', which arises because of the single guidance weight used in CFG that is applied to all attributes. To combat this, they propose a method for 'factored CFG' that permits separate guidance weights per independent attribute (or per independent group of attributes, e.g. by grouping the intervened upon and unaffected attributes, and guiding those separately). FCFG leads to better counterfactual image generation on CelebA-HQ, and on two medical imaging tasks. FCFG can also be adopted in recent sampling methods for CFG, like CFG++ and ADP

**Compliance With Llm Reviewing Policy:**

Affirmed.

**Final Justification:**

The rebuttal addressed all my concerns. I was initially positive about the paper, and this helped confirm my position.

**Key Questions For Authors:**

- How do other, non CFG-based counterfacutal generation methods compare empirically to FCFG, like those already cited in the paper?
- How does the quality of images produced by FCFG compare to standard CFG, in terms of FID?
- How does the performance of FCFG vary as the degree of independence between attribute groups varies? Is it sensitive to this assumption being violated? This would be useful to known for potential applications beyond counterfactual generation, or with grouping other than 'affected' and 'unaffected'.
- What is the wallclock or memory overhead introduced into training/sampling by requiring multiple MLPs for condition embedding, and needing to calculate multiple score estimates per timestep during sampling?

**Limitations:**

Yes

**Strengths And Weaknesses:**

**Strengths**
- The idea behind FCFG is very simple and intuitive
- The elucidation of the problem of attribute amplification is compelling, making the drawbacks of vanilla CFG for this use case very clear
- The manuscript is well written, and largely easy to follow
- The empirical takeaways, in terms of counterfactual metrics/qualitative demonstrations, hold across different realistic imaging datasets
- FCFG has immediate compatibility with various sampling methods, including recent CFG extensions like CFG++ and APG, enhancing its applicability

**Weaknesses**
- Comparions with other counterfactual generation methods, besides CFG, are missing. This leaves the reader unsure as to whether FCFG leads to relatively good counterfactual data, and makes the significance of the method unclear. While the authors cite multiple other recent methods (diffusion and other) that permit counterfactual inference, e.g. [1,2,3], they offer no empirical comparisons.
- Quantification of general image quality using FCFG, e.g. via FID, alongside the counterfactual metrics, is missing. Image quality may degrade using FCFG compared to CFG because of the difference in the condition embedding mechanism, or if the assumed independence between attribute groups is violated, and this should be quantified.
- The qualitative takeaway about the middle row in Figure 2 are unconvincing. It is not clear to me that increasing the guidance in CFG "reduces Male expression" in this case.
- The graph in the bottom left of Figure 4 is incorrect. It is a duplicate of the graph from Figure 5.

---

[1] Pan, Yushu, and Elias Bareinboim. "Counterfactual image editing with disentangled causal latent space." The Thirty-ninth Annual Conference on Neural Information Processing Systems. 2025.

[2] Rasal, R. R., Kori, A., Ribeiro, F. D. S., Xia, T., and Glocker, B. Diffusion counterfactual generation with semantic abduction. In Forty-second International Conference on Machine Learning, 2025.

[3] Monteiro, M., Ribeiro, F. D. S., Pawlowski, N., Castro, D. C., and Glocker, B. Measuring axiomatic soundness of counterfactual image models. In The Eleventh International Conference on Learning Representations, 2023.

---

> ### Author Rebuttal · Authors · 2026-03-30
>
> We thank the reviewer for the constructive feedback. We are encouraged that the reviewer found FCFG simple and intuitive, the analysis of attribute amplification compelling, and the paper generally clear and easy to follow.
>
> **Q1. Comparisons with other counterfactual models**
>
> To strengthen the empirical evaluation, we have now **added comparisons with three baselines**: HVAE [1], HVAE-soft [2], and SA-DCG [3]. The main takeaway is consistent across datasets: compared with these methods, FCFG achieves a substantially better trade-off between strong target intervention and low off-target change, while remaining a simple inference-time plug-in. On CelebA, FCFG reduces attribute amplification relative to both CFG and SA-DCG while also improving reversibility. On MIMIC-CXR, FCFG achieves the strongest or comparable target intervention with dramatically lower off-target change than HVAE, HVAE-soft, and standard CFG. For space, we summarize only the main takeaway here; **we provide the full quantitative tables and discussion in our response to Reviewer 4uCy**.
>
> [1] Ribeiro et al., “High Fidelity Image Counterfactuals with Probabilistic Causal Models,” ICML 2023.
>
> [2] Xia et al., “Mitigating Attribute Amplification in Counterfactual Image Generation,” MICCAI 2024.
>
> [3] Rasal et al., “Diffusion Counterfactual Generation with Semantic Abduction,” ICML 2025.
>
>
> **Q2. Quantitative metrics for image quality**
>
> We did not originally report FID because we do not view it as a primary counterfactual metric: a good counterfactual should differ from the source in the intervened attribute, so closeness to the source distribution alone is not sufficient. We agree, however, that FID is a useful complementary measure of realism and consistency, and we therefore now include it in Table 4.
>
> **Table 4.** FID comparison of CFG and FCFG on CelebA-HQ. Lower is better.
> | Method | do(Smiling) | do(Male) | do(Young) |
> |---|---|---|---|
> | CFG, $ω= 2.0$ | 19.0 | 19.3 | 19.8 |
> | **Ours FCFG**, $ω_{\mathrm{aff}} = 2.0,\ ω_{\mathrm{inv}} = 1.2$ | **18.3** | **18.8** | **19.2** |
> | CFG, $ω= 3.0$ | 21.3 | 21.2 | 22.2 |
> | **Ours FCFG**, $ω_{\mathrm{aff}} = 3.0,\ ω_{\mathrm{inv}} = 1.2$ | **19.6** | **19.4** | **20.3** |
>
> Across interventions on CelebA-HQ, FCFG consistently achieves lower FID than CFG. Stronger guidance generally increases FID, likely because the counterfactuals move farther from the inputs, while FCFG lowers FID by reducing non-target amplification and making edits more localized. We will include these results in the revised version.
>
> **Q3. Qualitative takeaway of middle row in Fig. 2 is unclear**
>
> We agree that this might be a less straightforward qualitative example. The off-target amplification there concerns `Male` under `do(Young)`, which is visually hard to assess because both attributes affect the face in a relatively global way. We had hoped hair length might serve as a proxy cue, but this is too subtle. We have prepared an **[anonymous folder](https://anonymous.4open.science/r/ICML6996)** with several candidate replacements and will revise this row using a more visually direct example.
>
> **Q4. The left panel in Figure 4 is incorrect**
>
> Thank you for pointing this out. We mistakenly duplicated the left panel from Figure 5. In the revised version, we will replace it with the correct figure from the **[folder](https://anonymous.4open.science/r/ICML6996)**, which shows that standard CFG amplifies `density` under `do(circle)`.
>
> **Q5. Sensitivity to violations of the group-independence assumption**
>
> Our derivation is motivated by settings where grouped guidance terms are well separated. If this assumption does not hold exactly, the factorized decomposition is no longer exact. That said, some CelebA attributes, e.g. Male and Young, are correlated, so our experiments already reflect an approximate rather than ideal setting. Empirically, FCFG still improves over CFG in this regime, suggesting that moderate violations do not eliminate its practical benefit. More generally, as dependence between groups increases, the benefit of finer factorization is expected to diminish, and grouping strongly coupled attributes together provides a natural fallback that approaches standard CFG. We will clarify this point in the revised paper.
>
> **Q6. Wall-clock and memory overhead of factorized guidance.**
>
> We already discuss parameter and memory overhead in Appendix D (p. 18). The main takeaway is that the per-attribute MLPs are lightweight, so their parameter and memory cost is negligible relative to the diffusion backbone. The main extra cost appears at sampling time: standard CFG requires 2 score evaluations per timestep, whereas FCFG requires M+1, where M is the number of groups. In our two-group setting, this is 3 evaluations per timestep. Thus, training overhead is essentially unchanged, while sampling cost increases only modestly and linearly with the number of groups. We will clarify this more explicitly in the revised version.

---

> > ### Author Rebuttal · Reviewer_ZHkR · 2026-04-02
> >
> > I thank the authors for their efforts during the rebuttal phase, especially with the new experimental results and metrics. The comparisons with new baselines are good, and the addition of the FID metric helps to clarify how general image quality is maintained under guidance.
> >
> > My comments were all addressed, and I will raise my score.

---

> > > ### Author Response · Authors · 2026-04-03
> > >
> > > We sincerely thank the reviewer for the constructive comments and thoughtful feedback, which helped improve the paper. We are glad that the additional experiments, new baseline comparisons, and FID results addressed the concerns and clarified the contribution. We also greatly appreciate the reviewer’s positive reassessment.

---

### Official Review · Reviewer_4uCy · 2026-03-11

**Soundness:** 2
**Presentation:** 3
**Significance:** 3
**Originality:** 2
**Overall Recommendation:** 4
**Confidence:** 3

**Summary:**

This work presents a novel model-agnostic guidance technique called Factored Classifier-Free Guidance to overcome the limitations of the global guidance scale, providing attribute-wise control for generating counterfactuals for selective control. They propose using, as conditioning for the diffusion model, a concatenation of embeddings for each attribute and separate guidance weights for groups of them. Authors demonstrate that FCFG reduces attribute amplification and preserves reverse reconstruction against CFG and CFG++.

**Compliance With Llm Reviewing Policy:**

Affirmed.

**Key Questions For Authors:**

1. Do you have any ablation experiments demonstrating that p∅=0.5 improves identity preservation over standard conditioning dropout ($p_\emptyset=0.2$) or is based on qualitative observation? Does this affect in the same way both FCFG and CFG (and its variants, CFG++ and APG)?
2. Could you provide a comparative evaluation of your method against other diffusion-based counterfactual generation approaches, such as DiME [1]? This would help clarify the relative advantages of your approach.

[1] Jeanneret, Guillaume, Loïc Simon and Frédéric Jurie. "Diffusion models for counterfactual explanations." Proceedings of the Asian conference on computer vision. 2022.

**Limitations:**

Yes.

**Strengths And Weaknesses:**

## Soundness

The proposed approach is technically feasible, but some aspects of the methodology are not sufficiently clarified.

While the authors state *"we modify it to support CFG and FCFG"* and *"each conditioning attribute is projected via a dedicated MLP embedder,"* they fail to provide sufficient detail on whether the MLP embeddings are trained jointly with the diffusion model or independently as feature extractors feeding into a separate conditional network. Furthermore, while they say that conditioning dropout with $p_\emptyset=0.5$ improves identity preservation compared to standard $p_\emptyset=0.2$, there is no ablation study to validate this choice, nor any citation linking the parameter selection to prior work.

Lastly, the approach provides an intuitive way to enable attribute-level control through dedicated embeddings. While it improves on CFG in terms of identity preservation, it lacks direct comparisons with other state-of-the-art diffusion-based counterfactual methods. Without such benchmarks, the extent of its performance gains is difficult to fully evaluate.

## Significance
FCFG addresses the limitation of attribute amplification on current counterfactual generation methods by enabling interpretable control using factored guidance. In domains like healthcare or autonomous systems, where spurious correlations can lead to biased outcomes, such control is essential for trustworthy AI.

## Originality
The method is highly practical due to its compatibility with existing model-agnostic guidance frameworks such as CFG++ and APG, making it widely applicable. While it operates only within contexts where discrete attributes can be projected into independent conditional vectors, this flexibility enhances its applicability across diverse models and downstream tasks.

## Presentation
The paper presents a well-organized, technically clear narrative of the approach and its results. The primary source of confusion lies in the training procedure for the MLP embedders. Clarifying this would strengthen the reproducibility and understanding of the method’s design.

---

> ### Author Rebuttal · Authors · 2026-03-30
>
> We thank the reviewer for the careful and constructive review. We appreciate the reviewer’s recognition of the practicality, interpretability, and broad applicability of FCFG. Below we address the concerns regarding the training procedure of the MLP embedders, the conditioning-dropout choice, and comparison with other diffusion-based counterfactual methods.
>
> **Q1. Ablation study on CFG dropout rate.**
>
> We performed an ablation on the CFG dropout rate. Consistent with Rasal et al. (2025) [3], we find that using `p∅=0.5` improves identity preservation over the standard conditioning dropout rate `p∅=0.2`. Table 1 reports results for `do(Smiling)`. Lower reversibility MAE and LPIPS indicate better identity preservation.
>
> **Table 1.** Ablation study of $p_{\emptyset}=0.2$ vs. $p_{\emptyset}=0.5$ for `do(Smiling)`.
>
> | Guidance | MAE $↓$ ($p_{∅}=0.2$) | **MAE $↓$ ($p_{∅}=0.5$)** | LPIPS $↓$ ($p_{\emptyset}=0.2$) | **LPIPS $↓$ ($p_{∅}=0.5$)** |
> |---|---|---|---|---|
> | CFG | 0.309 | **0.263** | 0.180 | **0.155** |
> | FCFG | 0.206 | **0.177** | 0.140 | **0.122** |
> | CFG++ | 0.298 | **0.251** | 0.177 | **0.154** |
> | FCFG++ | 0.212 | **0.175** | 0.149 | **0.121** |
> | APG | 0.280 | **0.269** | 0.187 | **0.166** |
> | FAPG | 0.231 | **0.195** | 0.161 | **0.136** |
>
> Across all guidance variants,`p∅=0.5`yields lower MAE and LPIPS than `p∅=0.2`, supporting its use in our main experiments. We will include this ablation in the revised paper.
>
> **Q2. Clarification of the training procedure for the MLP embedders**
>
> All attribute-specific MLP embedders are trained jointly with the diffusion model end-to-end. They project each conditioning attribute into an embedding used directly by the diffusion model during training and sampling. We agree this was not stated clearly enough and will make it explicit in the revision.
>
> **Q3. Comparisons with other counterfactual models**
>
> We have now added comparisons with three counterfactual baselines: HVAE [1], HVAE-soft [2], and SA-DCG [3].
>
> On CelebA, we compare with Rasal et al. [3], reproduced from the official codebase and denoted SA-DCG. Because the released implementation conditions only on Smiling and Male, we evaluate intervention effectiveness on Smiling, off-target amplification on Male, and reversibility.
>
> **Table 2.** Comparison of CFG, SA-DCG, and FCFG on CelebA under `do(Smiling)`. AUC columns report differences relative to the CFG baseline with ω=1.0. ↑ indicates higher is better, and ↓ indicates lower is better.
> | Method | Target (Smiling) AUC ↑ | Non-target (Male) AUC ↓ | MAE ↓ | LPIPS ↓ |
> |---|---|---|---|---|
> | CFG, ω = 2.0 | +11.2 | +2.7 | 0.203 | 0.127 |
> | SA-DCG [3], ω = 2.0 | **+12.7** | +3.0 | 0.178 | 0.111 |
> | **FCFG, $\omega_{\mathrm{aff}} = 2.0,\ \omega_{\mathrm{inv}} = 1.2$** | +10.5 | **+0.4** | **0.146** | **0.098** |
> | CFG, ω = 3.0 | +12.3 | +3.0 | 0.263 | 0.155 |
> | SA-DCG [3], ω = 3.0 | +12.9 | +3.0 | 0.221 | 0.145 |
> | **FCFG, $\omega_{\mathrm{aff}} = 3.0,\ \omega_{\mathrm{inv}} = 1.2$** | **+13.1** | **-1.5** | **0.177** | **0.122** |
>
> These results show that SA-DCG, like standard CFG, still exhibits noticeable attribute amplification, although its diffusion autoencoder structure provides better reversibility than CFG. In contrast, FCFG reduces amplification while also improving reversibility over both CFG and SA-DCG. Notably, FCFG achieves this without relying on a diffusion autoencoder structure, which further demonstrates that the improvement comes from the proposed factorized guidance itself rather than from architectural changes.
>
> We also compare with Ribeiro et al. [1] and Xia et al. [2] on MIMIC-CXR, using the same train/test split as [1].
>
> **Table 3.** Comparison with Ribeiro et al. [1] and Xia et al. [2] on MIMIC-CXR. Higher target gain is better, while lower off-target change indicates less attribute amplification. CFG uses $\omega=3.0$, and FCFG uses $(\omega_{\mathrm{aff}}, \omega_{\mathrm{inv}})=(3.0, 1.2)$.
> | Intervention | Method | Target AUC ↑ | Off-target AUC ↓ |
> |---|---|---|---|
> | do(Sex) | HVAE [1] | +7.4 | +9.0 |
> | do(Sex) | HVAE-soft [2] | +7.3 | +5.7 |
> | do(Sex) | CFG | +7.5 | +10.7 |
> | do(Sex) | **Ours FCFG** | **+7.6** | **+0.6** |
> | do(Disease) | HVAE [1] | +17.1 | +5.1 |
> | do(Disease) | HVAE-soft [2] | +17.3 | +5.0 |
> | do(Disease) | CFG | +17.8 | +5.3 |
> | do(Disease) | **Ours FCFG** | **+18.8** | **+0.6** |
>
> The results show FCFG achieves the best trade-off between strong target intervention and low off-target change. In particular, it suppresses amplification much more effectively than HVAE, HVAE-soft, and standard CFG, while operating purely at inference time. We will include these results and discussion in the revised version.
>
> [1] Ribeiro et al., “High Fidelity Image Counterfactuals with Probabilistic Causal Models,” ICML 2023.
>
> [3] Xia et al., “Mitigating Attribute Amplification in Counterfactual Image Generation,” MICCAI 2024.
>
> [3] Rasal et al., “Diffusion Counterfactual Generation with Semantic Abduction,” ICML 2025.

---

> > ### Author Rebuttal · Reviewer_4uCy · 2026-04-03
> >
> > The authors have adequately addressed my concerns in the rebuttal. The added ablation on the CFG dropout rate provides clear quantitative evidence that p∅=0.5 improves identity preservation, with consistent effects across multiple guidance variants. Additionally, including further baselines strengthens the empirical evaluation and better positions the method relative to prior diffusion-based approaches.
> >
> > I keep my score unchanged, as my original evaluation already assumed these issues were likely addressable, and my remaining uncertainty is mainly due to limited familiarity with closely related work rather than unresolved problems.

---

> > > ### Author Response · Authors · 2026-04-03
> > >
> > > We sincerely thank the reviewer for the constructive comments and positive assessment of our work. We are pleased that our rebuttal addressed the concerns, and we appreciate the reviewer’s recognition that the added dropout ablation and baseline comparisons strengthen the paper.
> > >
> > > Regarding the reviewer’s remaining uncertainty, which they attribute mainly to limited familiarity with closely related work rather than unresolved issues with the paper, we would like to take this opportunity to further clarify the positioning of our contribution relative to prior methods. To the best of our knowledge, this is the **first** work to explicitly identify this CFG-specific failure mode in counterfactual generation, namely that a single global guidance weight can couple intervened and non-intervened factors and thereby induce attribute amplification. Our contribution is twofold: (1) we are the **first** to identify and characterize this largely overlooked limitation of standard CFG, and (2) we propose a simple and effective factorized guidance solution.
> > >
> > > The reviewer’s feedback has helped us improve both the clarity of the methodology and the positioning of the paper, and we will incorporate these clarifications and additional results in the revised version.

---

### Official Review · Reviewer_egEu · 2026-03-11

**Soundness:** 3
**Presentation:** 2
**Significance:** 2
**Originality:** 2
**Overall Recommendation:** 4
**Confidence:** 2

**Summary:**

The authors address an increasingly acknowledged problem in diffusion-based counterfactual image generation: edits also change attributes that a reasonable evaluator would hold fixed under the edit. The authors identify that the global guidance signal used in standard Classifier Free Guidance (CFG) confounds all the attributes, and propose an inference-time fix to decouple the guidance for the attributes that should and that needn't remain invariant under the edit (using a causal DAG as input). This is achieved using factorized diffusion guidance terms. Applying this to CelebHQ and a medical-imaging dataset shows that this is a plug-in method that can (in combination with other performance improvement methods) improve causal faithfulness of generations without requiring fine-tuning or latent representation learning.

**Compliance With Llm Reviewing Policy:**

Affirmed.

**Final Justification:**

I have updated from (2) to (4) for addressing my main concern about the technical novelty of the methods. My remaining reservations are about the quality of the empirical results shown. II am not as familiar with the two new baseline papers compared to - this is reflected in my lower confidence (2).

**Key Questions For Authors:**

Please address the two main weaknesses indicated above:
1. Technical comparison to prior work that does this using causal DAGS to inform which variables remain invariant under edit, and specifically methods that do this at inference-time.
2. How does this method compare to other competing methods for de-confounding using a causal DAG (or similar causal assumptions)?

**Limitations:**

Yes.

**Strengths And Weaknesses:**

**Strengths**
1. This is indeed a topical concern and shows a clean, simple way to translate causal graphical assumptions into guidance of which attributes should remain invariant under an edit.
2. The edits are at inference-time and don't require retraining or representation learning

**Weaknesses**
My main objection is that the theoretical contributions are not significant, and the empirical results are not terribly impressive.

1. Theory: a more direct *theoretical* comparison of this method with prior work on causally-informed counterfactual generation using diffusion models; some of the references are mentioned, but I still can't tell if there is novelty in this approach - fixing some variables to be invariant does not seem exciting. See Pan & Bareinboim (2024, 2025) for instance on using causal graphs for causal-validity of edits by holding upstream variables fixed. See also for instance, Spyrou et al (2025) on Causally Steered Diffusion that also does this at inference time, is model-agnostic, and does not require fine-tuning etc. Abduction-Action-Inference based generation has a well-established body of work by now. Comparing to some of these would be needed to convince that there is *theoretical* novelty going on.

2. Results:
- The baseline being standard CFG is quite insufficient. Certainly, we would expect confounding because of statistical artefacts in the dataset - this is not new. This method should be compared to *other methods that address this problem*, to show that this is a compelling, more efficient, more effective alternative to causal fine-tuning, or learning disentangled causal representations, or what have you. It is especially convincing if you can show that some of these properties you show (e.g. compatibility with advanced guidance schemes) are not found for competing methods.
- The results themselves are visually not super convincing. Fig 2 - only row 3 offers compelling evidence of this de-confounding. I have to squint to notice a difference in row 2 - that there is an AUROC difference leads me to suspect that the "ground truth" classifier is itself perhaps being affected by confounding? Some artefact unrelated to "Male" attribute in row 2 is being interpreted as a decrease in "Male" for CFG.

---

> ### Author Rebuttal · Authors · 2026-03-30
>
> We thank the reviewer for the careful reading and thoughtful comments. We appreciate that the reviewer recognized the practical motivation of the problem, as well as the simplicity and inference-time nature of our method. Below we clarify the novelty of our contribution and address the empirical concerns.
>
> **Q1. Theoretical novelty relative to prior causal-DAG counterfactual diffusion methods**
>
> Our contribution is to identify a previously overlooked failure mode of standard CFG and to propose a simple, effective remedy, supported by comprehensive validation. **To the best of our knowledge, this CFG-specific failure mode has not been identified in prior work, and no existing method is specifically designed to address it. We therefore believe that the first identification and systematic treatment of this problem is itself a meaningful contribution.**
>
> The cited prior works operate at a different level. Pan & Bareinboim (2024) formalize counterfactual image editing in an ASCM framework, emphasize its identifiability limits, and develop causal generative models to approximate counterfactual-consistent edits. Pan & Bareinboim (2025) instead focus on learning a backdoor-disentangled causal latent space from a pretrained diffusion model for counterfactual editing. Spyrou et al. (2025) perform causal counterfactual video generation by optimizing prompts around a frozen black-box video diffusion editor using VLM-based feedback. By contrast, our method targets the guidance mechanism itself: even with a known causal graph, global CFG can amplify intervened and non-intervened attributes together. We address this through a simple inference-time factorization of the guidance rule, while leaving the trained model unchanged. We will revise the paper to make this distinction clearer and add a direct paragraph discussing these related works.
>
> **Q2. Confounding due to statistical artefacts in the dataset.**
>
> We do not study “confounding due to statistical artefacts in the dataset.” **Rather, the phenomenon studied here, namely attribute amplification, is induced by the global guidance mechanism of standard CFG itself**. This is precisely why standard CFG was the relevant baseline in our setting.
>
> **Q3. Comparison with other counterfactual methods**
>
> To strengthen the empirical evaluation, we have now **added comparisons with three counterfactual baselines**: Ribeiro et al. 2023 [1], a causal framework based on HVAE for high-fidelity counterfactual generation; Xia et al. 2024 [2], which extends [1] with soft-label supervision to mitigate amplification; and Rasal et al. 2025 [3], a diffusion-autoencoder-based method with semantic abduction. All of them use a causal DAG.
>
> The main takeaway is consistent across datasets: compared with these methods, FCFG achieves a substantially better trade-off between strong target intervention and low off-target change, while remaining a simple inference-time plug-in. On CelebA, FCFG reduces attribute amplification relative to both CFG and SA-DCG while also improving reversibility. On MIMIC-CXR, FCFG achieves the strongest or comparable target intervention with dramatically lower off-target change than HVAE, HVAE-soft, and standard CFG. For space, we summarize only the main takeaway here; **we provide the full quantitative tables and discussion in our response to Reviewer 4uCy**.
>
> [1] Ribeiro et al., “High Fidelity Image Counterfactuals with Probabilistic Causal Models,” ICML 2023.
>
> [3] Xia et al., “Mitigating Attribute Amplification in Counterfactual Image Generation,” MICCAI 2024.
>
> [3] Rasal et al., “Diffusion Counterfactual Generation with Semantic Abduction,” ICML 2025.
>
> **Q4. Qualitative evidence in Fig. 2**
>
> The purpose of Fig. 2 is not to demonstrate overall fidelity or realism, but to visualize off-target attribute amplification under standard CFG and its mitigation by FCFG. We believe the top row of Fig. 1 already shows this clearly: under do(Male=No) increasing CFG makes the edited samples appear more Smiling=Yes, even though Smiling is not part of the intervention.
>
> The middle row of Fig. 2 is a less straightforward example. There, the off-target amplification concerns Male under do(Young), which is difficult to assess visually because both attributes affect the face in a relatively global way. We had hoped hair length might act as a proxy cue for Male, but we agree this is too subtle. We also do not interpret the classifier-based result as perfect ground truth in isolation, but as an operational diagnostic of off-target attribute shift. To strengthen the qualitative evidence, we have prepared an **[anonymous folder](https://anonymous.4open.science/r/ICML6996)** with several candidate replacements and will revise this row using a more visually direct example.

---

> > ### Author Rebuttal · Reviewer_egEu · 2026-04-03
> >
> > Thanks to the authors for the clarifications and the extra baseline comparisons, this is appreciated.
> >
> > I am likely going to increase my score, based on the new comparisons. I am trying to assess how exciting the technical contribution is. To my mind, the key novelty lies in Eq 10, where you split the guidance parameter for upstream vs downstream split, as a way of incorporating the DAG assumptions, as opposed to some other ways of incorporating this same split. Could you perhaps share a sample of what you would say in your edits when comparing this to other ways of incorporating the causal knowledge?
> >
> > Further, could you summarize any other reasons, besides the performance table, why this method might be desirable compared to the baselines you have chosen? Time, memory, or what have you. This would help me make my determination. Thanks!

---

> > > ### Author Response · Authors · 2026-04-06
> > >
> > > Thank you very much for this helpful follow-up.
> > >
> > > # Q1. Positioning of FCFG relative to prior methods.
> > >
> > > We would like to clarify that the main novelty is not just Eq. 10, but that we are, to the best of our knowledge, **the first to identify this CFG-induced failure mode** in counterfactual generation, and **the first to propose a simple yet effective method to address it**. Concretely, we show that standard CFG can amplify intervened and non-intervened attributes together, even when the causal graph is known, thereby violating the intended causal graph.
> > >
> > > Eq. 10 should be viewed as the theoretical justification for a general group-wise guidance formulation. It is more general than the specific two-group counterfactual setting used in our main experiments. The causal-graph-based split into intervened and unaffected attribute groups is one natural special case of this formulation, under the conditional independence assumption in Proposition 3.1.
> > >
> > > A sample of the text we would add in the revision is:
> > >
> > > >Earlier SCM-based works formulated counterfactual inference through the standard abduction–action–prediction pipeline, instantiated with generative models such as normalizing flows and VAEs/HVAEs [e.g., Pawlowski et al., 2020; De Sousa Ribeiro et al., 2023; Xia et al., 2024]. Pan and Bareinboim (2024) instead focus on the causal formalization of counterfactual image editing and the identifiability of the editing target.
> > >
> > > > More recently, diffusion models have become a common backbone for counterfactual image generation. Pan and Bareinboim (2025) focus on learning a disentangled causal latent space from a pretrained diffusion model for editing. Spyrou et al. (2025) focus on prompt-based causal steering of a frozen black-box video diffusion editor via VLM feedback. Rasal et al. (2025) use a diffusion-autoencoder-based framework with semantic abduction for counterfactual generation. Many diffusion-based methods use classifier-free guidance (CFG) to improve counterfactual effectiveness, i.e., the extent to which the generated counterfactual reflects the intended intervention [e.g., Komanduri et al., 2024; Pérez-García et al., 2024; Kumar et al. 2025; Rasal et al., 2025; Pan and Bareinboim 2025; Spyrou et al. 2025].
> > >
> > > > **Despite the widespread use of CFG, we identify a previously overlooked failure mode of standard global CFG in counterfactual generation: a single shared guidance weight can amplify intervened and non-intervened attributes together, thereby violating the intended causal graph.**
> > >
> > > > We claim that this failure mode is induced by the guidance mechanism itself, rather than being merely a consequence of dataset artefacts, spurious correlations, or causal-graph misspecification. To address this off-target amplification, we propose a simple yet effective inference-time method that factorizes the guidance signal across attribute groups, allowing intervention-relevant and intervention-invariant factors to be controlled separately, with the grouping optionally informed by prior causal or semantic structure. The method is model-agnostic, does not require retraining or architectural modification, and can be seamlessly integrated with advanced guidance schemes such as CFG++ and APG.
> > >
> > > > **To the best of our knowledge, this CFG-specific failure mode has not been identified in prior work, and our method is the first designed specifically to address it.**
> > >
> > >
> > > # Q2. Practical advantages beyond the performance table.
> > >
> > > Beyond the performance table, we believe FCFG is attractive for several practical reasons. Compared with Ribeiro et al. (2023) and Xia et al. (2024), FCFG does not require an additional predictor-based fine-tuning stage or auxiliary pretrained predictors. Compared with Rasal et al. (2025), FCFG avoids a heavier autoencoding architecture and additional identity-preserving inference-time optimization, while still achieving better reversibility in our comparisons. It is also lighter and faster in practice, as summarized below.
> > >
> > > | Method | Parameters | CF generation time (128 samples) | Target AUC ↑ | Off-target AUC ↓ | MAE ↓ | LPIPS ↓ |
> > > |---|---|---|---|---|---|---|
> > > | Rasal et al. (2025) | 72.6M | 180 s | +12.9 | +3.0 | 0.221 | 0.145 |
> > > | FCFG | **60.5M** | **140 s** | **+13.1** | **-1.5** | **0.177** | **0.122** |
> > >
> > > This difference is because FCFG only modifies the guidance computation at inference time, whereas Rasal et al. combines standard global CFG with a heavier architecture and inference-time null-token optimization.
> > >
> > > Overall, FCFG offers a favourable simplicity-control trade-off: it is model-agnostic, inference-time only, does not require retraining or architectural modification, and can be integrated directly with advanced guidance schemes including CFG++ and APG.
> > >
> > > We are truly grateful for the reviewer’s continued engagement, which has helped us refine both the positioning of the work and the presentation of its contribution. We hope this clarification is useful for the reviewer’s final assessment.

---

### Official Review · Reviewer_ut8m · 2026-03-13

**Soundness:** 3
**Presentation:** 3
**Significance:** 3
**Originality:** 2
**Overall Recommendation:** 4
**Confidence:** 2

**Summary:**

This paper introduces Factored Classifier-Free Guidance (FCFG), as an inference-time technique to improve counterfactual image generation using diffusion models. The authors first discuss a key failure mode in non-factored Classifier-Free Guidance (CFG), which applies a single global guidance scale, inadvertently amplifying attributes that should remain invariant under a causal intervention. Intuitively, CFG deploys a diffusion model with the conditional distribution of the image given the context re-weighted (the probability of the context given the image receiving an exponential factor in an application of Bayes rule).

To resolve this limitation, FCFG partitions the probability of the conditioning attributes into disjoint groups (e.g., intervened on vs. invariant under intervention) and assigns separate guidance weights to each group. Selection of the weights is done manually/empirically (presumably, through grid search to optimize the target objectives). Work is validated on CelebA-HQ, EMBED, and MIMIC-CXR datasets, with FCFG seeming to improve intervention effectiveness and better preservation of non-intervened attributes compared to standard CFG. The approach requires architectural modifications, so it can be applied broadly to diffusion modeling setups.

**Compliance With Llm Reviewing Policy:**

Affirmed.

**Key Questions For Authors:**

## Question 1

In standard CFG, the null token $\emptyset$ (a single null vector) replaces the *entire* conditioning vector. The neural network is explicitly trained to expect this all-zero condition. In FCFG, we seem to construct "masked" embeddings where *only some* attributes are zeroed out while others are active (Eq. 12).

Because the network wasn't explicitly trained on these partially-masked conditions (is that correct?), feeding them into the network at inference time could cause the magnitudes of the activations to be structurally different from what the normalization layers saw during training. This can lead to out-of-distribution feature shifts, potentially creating artifacts. The authors essentially assume the network will interpolate these partial nulls smoothly, but mathematically, normalization layers can be highly sensitive to exactly this kind of masking. Can the authors elaborate on whether / when this would be a potential threat?

## Question 2

Do you have a sense for how CFG vs. FCFG behave when the conditional independence assumption is heavily violated (e.g., when attributes are highly entangled)?

## Question 3

What exactly makes the generated changes in MIMIC-CXR "clinically meaningful"? (I might take that statement out, unless the clinical implications of counterfactual image generation can be more clearly articulated...consider removing).

**Limitations:**

Yes.

**Strengths And Weaknesses:**

## Strengths

There are aspects of the paper I think are strengths.

The problem identified (attribute amplification in CFG during counterfactual generation) is overall well-articulated and cleanly stated,.

The proposed solution, FCFG, is methodologically simple, operates entirely at inference time, and is model-agnostic, meaning it can be plugged into existing diffusion models trained with standard CFG, all of which are real benefits. As a reader, I do get the sense that I take away a real/tangible insight from the paper about forward conditional diffusion, which I view to be an important asset of the paper.

The evaluation is also thorough, spanning both natural images (CelebA-HQ) and medical imaging applications (EMBED and MIMIC-CXR). The evaluations are also fair and careful --- not just hyper-parameter optimizing FCFG (but also CFG, the baseline).

## Weaknesses

There are aspects of the paper I think might be weaknesses.

* The paper struggles a bit with novelty framing. Since CFG is a cornerstone of modern diffusion models, it's going to take robust justification to convince readers that this factored approach is a sufficiently novel leap. The prior work section exhibits some slippage between what is truly new and what is established, which muddies the contributions.

* The reliance on the conditional independence assumption for the proxy posterior is restrictive.  If we consider highly entangled or subjective concepts (like "sentiment" or age/disease correlated factors), this assumption likely breaks down, limiting the scope of the method to arbitrary counterfactual image generation. Indeed, Proposition 3.1 mathematically requires the attribute groups to be conditionally independent given the latent image representation $x_t$. If you were trying to manipulate a vague or holistic concept like "sentiment" (e.g., making an image "happier"), that concept is possibly entangled with other attributes (facial expression, posture, etc.).
If the independence assumption is violated, the factorized proxy posterior doesn't hold cleanly, so the guidance directions will presumably interfere with each other. This could cause the very "attribute amplification" or manifold drift that the paper is trying to solve. This issue seems to limit the scope to strictly definable, disentangled causal graphs.

*  Images have extremely high dimensionality, and the "important" dimensions are rarely known *a priori*. Defining neat causal groups for masking assumes we have a good semantic handle on the latent space, which might be difficult to obtain in practice. Relatedly, it is not clear how I know the assumed causal graph that generates subgroups in the factorization, and it seems to require an interpretability step not fully explicated in the manuscript.

In general, the FCFG, as I see it, is no "catch-all" solution for counterfactual image generation but seems like a potentially useful approach in a subset of generation tasks/cases involving a subset of known/meaningful categories.

## Minor points

* Phrasing around a "single guidance scale" in the abstract and introduction could be more clearly explained. For readers who aren't familiar with CFG, it would be beneficial to introduce the $\omega$ (guidance weight) notation and conceptually explain the weighting mechanism earlier in the text.

* "Effectiveness" is the chosen term for one of the key evaluation metrics. This might be standard in this area, but here, with causal inference contexts implicated, the meaning is arguably overloaded (i.e., usually has meaning regarding treatment effect size).

---

> ### Author Rebuttal · Authors · 2026-03-30
>
> We thank the reviewer for the detailed and constructive review. We are glad that the reviewer found the attribute-amplification problem to be clearly articulated, viewed FCFG as a simple practical inference-time method, and noted that the paper provides a “real/tangible insight” into forward conditional diffusion. We also appreciate the reviewer’s recognition that the evaluation is thorough and fair across both natural-image and medical-imaging datasets. Below we address the comments.
>
> **Q1 Unclear novelty framing**
>
> The novelty of our paper can be expressed simply: we identify a problem that has been largely overlooked despite the wide use and popularity of standard CFG, and we propose a simple yet effective solution with comprehensive validation. Specifically, standard global CFG uses one shared guidance weight across all conditioning attributes, coupling intervened and non-intervened factors during counterfactual generation. **To the best of our knowledge, this CFG-specific failure mode has not been explicitly identified or directly addressed in prior work.** Our solution is a factorized inference-time guidance scheme that decouples these effects without retraining. We will revise the paper to make this contribution much clearer.
>
>
> **Q2 The reliance on the conditional independence assumption / Behavior under highly entangled attributes**
>
> Some concepts can be highly entangled or subjective in practice. However, Proposition 3.1 only requires conditional independence across different groups, not among variables within the same group. Therefore, strongly entangled concepts are not meant to be separated into different guidance groups. For example, if one models “happy” as a parent of “facial expression,” then intervening on “happy” would also affect “facial expression,” so these variables would belong to the same intervened group and be assigned the same guidance strength. We also note that this assumption is often only approximate in practice. For example, in CelebA-HQ, Male and Young exhibit a moderate Pearson correlation (𝜌=−0.33), yet FCFG still improves over standard CFG in this regime. This suggests moderate violations do not necessarily remove the practical benefit of factorized guidance. However, if two variables are treated as independent when they are in fact highly entangled, then the assumption in Proposition 3.1 is violated, the factorized proxy posterior becomes less accurate, and the effectiveness of FCFG may be reduced. In such cases, a safer choice is to place strongly coupled variables in the same group rather than separate them. We will revise the paper to clarify this point and make the intended scope more explicit.
>
> **Q3 The reliance on a pre-specified causal graph**
>
> Specifying an appropriate causal graph or semantic grouping can be difficult in practice, especially in high-dimensional image domains. FCFG does not solve this upstream problem. Rather, it assumes access to structured conditioning variables with an assumed causal grouping or graph, and studies how to use that structure for counterfactual generation via factorized guidance. Discovering such a causal graph is an orthogonal problem, known as causal discovery. We will revise the manuscript to make this assumption explicit and clarify FCFG’s scope.
>
> **Q4 Potential distribution shift from partial-mask inference**
>
> Thank you for this insightful question. We do not explicitly train on partially masked conditions; this is intentional, since we keep the training setup identical to standard CFG rather than introducing an FCFG-specific training scheme. At inference time, FCFG constructs partially masked embeddings by null-tokenizing selected attribute groups while leaving others active. In principle, this introduces a train-test mismatch, and partial masking could potentially cause the kind of normalization-sensitive feature shifts the reviewer mentions. However, we do not observe systematic artifacts or instability from this effect in our experiments; samples remain visually coherent while improving control over intervened versus invariant attributes. This is an empirical observation rather than a formal guarantee, and the risk could become more pronounced with more groups, stronger guidance weights, or a more severe train-test mismatch. We will revise the paper to state this limitation more explicitly.
>
> **Q5 Clinical meaning of the MIMIC-CXR changes**
>
> The phrase “clinically meaningful” is indeed too strong. What we intended to convey is only that, in the MIMIC-CXR setting, the generated edits are better aligned with the target clinical labels and show improved preservation of non-intervened factors relative to standard CFG. We will revise the manuscript accordingly and replace this phrase with a more precise description.
>
> **Minor comments about phrasing**
>
> These  will be addressed in revision.

---

> > ### Author Rebuttal · Reviewer_ut8m · 2026-04-04
> >
> > Thanks for taking the time to write out these helpful thoughts. I am inclined to maintain the broadly positive score -- there seem to be strengths of the proposed method, but at the cost of more assumed structure (i.e., investigators need to know how to specify both the nodes and edges of a desired DAG [edges and nodes] underlying the relevant counterfactual generation process).

---

> > > ### Author Response · Authors · 2026-04-04
> > >
> > > We sincerely thank the reviewer for the constructive comments and positive assessment of our work. We agree that the need for a pre-defined causal graph or semantic grouping is an important limitation to acknowledge. We would also like to note that this is a **common upstream assumption** in many prior counterfactual generation works that rely on structured causal information, including, for example, Pawlowski et al. (2020), De Sousa Ribeiro et al. (2023), Pan and Bareinboim (2024), Rasal et al. (2025), etc.
> > >
> > > Our contribution is **orthogonal** to this limitation: to the best of our knowledge, we are the first to identify that **even when such a causal graph or semantic grouping is already available**, standard global CFG can still lead to off-target amplification, because one shared guidance weight couples intervened and non-intervened factors during sampling. FCFG addresses this specific problem at the level of inference-time guidance.
> > >
> > > We hope this helps further clarify both the intended scope of FCFG and the specific contribution of our work.

---

### Decision · Program_Chairs · 2026-04-30

**Decision:**

Accept (regular)

**Comment:**

**Summary and Decision**
While reviewers noted minor concerns regarding the potential significance, the general consensus is that the paper proposes a simple yet effective method for counterfactual generation via factored classifier-free guidance.

The reviewers specifically highlighted the following strengths:
* Simple approach for counterfactual generation that is model-agnostic.
* Strong evaluations on both natural images and medical imaging applications.
* Carefully explained novelty w.r.t. prior works that incorporate DAG assumptions.

**Rebuttal and Discussion**
During the discussion phase, the authors addressed concerns about differences with prior work by explaining how their work incorporates DAG assumptions differently, which satisfied the reviewers. Given the potential impact of the work in this particular subfield, the Area Chair recommends acceptance.

**Final Instructions to Authors**
Please ensure that the promised revisions from the rebuttal are incorporated into the camera-ready version.